# A Single-Loop Gradient Algorithm for Pessimistic Bilevel Optimization via Smooth Approximation

**Qichao Cao[1]**
caoqc2024@mail.sustech.edu.cn

**Shangzhi Zeng[2,1]**
zengsz@sustech.edu.cn

**Jin Zhang [1,2,3,*]**
zhangj9@sustech.edu.cn

## Abstract

Bilevel optimization has garnered significant attention in the machine learning community recently, particularly regarding the development of efficient numerical methods. While substantial progress has been made in developing efficient algorithms for optimistic bilevel optimization, the study of methods for solving Pessimistic Bilevel Optimization (PBO) remains relatively less explored, especially the design of fully first-order, single-loop gradient-based algorithms. This paper aims to bridge this research gap. We first propose a novel smooth approximation to the PBO problem, using penalization and regularization techniques. Building upon this approximation, we then propose SiPBA (Single-loop Pessimistic Bilevel Algorithm), a new gradient-based method specifically designed for PBO which avoids second-order derivative information or inner-loop iterations for subproblem solving. We provide theoretical validation for the proposed smooth approximation scheme and establish theoretical convergence for the algorithm SiPBA. Numerical experiments on synthetic examples and practical applications demonstrate the effectiveness and efficiency of SiPBA.

## 1 Introduction

Bilevel optimization constitutes a hierarchical optimization problem formulated as follows:

$$\min_{x \in X} \ F(x,y) \quad s.t. \quad y \in \mathcal{S}(x) := \operatorname*{arg\,min}_{y' \in Y} f(x,y'),$$

where $x \in \mathbb{R}^n$ and $y \in \mathbb{R}^m$ represent the upper-level and lower-level decision variables, respectively, and $X \subseteq \mathbb{R}^n$ and $Y \subseteq \mathbb{R}^m$ are closed convex sets. The functions $F : \mathbb{R}^n \times \mathbb{R}^m \to \mathbb{R}$ and $f : \mathbb{R}^n \times \mathbb{R}^m \to \mathbb{R}$ are the upper-level and lower-level objective functions, respectively. Bilevel optimization naturally models non-cooperative game between two players, often referred to as a Stackelberg game [66]. When the lower-level problem in bilevel optimization admits multiple optimal solutions for a given $x$, the corresponding decision variable $y$ in the upper-level objective becomes ambiguous. To resolve this, bilevel optimization is commonly formulated in two distinct settings: Optimistic Bilevel Optimization (OBO) and Pessimistic Bilevel Optimization (PBO).

---

[1]Department of Mathematics, Southern University of Science and Technology, Shenzhen, China.

[2]National Center for Applied Mathematics Shenzhen, Southern University of Science and Technology, Shenzhen, China.

[3]Detection Institute for Advanced Technology Longhua-Shenzhen (DIATLHSZ), Shenzhen, China.

[*]Corresponding Author.

39th Conference on Neural Information Processing Systems (NeurIPS 2025).

In the OBO setting, it is assumed that the lower-level selects a solution $y \in \mathcal{S}(x)$ that is most favorable to the upper-level's objective $F$. The OBO formulation is thus:

$$\min_{x \in X} \min_{y \in \mathbb{R}^m} F(x, y) \quad s.t. \quad y \in \mathcal{S}(x),$$

Conversely, the PBO setting considers a cautious or adversarial scenario where the lower-level is assumed to choose a solution $y \in \mathcal{S}(x)$ that is least favorable to the upper-level. The PBO formulation is:

$$\min_{x \in X} \max_{y \in \mathbb{R}^m} F(x, y) \quad s.t. \quad y \in \mathcal{S}(x),$$

Therefore, OBO models scenarios predicated on cooperative or aligned lower-level responses, whereas PBO is essential when robustness against worst-case outcomes, often encountered under uncertainty or in adversarial contexts, is required.

In recent years, bilevel optimization has garnered substantial interest within the machine learning community, finding applications in areas such as hyperparameter optimization [25], adversarial learning [72], reinforcement learning [75], and meta-learning [28], among others, where first-order gradient-based methods are preferred for their efficiency and scalability[15, 41, 63].

Much of the existing bilevel research focuses on the OBO case, for which numerous fully first-order gradient-based algorithms suitable for large-scale machine learning tasks have been developed [38, 42, 45, 61], often by reformulating the problem using Karush-Kuhn-Tucker (KKT) conditions or through value-function-based constraints. While OBO benefits from a well-established algorithmic toolkit, PBO remains comparatively underexplored from an algorithmic standpoint. PBO offers a robust framework for leaders concerned with worst-case follower responses and a growing body of work has explored the potential of PBO in various machine learning applications such as adversarial learning [16, 13], contextual optimization [17, 36] and hyperparameter optimization [64]. Outside the machine learning domain, PBO has found applications in many other practical scenarios, including but not limited to demand response management[37], rank pricing and second-best toll pricing [18, 9], production-distribution planning [74], and gene knockout model[71]. Yet, the inherent max-structure at the upper level of PBO creates a more complex, three-level-like structure (min-max-min), making the direct application of gradient-based techniques developed for OBO challenging. Although several PBO single-level reformulations have been proposed [67, 71, 11] , their intricate structures continue to pose difficulties for the development of fully first-order gradient-based solution methods. Recently, [31] proposed AdaProx, a gradient-based method for PBO. This AdaProx method employs a double-loop procedure and requires second-order derivative information. This motivates our central research question:

**Can we design a fully first-order single-loop gradient-based algorithm for PBO?**

This paper demonstrates that the answer is affirmative. We approach PBO by reformulating it as the minimization of a value function:

$$\min_{x \in X} \phi(x), \quad \text{where } \phi(x) := \max_{y \in \mathbb{R}^m} \left\{ F(x, y) \quad \text{s.t. } y \in \mathcal{S}(x) \right\}. \tag{1}$$

As indicated by the formulation in (1), the PBO can be solved by minimizing the function $\phi(x)$. However, $\phi(x)$ is the value function of a maximization problem whose feasible region depends on the solution set of another optimization problem. Consequently, $\phi(x)$ is generally non-smooth [32], and evaluating its value and gradient (or subgradient) poses significant computational challenges. The non-smoothness of $\phi(x)$ constitutes a primary challenge in solving PBO, rendering the direct minimization of $\phi(x)$ difficult.

To surmount the challenge posed by the non-smoothness of $\phi(x)$, we introduce a smooth approximation of $\phi(x)$ by employing penalization and regularization techniques. This transforms the PBO into a tractable, smooth optimization problem, enabling the application of efficient gradient-based methods. However, calculating the gradient of this smooth approximation function requires solving an associated minimax subproblem to find its saddle point, which can be computationally demanding and complicate the implementation of gradient-based methods. To address this complexity, we propose a one-step gradient ascent-descent update strategy to obtain an inexact saddle point solution. This inexact solution is then used to construct an inexact gradient for the minimization of the smoothed objective. Through this approach, we propose SiPBA (Single-loop Pessimistic Bilevel Algorithm), a novel fully first-order single-loop gradient-based algorithm designed to solve PBO problem (1).

## 1.1 Contribution

This paper presents the following key contributions to the study of PBO problem:

**New Smooth Approximation for PBO:** We introduce a novel smooth approximation for PBO. This is achieved by constructing a continuously differentiable surrogate for the potentially non-smooth value function $\phi(x)$, using penalization and regularization techniques. Based on this, we formulate a smooth approximation problem corresponding to the original PBO. The validity of this smooth approximation is rigorously established by demonstrating the asymptotic convergence of the solutions of the smoothed problem to those of the original PBO. These results are detailed in Section 2.

**Single-Loop Algorithm (SiPBA) and Theoretical Guarantees:** Building upon the proposed smooth approximation, we develop SiPBA (Single-loop Pessimistic Bilevel Algorithm). SiPBA is a gradient-based algorithm designed for solving PBO problems, which avoids the computation of second-order derivatives and eliminates the need for iterative inner-loop procedures to solve subproblems (Section 3). We provide a rigorous convergence analysis of SiPBA in Section 4. This analysis includes the derivation of non-asymptotic convergence rates for relevant error metrics and establishes guarantees for achieving a relaxed stationarity condition for the iterates generated by the algorithm.

**Empirical Validation:** The practical effectiveness and computational efficiency of the proposed SiPBA algorithm are validated through numerical experiments, presented in Section 5. We evaluate SiPBA across synthetic problems, email spam classification, and hyper-representation learning. The results provide empirical evidence supporting the competitive performance of SiPBA.

## 1.2 Related work

**Optimistic Bilevel Optimization**: OBO has been extensively studied, with surveys detailing its theory, algorithms, and applications [19, 22, 23]. A common approach for solving OBO is to reduce it to a single-level problem, using Karush-Kuhn-Tucker (KKT) conditions, leading to Mathematical Programs with Complementarity Constraints (MPCC) [4, 51], or through value-function-based inequality constraints [70, 57]. Approximating the lower-level solution with a finite trajectory is another strategy [53, 26]. These approaches have yielded scalable and efficient algorithms suitable for large-scale machine learning tasks [58, 27, 48, 46, 44, 62, 33, 7, 49, 35, 61, 38, 42, 50, 39]. However, the max-structure at the upper level of PBO creates a more complex, three-level-like structure (min-max-min), hindering the direct application of OBO algorithms to PBO.

**Pessimistic Bilevel Optimization:** PBO has been surveyed in [43, 23]. Theoretical studies include [1], which investigates sufficient conditions for the existence of optimal solutions, and [47], which studies properties of approximate solutions. Optimality conditions for PBO have been explored, including KKT-type conditions for smooth and non-smooth cases [20, 21]. [8] studies the relationship between PBO and its MPCC reformulation. For PBO algorithms, [2, 73] propose penalty methods for solving weak linear PBO problems. [67] introduces a semi-infinite programming reformulation of PBO. [40] reformulates PBO as an OBO problem with a two-follower Nash game, solving it as an MPCC. [71] transforms PBO into a minimax problem with coupled constraints, proposing methods for the linear case. More recently, [11, 12] explores relaxation methods for solving PBO's KKT conditions. [31] combines the lower-level value function with the KKT conditions of the upper-level max problem, resulting in a constrained minimization problem solved by a gradient-based method. Several heuristic algorithms have also been proposed, though without convergence guarantees [5, 3]. Recently, gradient-based algorithms for minimax bilevel optimization have been developed [30, 34, 68]. However, these problems differ from PBO in that their max structure is on the upper-level variable, not the lower-level variable, making these algorithms inapplicable to PBO. To our knowledge, fully first-order, single-loop gradient-based algorithms for solving PBO remain limited.

## 2 Smooth approximation of PBO

Throughout this paper, we make the following standing assumptions:

**Assumption 1** *The upper-level objective function $F(x,y)$ is continuously differentiable, and its gradient $\nabla F(x,y)$ is Lipschitz continuous on $X \times Y$. For any fixed $x \in X$, $F(x,y)$ is $\mu$-strongly concave with respect to $y$ on $Y$ for some $\mu > 0$.*

**Assumption 2** *The lower-level objective function $f(x, y)$ is continuously differentiable, and and its gradient $\nabla f(x, y)$ is Lipschitz continuous on $X \times Y$. For any fixed $x \in X$, $f(x, y)$ is convex with respect to $y$ on $Y$. Furthermore, $\mathcal{S}(x)$ is nonempty for any $x \in X$. For any bounded set $B \subseteq X$, there exists a bounded set $D$ such that $\mathcal{S}(x) \cap D \neq \varnothing$ for every $x \in B$.*

In this section, we introduce a smooth approximation for $\phi(x)$, leading to a smooth approximation of the PBO problem. All proofs for the results presented in this section are provided in Appendix B.

## 2.1 Smooth approximation of $\phi(x)$

To construct a smooth approximation of $\phi(x)$, we first consider an equivalent reformulation of $\phi(x)$ as the value function of a constrained minimax problem:

$$\phi(x) = \min_{z \in Y} \max_{y \in Y} \; \{F(x, y) \quad \text{s.t.} \; f(x, y) \leq f(x, z)\}. \tag{2}$$

This reformulation was explored in [71] as an application of the value function approach for designing numerical methods for PBO problem. The equality in (2) is justified by [71, Lemmas 1, 2]; for completeness, a proof is provided in Appendix B.1.

Next, we use this constrained minimax formulation to develop a smooth approximation of $\phi(x)$. To address the nonsmoothness introduced by the constraint in (2), we consider a penalized approximation:

$$\min_{z \in Y} \max_{y \in Y} \; F(x, y) - \rho(f(x, y) - f(x, z)),$$

where $\rho > 0$ is a penalty parameter. Under the stated assumptions, this minimax problem is convex in $z$ and concave in $y$, making it computationally tractable. However, the potential non-uniqueness of the optimal $z$, can result in the value function of this penalized problem being nonsmooth with respect to $x$. To ensure smoothness and well-posedness, we introduce a regularization term for $z$ and a coupling term $\langle y, z \rangle$, leading to the following regularized objective function:

$$\psi_{\rho,\sigma}(x, y, z) := F(x, y) - \rho(f(x, y) - f(x, z)) + \frac{\sigma}{2}\|z\|^2 - \sigma\langle y, z \rangle, \tag{3}$$

where $\sigma > 0$ is a regularization parameter. This function $\psi_{\rho,\sigma}(x, y, z)$ is designed to be strongly convex in $z$ and strongly concave in $y$. Based on this, we propose the approximation for $\phi(x)$ as:

$$\phi_{\rho,\sigma}(x) := \min_{z \in Y} \max_{y \in Y} \; \psi_{\rho,\sigma}(x, y, z). \tag{4}$$

The strong convexity-concavity of $\psi_{\rho,\sigma}$ ensures that $\phi_{\rho,\sigma}(x)$ is well defined for any $x \in X$. Furthermore, it guarantees the existence and uniqueness of a saddle point, denoted by $(y_{\rho,\sigma}^*(x), z_{\rho,\sigma}^*(x))$, and allows the interchange of minimization and maximization operators, i.e., $\phi_{\rho,\sigma}(x) = \min_{z \in Y} \max_{y \in Y} \; \psi_{\rho,\sigma}(x, y, z) = \max_{y \in Y} \min_{z \in Y} \psi_{\rho,\sigma}(x, y, z)$.

It is important to highlight the role of the coupling term $\langle y, z \rangle$ in (3), introduced alongside the regularization term $\frac{\sigma}{2}\|z\|^2$. This coupling term is crucial for establishing Lemma 2.2, which is the foundation of the asymptotic convergence of the proposed approximation $\phi_{\rho,\sigma}(x)$ to $\phi(x)$, and of the saddle point $(y_{\rho,\sigma}^*(x), z_{\rho,\sigma}^*(x))$ as established in Theorems 2.5 and 2.6, respectively.

We now establish a key smoothness property of $\phi_{\rho,\sigma}(x)$: its differentiability, and provide an explicit formula for its gradient.

**Theorem 2.1** *Let $\rho, \sigma > 0$ be given constants. Then, for any $x \in X$, $\phi_{\rho,\sigma}(x)$ is differentiable. Its gradient is given by:*

$$\nabla\phi_{\rho,\sigma}(x) = \nabla_x F(x, y_{\rho,\sigma}^*(x)) - \rho\nabla_x f(x, y_{\rho,\sigma}^*(x)) + \rho\nabla_x f(x, z_{\rho,\sigma}^*(x)), \tag{5}$$

*where $(y_{\rho,\sigma}^*(x), z_{\rho,\sigma}^*(x))$ is the unique saddle point for the minimax problem defining $\phi_{\rho,\sigma}(x)$ in (4).*

## 2.2 Asymptotic convergence of the approximation

Using the smooth approximation function $\phi_{\rho,\sigma}(x)$, we formulate the corresponding smoothed optimization problem intended to approximate the original PBO (1):

$$\min_{x \in X} \; \phi_{\rho,\sigma}(x). \tag{6}$$

This subsection validates the use of (6) by establishing the asymptotic convergence properties of $\phi_{\rho,\sigma}(x)$ to $\phi(x)$, and, consequently, the convergence of the solutions of (6) to those of (1) as $\rho \to \infty$ and $\sigma \to 0$. We begin by establishing a relationship between $\phi_{\rho,\sigma}(x)$ and $\phi(x)$ in the limit.

**Lemma 2.2** *Let $\{\rho_k\}$ and $\{\sigma_k\}$ be sequences such that $\rho_k \to \infty$ and $\sigma_k \to 0$. Then, for any $x \in X$, it holds that:*

$$\limsup_{k\to\infty} \phi_{\rho_k,\sigma_k}(x) \leq \phi(x). \tag{7}$$

*Furthermore, considering the optimal values, we have:*

$$\limsup_{k\to\infty} \left( \inf_{x\in X} \phi_{\rho_k,\sigma_k}(x) \right) \leq \inf_{x\in X} \phi(x). \tag{8}$$

Lemma 2.2 provides an upper bound on the limit of the approximate values. Building upon this, we can demonstrate the convergence of the optimal values under mild conditions.

**Proposition 2.3** *Let $\{\rho_k\}$ and $\{\sigma_k\}$ be sequences such that $\rho_k \to \infty$ and $\sigma_k \to 0$ as $k \to \infty$. If either $X$ or $Y$ is bounded, then the optimal values converge:*

$$\lim_{k\to\infty} \left( \inf_{x\in X} \phi_{\rho_k,\sigma_k}(x) \right) = \inf_{x\in X} \phi(x)$$

Establishing the convergence of optimal solutions (minimizers) requires additional structure related to the continuity properties of $\phi(x)$. To this end, we introduce the assumption of lower semi-continuity.

**Assumption 3** *$\phi(x)$ is lower semi-continuous (l.s.c.) on $X$. That is, for any sequence $\{x_k\} \subset X$ such that $x_k \to \bar{x} \in X$ as $k \to \infty$, it holds that, $\phi(\bar{x}) \leq \liminf_{k\to\infty} \phi(x_k)$.*

Lower semi-continuity is equivalent to the closedness of the function's epigraph and its level sets, and it guarantees the existence of a minimizer for $\phi(x)$ over a compact set $X$ (see, e.g., [60, Theorem 1.9]). Sufficient conditions for Assumption 3, such as the inner semi-continuity of the lower-level solution map $\mathcal{S}(x)$, are discussed in Appendix B.5. Under this assumption, we can establish the following result for epi-convergence.

**Lemma 2.4** *Assume $\phi(x)$ is lower semi-continuous on $X$. Let $\{\rho_k\}$ and $\{\sigma_k\}$ be sequences such that $\rho_k \to \infty$ and $\sigma_k \to 0$ as $k \to \infty$. Then, for any sequence $\{x_k\} \subset X$ converging to $\bar{x}$, we have:*

$$\liminf_{k\to\infty} \phi_{\rho_k,\sigma_k}(x_k) \geq \phi(\bar{x}). \tag{9}$$

Conditions (7) (applied with a constant sequence $x_k = x$) and (9) together imply the epi-convergence of the sequence of functions $\{\phi_{\rho_k,\sigma_k}\}$ to $\phi$ on $X$ as $k \to \infty$ (see, e.g., [60, Proposition 7.2]). This signifies that the epigraph of $\phi_{\rho_k,\sigma_k}(x)$ converges, in the set-theoretic sense, to the epigraph of $\phi(x)$. Leveraging this epi-convergence property, and employing results such as [14, Proposition 4.6] or [60, Theorem 7.31], we can establish the subsequential convergence of minimizers of problem (6).

**Theorem 2.5** *Assume $\phi(x)$ is lower semi-continuous on $X$. Let $\{\rho_k\}$ and $\{\sigma_k\}$ be sequences such that $\rho_k \to \infty$ and $\sigma_k \to 0$ as $k \to \infty$. Let $x_k \in \operatorname{argmin}_{x\in X} \phi_{\rho_k,\sigma_k}(x)$. Then, any accumulation point $\bar{x}$ of the sequence $\{x_k\}$ is an optimal solution to the original PBO (1), i.e., $\bar{x} \in \operatorname{argmin}_{x\in X} \phi(x)$.*

In the following, we characterize the asymptotic behavior of the saddle point $(y_{\rho,\sigma}^*(x), z_{\rho,\sigma}^*(x))$ as $\rho \to \infty$ and $\sigma \to 0$, and show that both components converge to the solution of the maximization problem that defines the value function $\phi(x)$ in (1).

**Theorem 2.6** *Assume $\phi(x)$ is lower semi-continuous on $X$. Let $\{\rho_k\}$ and $\{\sigma_k\}$ be sequences such that $\rho_k \to \infty$ and $\sigma_k \to 0$ as $k \to \infty$ and let $\{x_k\}$ be a sequence such that $x_k \in X$ and $x_k \to \bar{x}$ as $k \to \infty$. Then, we have:*

$$\lim_{k\to\infty} y_{\rho_k,\sigma_k}^*(x_k) = \lim_{k\to\infty} z_{\rho_k,\sigma_k}^*(x_k) = y^*(\bar{x}), \tag{10}$$

*where $y^*(\bar{x}) := \arg\max_{y\in\mathcal{S}(\bar{x})} F(\bar{x}, y)$.*

# 3 Single-loop gradient-based algorithm

In this section, we introduce the Single-loop Pessimistic Bilevel Algorithm (SiPBA), a novel single-loop gradient-based method designed to solve the PBO problem (1). The foundation of our approach is the smooth approximation problem (6), $\min_{x \in X} \phi_{\rho,\sigma}(x)$, developed in the previous section.

Owing to the continuous differentiability of the function $\phi_{\rho,\sigma}(x)$, gradient-based methods can be employed for solving it. However, as established in Theorem 2.1, the computation of the gradient $\nabla \phi_{\rho,\sigma}(x)$ necessitates the saddle point solution, denoted $(y_{\rho,\sigma}^*(x), z_{\rho,\sigma}^*(x))$, of the minimax subproblem $\min_{z \in Y} \max_{y \in Y} \psi_{\rho,\sigma}(x,y,z)$. Although this minimax problem is strongly convex in $z$ and strongly concave in $y$, finding its exact saddle point solution can be computationally expensive.

To mitigate this challenge, we propose constructing an inexact gradient at each iteration $k$ for updating $x^k$. Specifically, iterates $(y^k, z^k)$ are introduced to approximate the the exact saddle point solution to the minimax subproblem. At iteration $k$, given parameters $\rho_k, \sigma_k > 0$ and the current iterate $x^k$, a single projected gradient ascent-descent step is applied to the minimax subproblem $\min_{z \in Y} \max_{y \in Y} \psi_{\rho_k,\sigma_k}(x^k, y, z)$ to update $(y^k, z^k)$. The update rules are:

$$y^{k+1} = \text{Proj}_Y \left( y^k + \beta_k d_y^k \right), \quad z^{k+1} = \text{Proj}_Y \left( z^k - \beta_k d_z^k \right),$$

where $\beta_k > 0$ is the step size, $\text{Proj}_Y$ represents the Euclidean projection onto to set $Y$, and the update directions $d_y^k$ and $d_z^k$ are defined as:

$$d_y^k = \nabla_y F(x^k, y^k) - \rho_k \nabla_y f(x^k, y^k) - \sigma_k z^k, \quad d_z^k = \rho_k \nabla_y f(x^k, z^k) + \sigma_k(z^k - y^k). \quad (11)$$

Subsequently, the newly updated iterates $(y^{k+1}, z^{k+1})$ are used in place of the exact saddle point solution $(y_{\rho_k,\sigma_k}^*(x^k), z_{\rho_k,\sigma_k}^*(x^k))$ within the formula for $\nabla \phi_{\rho_k,\sigma_k}(x^k)$ (given in (5)). This yields an inexact gradient, which serves as the update direction $d_x^k$ for the iterate $x^k$:

$$d_x^k = \nabla_x F(x^k, y^{k+1}) - \rho_k \left( \nabla_x f(x^k, y^{k+1}) - \nabla_x f(x^k, z^{k+1}) \right). \quad (12)$$

The iterate $x^k$ is then updated as:

$$x^{k+1} = \text{Proj}_X \left( x^k - \alpha_k d_x^k \right),$$

where $\alpha_k > 0$ is the step size.

Furthermore, the parameters $\rho_k$ and $\sigma_k$ are updated throughout the iterative process, specifically by ensuring $\rho_k \to \infty$ and $\sigma_k \to 0$ as $k \to \infty$. The precise update strategies for selecting these parameters, along with the step sizes $\alpha_k$ and $\beta_k$, are detailed in Theorem 4.2 presented in Section 4.

Based on the preceding components, we now formally present the Single-loop Pessimistic Bilevel Algorithm (SiPBA) for solving the PBO problem (1) in Algorithm 1. In many practical applications where projections onto $X$ and $Y$ are computationally efficient, SiPBA offers the significant advantage of a single-loop structure, making it straightforward to implement.

---

**Algorithm 1: Si**ngle-loop **P**essimistic **B**ilevel **A**lgorithm (**SiPBA**)

---

**Input:** *Initial points* $(x^0, y^0, z^0) \in X \times Y \times Y$, *stepsizes* $\alpha_k, \beta_k > 0$, *parameters* $\rho_k, \sigma_k > 0$
**for** $k = 0, 1, \ldots, K-1$ **do**

    calculate $d_y^k$ and $d_z^k$ as in (11) and update

$$y^{k+1} = \text{Proj}_Y \left( y^k + \beta_k d_y^k \right), \quad z^{k+1} = \text{Proj}_Y \left( z^k - \beta_k d_z^k \right);$$

    calculate $d_x^k$ as in (12) and update

$$x^{k+1} = \text{Proj}_X \left( x^k - \alpha_k d_x^k \right).$$

---

# 4 Convergence analysis

This section establishes the convergence properties of the proposed SiPBA. All proofs for the results presented herein are provided in Appendix C.

Throughout this section, we introduce an additional assumption regarding the boundedness of $X$.

**Assumption 4** *The set $X$ is compact.*

To streamline the notation in this section, given the sequences $\rho_k$ and $\sigma_k$, we adopt the following shorthand: $\phi_k(x)$, $\psi_k(x, y, z)$, $y_k^*(x)$ and $z_k^*(x)$ will denote $\phi_{\rho_k,\sigma_k}(x)$, $\psi_{\rho_k,\sigma_k}(x, y, z)$, $y_{\rho_k,\sigma_k}^*(x)$ and $z_{\rho_k,\sigma_k}^*(x)$, respectively. Furthermore, let $u := (y, z)$, $u^k := (y^k, z^k)$ and $u_k^*(x) = (y_k^*(x), z_k^*(x))$. Let $L_F$ and $L_f$ denote the Lipschitz constants of $\nabla F(x, y)$ and $\nabla f(x, y)$ on $X \times Y$, respectively.

To facilitate the convergence analysis of SiPBA, we introduce a merit function $V_k$ incorporating dynamic positive coefficients $a_k > 0$ and $b_k > 0$:

$$V_k = a_k(\phi_k(x^k) - \underline{\phi}) + b_k\|u^k - u_k^*(x^k)\|^2, \tag{13}$$

where $\underline{\phi}$ represents a uniform lower bound for $\phi_k(x^k)$, such that $\phi_k(x^k) \geq \underline{\phi}$ for all $k$. The existence of such a lower bound is formally established in Lemma C.8 in the Appendix, under the condition that $\phi(x)$ is bounded below on $X$. Consequently, as $a_k > 0$, $b_k > 0$, $\phi_k(x^k) \geq \underline{\phi}$, and the squared norm term is inherently nonnegative, $V_k$ is always nonnegative.

Through a careful selection of the parameters $\rho_k, \sigma_k$, step sizes $\alpha_k, \beta_k$, and merit function coefficients $a_k, b_k$, we establish the following descent property for the merit function $V_k$.

**Proposition 4.1** *Let $\{(x^k, y^k, z^k)\}$ be the sequence generated by SiPBA(Algorithm 1) with parameters selected as:*

$$\alpha_k = \alpha_0 k^{-s}, \quad \beta_k = \beta_0 k^{-2p-q}, \quad \sigma_k = \sigma_0 k^{-q}, \quad \rho_k = \rho_0 k^p, \tag{14}$$

*with $\alpha_0, \beta_0, \sigma_0, \rho_0, s, p, q > 0$. Assume that $s > t + 4p + 2q$, $t > 4p + 4q$ and $p, q < 1$. If $\beta_0/\sigma_0$ is sufficiently small, then for all sufficiently large $k$, the following inequality holds:*

$$V_{k+1} - V_k \leq -\frac{a_k}{4\alpha_k}\|x^{k+1} - x^k\|^2 - \frac{1}{4}b_k\beta_k\bar{\sigma}_k\|u^k - u_k^*(x^k)\|^2 + \zeta_k, \tag{15}$$

*where $V_k$ is defined in (13) with $a_k = k^{-s}$, $b_k = k^{-t}$, $\bar{\sigma}_k = \min\{\sigma_k, \mu\}$, and $\{\zeta_k\}$ is a summable sequence, i.e., $\sum_{k=0}^{\infty} \zeta_k < \infty$.*

Using this descent property of $V_k$, we establish the following convergence result and derive non-asymptotic convergence rates for the error terms $\|x^{k+1} - x^k\|/\alpha_k$ and $\|u^k - u_k^*(x^k)\|$.

**Theorem 4.2** *Let $\{(x^k, y^k, z^k)\}$ be the sequence generated by SiPBA(Algorithm 1) with parameters selected as in (14). Suppose that the function $\phi(x)$ is bounded below on the set $X$. Assume further that $0 < s < 1/2$, $0 < p, q < 1$ and $8p + 8q \leq s$. If $\beta_0/\sigma_0$ is sufficiently small, then the following hold:*

$$\min_{0 < k < K} \frac{1}{\alpha_k^2}\|x^{k+1} - x^k\|^2 = \mathcal{O}(1/K^{1-2s}), \quad \text{and} \quad \min_{0 < k < K}\|u^k - u_k^*(x^k)\|^2 = \mathcal{O}(1/K^{1-6p-7q}).$$

*Moreover,*

$$\liminf_{k \to \infty} \|x^k - \text{Proj}_X\left(x^k - \alpha_k\nabla\phi_k(x^k)\right)\|/\alpha_k = 0, \quad \text{and} \quad \liminf_{k \to \infty}\|u^k - u_k^*(x^k)\| = 0.$$

Based on Theorem 4.2, a practical parameter selection strategy for SiPBA is provided in Appendix A.4. Furthermore, we can establish a modified stationarity result for the iterates $x^k$ generated by SiPBA in terms of the $\epsilon$-subdifferential (cf. [55, Theorem 1.26]).

**Corollary 4.3** *Assume $\phi(x)$ is lower semi-continuous on $X$. Let $\{(x^k, y^k, z^k)\}$ be the sequence generated by SiPBA(Algorithm 1) with parameters chosen as specified in Theorem 4.2. Suppose that the function $\phi(x)$ is bounded below on the set $X$. If $\beta_0/\sigma_0$ is sufficiently small, then there exists a subsequence $\{x^{k_j}\}$ such that for any $\epsilon > 0$ and $\tilde{\epsilon} > 0$, there exists an integer $K > 0$ such that for all $k_j > K$, there exists a corresponding $\delta_j > 0$ for which the following inequality holds:*

$$\phi(x) + \epsilon\|x - x^{k_j}\| \geq \phi(x^{k_j}) - \tilde{\epsilon}, \qquad \forall x \in \mathbb{B}_{\delta_j}(x^{k_j}) \cap X.$$

## 5  Numerical experiments

To evaluate the performance of SiPBA, we conducted comprehensive validation through both synthetic examples and real-world applications. All computational experiments were performed on a server provisioned with dual Intel Xeon Gold 5218R CPUs (a total of 40 cores/80 threads, with 2.1-4.0 GHz) and an NVIDIA H100 GPU. Detailed information regarding the specific implementation of algorithms, along with the configurations for each experimental setup, is available in Appendix A.

## 5.1 Synthetic example

To empirically demonstrate the performance of SiPBA, we consider the following synthetic PBO:

$$\min_{x\in[0.1,10]^n} \max_{y\in\mathbb{R}^n} \frac{1}{n}\|x-\mathbf{e}\|^2 - \|y-\mathbf{e}\|^2, \quad \text{s.t. } y \in \arg\min_{y'\in[\frac{1}{2\sqrt{n}},\infty)^n} \left\|\langle\mathbf{e},y'\rangle - \|x\|\right\|^2. \quad (16)$$

where $\mathbf{e}$ denotes the all-ones vector of appropriate dimension. For $n \geq 2$, it can be shown that the unique optimal solution is given by $(x^*, y^*) = (\mathbf{e}/2, \mathbf{e}/(2\sqrt{n}))$. The performance is assessed by the relative error, $\epsilon_{rel} = (\|x^k - x^*\|^2 + \|y^k - y^*\|^2)/(\|x^0 - x^*\|^2 + \|y^0 - y^*\|^2)$. For all experiments in this subsection, the results are averaged over 10 independent runs, each initialized from a distinct, randomly generated starting point. The initial point $(x^0, y^0)$ is generated by sampling each component of $x^0$ from a uniform distribution $\mathcal{U}[0.1, 10]$ and each component of $y^0$ from $\mathcal{U}[1/(2\sqrt{n}), 10]$.

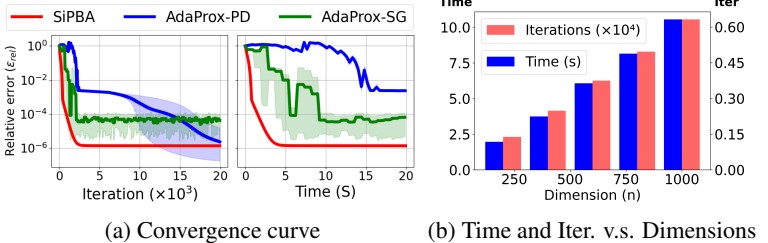

(a) Convergence curve      (b) Time and Iter. v.s. Dimensions

Figure 1: **(a)**: Convergence curves of SiPBA, AdaProx-PD and AdaProx-SG on (16) with $n = 100$; **(b)**: Iterations and runtime required for SiPBA on (16) for varying problem dimensions $n$.

Table 1: Performance comparison of the SiPBA, AdaProx-PD, AdaProx-SG, Scholtes-C, and Scholtes-D with $n = 100$.

|  | SiPBA | AdaProx-PD | AdaProx-SG | Scholtes-C | Scholtes-D |
|---|---|---|---|---|---|
| Min. ($\epsilon_{rel}$) | $1.22 \times 10^{-6}$ | $1.79 \times 10^{-7}$ | $3.80 \times 10^{-6}$ | $1.12 \times 10^{-5}$ | $9.60 \times 10^{-5}$ |
| Max. ($\epsilon_{rel}$) | $1.45 \times 10^{-6}$ | $1.53 \times 10^{-5}$ | $1.02 \times 10^{-4}$ | 0.10 | 1.06 |
| Valid Runs | 10/10 | 10/10 | 9/10 | 1/10 | 1/10 |
| Ave Time (s) | 1.03 | 87.44 | 4.07 | 23.12 | 23.81 |

We compare SiPBA against two other gradient-based methods—AdaProx-PD and AdaProx-SG [31]—as well as two MPCC-based approaches— Compact Scholtes (Scholtes-C) and Detailed Scholtes (Scholtes-D) relaxation method [12]. SiPBA, AdaProx-PD, and AdaProx-SG are run for 20,000 iterations, while Scholtes-C and Scholtes-D are run for 10 outer iterations (as they converge within this range). We report the minimum and maximum relative errors, the number of successful runs achieving the tolerance $\epsilon_{rel} < 10^{-4}$ (Valid Runs), and the average runtime to reach this tolerance for those valid runs (Ave. Time). Figure 1(a) shows the convergence curve of the gradient-based algorithms and Table 1 summarizes the final performance metrics of all the methods. We further evaluate SiPBA's robustness to hyperparameters (stepsizes $\alpha_0, \beta_0$ and update factors $p, q, s$) and its scalability by measuring runtime and iterations required to achieve the tolerance, $\epsilon_{rel} < 10^{-4}$, across varying hyperparameters and problem dimensions, with results shown in Table 2 and Figure 1 (b). All the results indicate the consistent performance and computational efficiency of SiPBA.

Table 2: Ablation analysis for SiPBA on (16) with $n = 100$.

| $\alpha_0$ | $\beta_0$ | $p$ | $q$ | $s$ | Time (s) | $\alpha_0$ | $\beta_0$ | $p$ | $q$ | $s$ | Time (s) |
|---|---|---|---|---|---|---|---|---|---|---|---|
| 0.1 | 0.001 | 0.001 | 0.001 | 0.1 | 1.0±0.1 | 0.1 | 0.001 | **0.0001** | 0.001 | 0.1 | 1.0±0.1 |
| **1** | 0.001 | 0.001 | 0.001 | 0.1 | 0.1±0.0 | 0.1 | 0.001 | 0.001 | **0.01** | 0.1 | 1.2±0.1 |
| **0.01** | 0.001 | 0.001 | 0.001 | 0.1 | 14.5±1.5 | 0.1 | 0.001 | 0.001 | **0.0001** | 0.1 | 1.1±0.2 |
| 0.1 | **0.01** | 0.001 | 0.001 | 0.1 | 0.5±0.1 | 0.1 | 0.001 | 0.001 | 0.001 | **0.3** | 5.0±0.2 |
| 0.1 | **0.0001** | 0.001 | 0.001 | 0.1 | 16.4±3.1 | 0.1 | 0.001 | 0.001 | 0.001 | **0.016** | 0.8±0.1 |
| 0.1 | 0.001 | **0.01** | 0.001 | 0.1 | 1.4±0.3 | 0.1 | 0.001 | **0.01** | **0.01** | **0.16** | 1.9±0.3 |

## 5.2 Spam classification

Spam classification is challenging due to adversarial dynamics and poor cross-domain generalization. We consider the PBO model for spam classification tasks, as proposed by [16]:

$$\min_{w\in\mathbb{R}^n} \max_{\hat{x}} \; l(w, \hat{x}, y) + \lambda_1\text{Reg}(w) \quad \text{s.t. } \hat{x} \in \arg\min_{x'\in\mathcal{X}} l'(w, x') + \lambda_2\|\varphi(x') - \varphi(x)\|^2, \quad (17)$$

where $w$ denotes the classifier parameters, $(x, y)$ represents vectorized training data, $l$ (resp. $l'$) corresponds to the classifier (resp. adversarial generator) loss, $\text{Reg}(\cdot)$ denotes the regularization term, and $\varphi(\cdot)$ characterizes the feature of data.

We conduct a two-part empirical comparison. First, we compare the PBO model (17) trained using SiPBA (with $\varphi(x)$ as the top $k$ principal components) to the same model trained using the SQP method with $\varphi(x) = x$, as proposed in [16]. Second, we compare the SiPBA-trained PBO model against a standard single-level model, $\min_w l(w, x, y) + \lambda_1 \text{Reg}(w)$, trained using the scikit-learn library [59]. We use either hinge loss or cross-entropy for both $l$ and $l'$, and refer to the resulting methods as SiPBA-Hinge/CE, SQP-Hinge/CE, and Single-Hinge/CE.

Experiments are conducted using four standard spam datasets: TREC2006 [56], TREC2007 [52], EnronSpam [54], and LingSpam [6]. The average results over ten independent runs are summarized in Table 3, which indicate that the PBO model (either trained with SQP or SiPBA) exhibits superior cross-domain performance compared to the single-level models. Moreover, the SiPBA-trained models achieve the best overall accuracy and F1 score.

Table 3: Accuracy (Acc) and F1 score (F1) on four spam corpora, training on TREC06, TREC07, EnronSpam or LingSpam.

| Train Set | Model | Test Set(Acc/F1) | | | | Ave(Acc/F1) |
|---|---|---|---|---|---|---|
| | | TREC06 | TREC07 | EnronSpam | LingSpam | |
| TREC06 | SiPBA-Hinge | 96.4/94.7 | 87.3/81.0 | 70.6/70.2 | 87.6/92.7 | **85.5/84.7** |
| | SiPBA-CE | 94.5/92.5 | 79.5/73.0 | 70.9/71.8 | 87.6/92.8 | **83.1/82.5** |
| | SQP-Hinge | 93.1/90.0 | 89.2/83.2 | 69.0/66.7 | 89.0/93.4 | 85.1/83.3 |
| | SQP-CE | 93.6/91.3 | 78.9/72.4 | 70.7/71.4 | 87.2/92.6 | 82.6/81.9 |
| | Single-Hinge | 95.4/93.1 | 89.3/82.8 | 63.9/46.5 | 75.5/82.5 | 81.0/76.2 |
| | Single-CE | 93.8/90.4 | 88.5/79.6 | 56.9/24.1 | 55.1/62.6 | 73.6/64.2 |
| TREC07 | SiPBA-Hinge | 68.9/16.8 | 93.7/89.7 | 57.0/33.7 | 50.5/57.6 | 67.5/49.5 |
| | SiPBA-CE | 71.7/56.9 | 98.1/97.2 | 68.3/68.8 | 64.6/75.5 | **75.7/74.6** |
| | SQP-Hinge | 68.9/17.2 | 95.3/92.5 | 55.0/21.0 | 29.9/28.1 | 62.3/39.7 |
| | SQP-CE | 71.3/56.9 | 97.7/96.6 | 68.4/69.7 | 70.1/80.5 | 76.9/75.9 |
| | Single-Hinge | 65.4/1.9 | 97.7/96.4 | 50.9/0.2 | 16.6/0.3 | 57.7/24.7 |
| | Single-CE | 66.4/3.4 | 95.7/93.0 | 51.0/0.8 | 17.3/1.8 | 57.6/24.8 |
| EnronSpam | SiPBA-Hinge | 75.8/61.8 | 72.1/28.0 | 95.9/95.8 | 59.6/67.4 | **75.9/63.3** |
| | SiPBA-CE | 76.3/62.8 | 74.0/34.4 | 95.2/95.0 | 64.0/72.0 | **77.4/66.1** |
| | SQP-Hinge | 77.5/61.7 | 70.5/22.8 | 96.1/96.0 | 52.3/59.3 | 74.1/60.0 |
| | SQP-CE | 76.0/62.6 | 73.4/32.9 | 94.9/94.8 | 63.0/71.0 | 76.8/65.3 |
| | Single-Hinge | 76.8/56.0 | 69.3/15.0 | 95.8/95.6 | 47.2/52.3 | 72.3/54.7 |
| | Single-CE | 76.4/55.4 | 70.0/19.2 | 95.6/95.3 | 43.1/46.9 | 71.3/54.2 |
| LingSpam | SiPBA-Hinge | 63.4/59.1 | 66.2/51.2 | 71.1/65.4 | 99.4/99.6 | **75.0/68.8** |
| | SiPBA-CE | 71.8/48.5 | 69.0/27.6 | 59.1/34.3 | 91.8/94.8 | **72.9/51.3** |
| | SQP-Hinge | 42.5/53.8 | 45.3/52.0 | 72.5/65.8 | 98.2/99.0 | 64.6/67.7 |
| | SQP-CE | 72.0/49.5 | 68.9/26.2 | 58.9/33.9 | 91.9/94.8 | 72.9/51.1 |
| | Single-Hinge | 37.2/51.9 | 38.6/50.6 | 56.7/69.0 | 95.7/97.5 | 57.1/67.3 |
| | Single-CE | 34.5/51.0 | 34.0/50.1 | 51.3/66.8 | 91.4/95.1 | 52.8/65.8 |

## 5.3 Hyper-representation

Hyper-representation[29, 27] aim to learn an effective representation of the input data for lower-level classifiers, where PBO model was used to handle the potential multiplicity of optimal solutions in the lower-level problem and robust learn the representation [31]. In this experiment, we further explore the potential of the PBO model and compare it with optimistic models.

### 5.3.1 Linear hyper-representation on synthetic data

We begin with a synthetic linear hyper-representation task, which is formulated as:

$$\min_{H \in \mathbb{R}^{n \times p}} \max_{w} \frac{1}{m_1} \|\mathbf{X}_{val}^T H w - \mathbf{y}_{val}\|^2, \quad \text{s.t. } w \in \arg\min_{w' \in \mathbb{R}^p} \frac{1}{m_2} \|\mathbf{X}_{train}^T H w' - \mathbf{y}_{train}\|^2, \quad (18)$$

where $\mathbf{X}_{val} \in \mathbb{R}^{n \times m_1}$ and $\mathbf{X}_{train} \in \mathbb{R}^{n \times m_2}$ are the validation and training feature matrices, and $\mathbf{y}_{val} \in \mathbb{R}^{m_1}, \mathbf{y}_{train} \in \mathbb{R}^{m_2}$ are the corresponding response vectors. The synthetic data is generated as in [29], with feature matrices $\mathbf{X}_{val}, \mathbf{X}_{train}, \mathbf{X}_{test}$ and ground-truth matrices $H_{real}$ and vectors $w_{real}$ sampled randomly. The response vectors are formed using the linear model $\mathbf{y}_{(\cdot)} = \mathbf{X}_{(\cdot)}^\top H_{real} w_{real}$, with Gaussian noise $\epsilon \sim \mathcal{N}(0, a^2)$ added to both $\mathbf{X}_{(\cdot)}$ and $\mathbf{y}_{(\cdot)}$ for train and valid data to simulate noise.

To evaluate solver efficiency and formulation effectiveness, we conduct a two-part comparison. First, we compare the SiPBA algorithm with PBO algorithms AdaProx-PD and AdaProx-SG. Second, we assess the impact of the bilevel formulation by comparing the pessimistic model (18) (solved by SiPBA) with its optimistic variant (solved by AID-FP , AID-CG [29] and PZOBO [62]), which replaces $\max_w$ with $\min_w$ in the upper level. To assess the robustness of each method under varying levels of noise, we conduct experiments with moderate ($a = 0.1$) and severe ($a = 1$) perturbations. The performance is measured by test loss, averaged over 10 random seeds. Results in Figure 2 demonstrate the stability and efficiency of SiPBA.

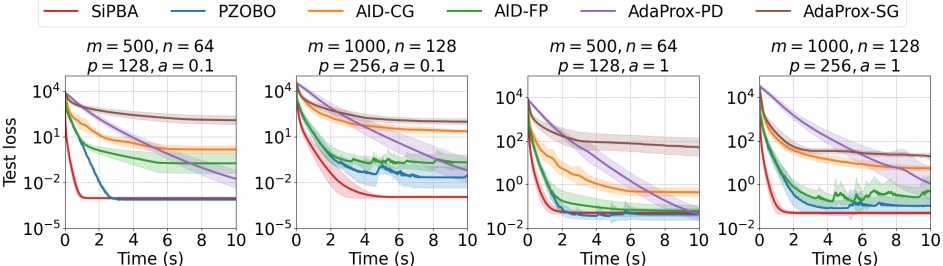

Figure 2: Test loss v.s. time in Hyper-representation with varying dimensions and noise levels.

### 5.3.2 Deep hyper-representation on MNIST and FashionMNIST

To further assess the practical effectiveness of the pessimistic model, we conduct a more complicated deep hyper-representation experiment on real-world classification tasks. The problem is formulated as:

$$\min_{\theta \in \Theta} \max_w \frac{1}{m_1} \| f(\mathbf{X}_{val}, \theta)w - \mathbf{y}_{val} \|^2, \text{ s.t. } w \in \arg\min_{w' \in \mathcal{W}} \frac{1}{m_2} \| f(\mathbf{X}_{train}, \theta)w' - \mathbf{y}_{train} \|^2, \quad (19)$$

where $f(\cdot, \theta)$ represents a neural network parameterized by $\theta$, and $w$ corresponds to a linear layer.

We adopt the LeNet-5 architecture [62] as the feature extractor $f(\cdot, \theta)$ and evaluate performance on the MNIST and FashionMNIST datasets. Each dataset is randomly split into 50,000 training samples, 10,000 validation samples, and 10,000 test samples, with performance evaluated by test accuracy. We compare the pessimistic formulation (19), trained with SiPBA, to its optimistic variant (replacing $\max_w$ with $\min_w$ in the upper level) trained with AID-FP, AID-CG [29], and PZOBO [62]. Mean results over ten runs are shown in Figure 3, which shows that SiPBA achieves the highest test accuracy.

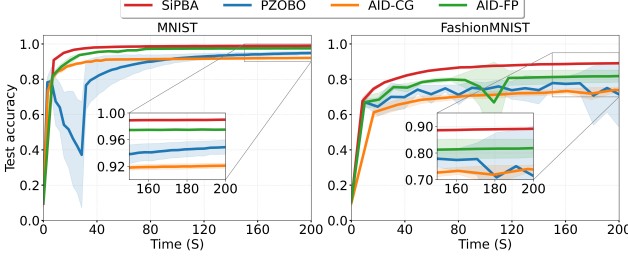

Figure 3: Hyper-representation on MNIST and FashionMNIST.

## 6   Conclusions and future work

This paper introduces a novel smooth approximation for PBO, which underpins the development of SiPBA, an efficient new gradient-based PBO algorithm. SiPBA avoids computationally expensive second-order derivatives and the need for iterative inner-loop procedures to solve subproblems.

The current study is confined to deterministic PBO problems. However, a significant number of practical applications feature PBO problems within stochastic settings. Extending the SiPBA methodology to effectively address these stochastic PBO problems presents a crucial and promising direction for future research. We hope this research stimulates further algorithmic development for stochastic PBO.

## Acknowledgements

Authors listed in alphabetical order. This work was supported by National Key R&D Program of China (2023YFA1011400), National Natural Science Foundation of China (12222106, 12326605), Guangdong Basic and Applied Basic Research Foundation (No. 2022B1515020082), the Longhua District Science and Innovation Commission Project Grants of Shenzhen (No. 20250113G43468522) and Natural Science Foundation of Shenzhen (No. 20250530150024003).

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

# A Numerical experiment

In this section, we provide the specific description of experiments in Section 5. All experiments were conducted on CPUs except for the spam classification task, which utilized an NVIDIA H100 GPU. The primary compute node features dual Intel Xeon Gold 5218R processors operating at 2.1GHz base frequency (4.0GHz turbo boost), featuring 40 physical cores (80 logical threads) with a three-tier cache architecture: 1.3MB L1, 40MB L2, and 55MB L3 shared cache. The NUMA-based memory architecture partitions resources across two distinct domains, with hardware support for AVX-512 vector instructions and VT-x virtualization. Security mitigations against Spectre/Meltdown vulnerabilities were implemented through combined microcode patches and kernel-level protections.

## A.1 Synthetic example

For the problem 5.1, we can get the value function by simple calculation:

$$\phi_p(x) = \frac{1}{n}\|x - \mathbf{e}\|^2 - \|y^*(x) - \mathbf{e}\|^2, \text{ where } y^*(x) := \begin{cases} \frac{\|x\|\mathbf{e}}{n}, \|x\| > \frac{\sqrt{n}}{2}, \\ \frac{\mathbf{e}}{2\sqrt{n}}, \|x\| \le \frac{\sqrt{n}}{2}, \end{cases} \tag{20}$$

which implies that $(x^*, y^*) = \left(\frac{\mathbf{e}}{2}, \frac{\mathbf{e}}{2\sqrt{n}}\right)$. Except for the stability tests of the initial step sizes reported in Table 2, we fix the hyper-parameters as

$$p = 0.001, \quad q = 0.001, \quad s = 0.1, \quad \alpha_0 = 0.1, \quad \beta_0 = 0.001 \quad \rho_0 = 10, \quad \sigma_0 = 0.01. \tag{21}$$

For the implementation of AdaProx-PD, we first fix $\xi = 0.001$, $\sigma = 0.001$, $\gamma_t = t$, $\theta_t = \gamma_{t+1}/\gamma_t$, $2/(L_g + 2\alpha) = 1/\tau_t$ and set $\tau_t = t\tau_0$, $\eta_t = \eta_0/t$, $K = 100$, and $N = \min\{log(1/\epsilon), 200\}$, $T = \min\{1/\sqrt{\epsilon}, 200\}$. Then we perform a grid search for

$$1/\eta_0, 1/\tau_0, \xi \in \{0.1, 0.01, 0.001, 0.0001\}, \quad \sigma, \beta \in \{0.1, 0.01, 0.001\}.$$

However, none of these yielded satisfactory convergence. We thus fixed parameters across iterations, set $K = 100$, $T = 200$, $N = 10$, $\theta = 1$, $2/(L_g + 2\alpha) = 1/\tau$, and conducted a grid search to find a best parameter to get lowest loss, where the grid is set as follows:

$$1/\eta, 1/\tau, \xi \in \{0.1, 0.01, 0.001, 0.0001\}, \quad \sigma, \beta \in \{0.1, 0.01, 0.001\}.$$

As a result, we have $1/\tau = 0.001$, $1/\eta = 0.001$, $\sigma = 0.001$, $\xi = 0.1$ and $\beta = 0.001$ for AdaProx-PD.

For the implementation of AdaProx-SG, we fix $\gamma_t = t$, $\theta_t = \gamma_{t+1}/\gamma_t$, $2/(L_g + 2\alpha) = 1/\gamma_t$ and $K = 100$, and $N = \min\{log(1/\epsilon), 200\}$, $T = \min\{1/\epsilon, 200\}$. Then we perform a grid search for

$$1/\gamma_0, \xi \in \{0.1, 0.01, 0.001, 0.0001\}, \quad \sigma, \beta \in \{0.1, 0.01, 0.001\}.$$

As a result, we have $1/\gamma_0 = 0.1$, $\sigma = 0.001$, $\xi = 0.1$ and $\beta = 0.1$ for AdaProx-SG.

For the implementation of the Compact Scholtes and Detailed Scholtes relaxation method in [12], we utilize the fsolve solver from the SciPy library [65]. In each outer iteration, the value of $t_{k+1}$ is updated as $t_{k+1} = 0.1t_k$ with $t_0 = 1$ and fix $\epsilon = t_k$.

## A.2 Spam classification

Spam classification remains a critical challenge in machine learning due to adversarial dynamics: spammers adapt their strategies in response to deployed classifiers, while models trained on specific datasets often exhibit poor cross-domain generalization. In this paper, we extend the pessimistic bilevel model for Spam classification in [16]:

$$\min_{w \in \mathbb{R}^n} \max_{\hat{x}} \ l(w, \hat{x}, y) + \lambda_1 \text{Reg}(w) \qquad \text{s.t. } \hat{x} \in \argmin_{x' \in \mathcal{X}} l'(w, x') + \lambda_2 \|\varphi(x') - \varphi(x)\|^2, \tag{22}$$

where $w$ denotes the classifier parameters, $(x, y)$ represents vectorized training data, $l$ (resp. $l'$) corresponds to the classifier (resp. adversarial generator) loss, $\text{Reg}(\cdot)$ denotes the regularization term, and $\phi(\cdot)$ characterizes the feature of data. This framework explicitly models spammer adaptations through adversarial samples $\hat{x}$, enhancing classifier robustness against evolving threats.

We evaluate our model on four benchmark datasets:

- **TREC06** (37,822 emails; 24,912 spam / 12,910 ham): `https://plg.uwaterloo.ca/cgi-bin/cgiwrap/gvcormac/foo06`
- **TREC07** (75,419 emails; 50,199 spam / 25,220 ham): `https://plg.uwaterloo.ca/cgi-bin/cgiwrap/gvcormac/foo07`
- **EnronSpam** (33,715 emails; 16,545 spam / 17,170 ham): `https://www.cs.cmu.edu/~enron/`
- **LingSpam** (2,893 emails; 481 spam / 2,412 ham): `https://www.aueb.gr/users/ion/data/lingspam_public.tar.gz`

The text was vectorized using a TfidfVectorizer that removed English stop words, retained only terms appearing in at least five documents, and limited the feature space to the top 9000 most informative terms. We represent the resulting vectors as the variable $x$ and train the model in the vectorized space. To simulate the real world situation, we assume that email authors always aim to have their messages classified as ham; accordingly, we define $l'$ as the loss incurred when an email is classified as spam, using the same formulation (cross-entropy or hinge) as $l$. The specific definition of $l$ and $l'$ used in our experiment is given by

$$\text{PBO-Hinge}: \begin{cases} l(w,x,y) &= \frac{1}{n}\sum_{i=1}^{n} \max\{0, 1 - w^\top x_i y_i\}, \\ l'(w,x) &= \frac{1}{n}\sum_{i=1}^{n} \max\{0, 1 - w^\top x_i\}, \end{cases}$$

$$\text{PBO-CE}: \begin{cases} l(w,x,y) &= \text{CrossEntropy}(w^\top x, \frac{y+1}{2}), \\ l'(w,x) &= \text{CrossEntropy}(w^\top x, 1). \end{cases}$$

where $x_i$ denotes the input data, $y_i \in \{-1, 1\}$ denotes the label (-1 for spam and 1 for non-spam) and CrossEntropy is defined by $\text{CrossEntropy}(w^\top x, y) = -\left(y \log(\sigma(w^\top x)) + (1-y)\log(1 - \sigma(w^\top x))\right)$ and $\sigma(z) = 1/(1 + e^{-z})$ is the Sigmoid function. The function $\phi$ is defined as

$$\varphi(x) := xP_k,$$

where the matrix $P_k$ consists of the top $k$ principal components obtained from the principal component decomposition of the sample matrix. This choice is motivated by the assumption that meaningful information in emails is primarily captured by the principal components, and modifications made by spammers generally do not alter this core content. Therefore, we penalize changes along the principal components to enforce robustness against adversarial modifications. In this experiment, we always set $k = 100, \lambda_1 = 0.01, \lambda_2 = 0.1$ for SiPBA.

For the implementation of SiPBA, we fix $\rho_0 = 10$, $\sigma_0 = 10^{-6}$, $p = 0.01$, $q = 0.01$ and $s = 0.16$, and we set the hyperparameter as follows:

$$\text{TREC06}: \begin{cases} \alpha_0 = 0.03, \ \beta_0 = 10^{-6}, & \text{for PBO-Hinge,} \\ \alpha_0 = 0.1, \ \beta_0 = 10^{-4}, & \text{for PBO-CE,} \end{cases}$$

$$\text{TREC07}: \begin{cases} \alpha_0 = 0.1, \ \beta_0 = 10^{-2} & \text{for PBO-Hinge,} \\ \alpha_0 = 0.05, \ \beta_0 = 10^{-4}, & \text{for PBO-CE,} \end{cases}$$

$$\text{EnronSpam}: \begin{cases} \alpha_0 = 0.02, \ \beta_0 = 10^{-7}, & \text{for PBO-Hinge,} \\ \alpha_0 = 0.01, \ \beta_0 = 10^{-7}, & \text{for PBO-CE,} \end{cases}$$

$$\text{LingSpam}: \begin{cases} \alpha_0 = 0.02, \ \beta_0 = 5 \times 10^{-5} & \text{for PBO-Hinge,} \\ \alpha_0 = 0.05, \ \beta_0 = 10^{-7}, & \text{for PBO-CE.} \end{cases}$$

For the implementation of SQP-Hinge and SQP-CE, we set $\varphi(x) := x$ (to ensure the lower level can be uniquely solved) and $\lambda_1 = 0.01, \lambda_2 = 0.001$ and solve the problem using the trust-constr method from the scipy.optimize solver [65].

For the implementation of Single-Hinge and Single-CE, we use SVC and Logistic Regression from scikit-learn [59] with default setting and $max\_iter = 10000$.

## A.3 Hyper-representation

In the linear hyper-representation on synthetic data, we follow the data generation procedure of [62]. Specifically, we generate the ground-truth matrix $H_{real} \in \mathbb{R}^{p \times d}$, the vector $w_{real} \in \mathbb{R}^d$, and

the inputs $\mathbf{X}_{train}$, $\mathbf{X}_{val}$, $\mathbf{X}_{test}$ by sampling each entry independently from the standard normal distribution $\mathcal{N}(0, 1)$. We then generate the train, valid and test data by $\mathbf{y}_{(\cdot)} = \mathbf{X}_{(\cdot)}^\top H w$. Finally, we add $\epsilon \sim \mathcal{N}(0, a^2)$ with $a = 0.1$ and $a = 1$ to $\mathbf{X}_{val}, \mathbf{X}_{train}$ and $\mathbf{y}_{val}, \mathbf{y}_{train}$ to simulate the noise condition. The parameters of the algorithms are initialized as

- For SiPBA, we set $p = 0.01, q = 0.01, s = 0.16, \rho_0 = 10, \sigma_0 = 10^{-4}$. And the stepsize is set as $\alpha_0 = 5 \times 10^{-4}, \beta_0 = 5 \times 10^{-4}$ for $m = 500, a = 0.1$ and $\alpha_0 = 10^{-4}, \beta_0 = 10^{-4}$ for the remaining senarios.
- For AdaProx-PD, we set $K = 100, T = 20, N = 10, \theta = 1, 2/(L_g + 2\alpha) = 1/\tau$ and $\sigma = 0.1, \xi = 0.001$ and $\beta = 0.001$. And the stepsize is setted as $\tau = \eta = 10^4, 2 \times 10^4, 2 \times 10^4, 5 \times 10^4$ for the four senarios in Figure 2.
- For AdaProx-SG, we set $K = 100, T = \min\{20, 1/\epsilon\}, N = \min\{10, log(1/\epsilon)\}, 2/(L_g + 2\alpha) = 1/\gamma$, $\sigma = 0.001, \xi = 0.001, \beta = 0.001$ and $\gamma_0 = 10^4$.
- For AID-FP, AID-CG and PZOBO, we keep the setting as presented in `https://github.com/sowmaster/esjacobians/tree/master`, except that the inner learning rate is set as 0.0001 as we found it's more stable for these algorithms.

For the classification tasks on MNIST and FashionMNIST, we split the dataset into 50,000 training samples, 10,000 validation samples, and 10,000 test samples. Both the upper and lower levels are trained using the LeNet architecture, following the setting in [62]. During each training iteration, we randomly select 256 samples to compute the loss and gradients. The parameters of the algorithms are initialized as follows:

- For SiPBA, we set $p = 0.01, q = 0.01, s = 0.16, \rho_0 = 10, \alpha_0 = 0.01, \beta_0 = 0.01$ and $\sigma_0 = 0.1$.
- For PZOBO, we adopt the implementations from `https://github.com/sowmaster/esjacobians/tree/master` and set number of inner iterations $T = 30$ for training on FashionMNIST to ensure proper convergence.
- For AID-CG, we set the learning rate to $lr = 0.001(0.0005)$ and the number of inner iterations to $T = 10\ (50)$ for training on MNIST (FashionMNIST).
- For AID-FP, we set $lr = 0.001$ and $T = 20(30)$ for training on MNIST (FashionMNIST) .

### A.4 Parameter Selection

The implementation of SiPBA includes seven parameters, namely $\alpha_0, \beta_0, \sigma_0, \rho_0, p, q, s$. The parameters $s$, $p$, and $q$ collectively govern the fundamental trade-off between value function approximation accuracy and iterative step size selection. The parameter $p$ controls the growth rate of the penalty coefficient $\rho_k = \rho_0 k^p$, while $q$ determines the decay rate of the regularization coefficient $\sigma_k = \sigma_0 k^{-q}$. Larger values of $p$ and $q$ yield faster convergence of the approximate value function $\phi_{\rho_k, \sigma_k}(x)$ to the true objective. The parameter $s$ regulates the step size decay rate $\alpha_k = \alpha_0 k^{-s}$ for the primal iterates $x^k$.

The theoretical requirement $s \geq 8p + 8q$ reveals an essential trade-off: choosing larger values for $p$ and $q$ accelerates the value function approximation but necessitates a larger $s$, resulting in smaller step sizes $\alpha_k$ that slows down the convergence rate of $x^k$. Conversely, smaller $p$ and $q$ permit more aggressive step sizes through reduced $s$, but at the cost of slower convergence of the approximate objective $\phi_{\rho_k, \sigma_k}(x)$ to the true value function, potentially degrading overall algorithmic performance.

We provide practical guidelines for parameter selection here. Specifically, the update rules are given by:

$$\alpha_k = \alpha_0 k^{-8p-8q}, \quad \beta_k = \beta_0 k^{-2p-q}, \quad \rho_k = \rho_0 k^{-p}, \quad \sigma_k = \sigma_0 k^{-q},$$

with default settings $p = q = 0.01$ and $\rho_0 = 10$. Therefore, tuning is only required for the three scalar parameters: $\alpha_0$, $\beta_0$, and $\sigma_0$.

## B  Proofs for Section 2

This section provides the proofs for the theoretical results established in Section 2.

## B.1 Equivalent minimax reformulation of $\phi(x)$

**Lemma B.1** *Consider the function*

$$\phi(x) := \max_y \ \{F(x,y) \quad \text{s.t. } y \in \mathcal{S}(x)\} \,,$$

*where*

$$\mathcal{S}(x) := \operatorname{argmin}_{y' \in Y} f(x,y').$$

*Then, for any $x \in X$, we have the following equivalent minimax reformulation:*

$$\phi(x) = \min_{z \in Y} \max_{y \in Y} \ \{F(x,y) \quad \text{s.t. } f(x,y) \leq f(x,z)\} \,.$$

*Proof.* Let $x \in X$ be an arbitrary point. The assumptions that $\mathcal{S}(x)$ is nonempty, and $F(x,y)$ is $\mu$-strongly concave with respect to $y$, and using the fact that $\mathcal{S}(x)$ is closed, we conclude that there exists some $y^* \in \mathcal{S}(x)$ such that $\phi(x) = F(x,y^*)$.

For any $z \in Y$, since $y^* \in \mathcal{S}(x)$, it follows that

$$f(x,y^*) \leq f(x,z).$$

Therefore, we have

$$\phi(x) = F(x,y^*) \leq \max_{y \in Y} \ \{F(x,y) \quad \text{s.t. } f(x,y) \leq f(x,z)\} \,.$$

Taking the minimum over all $z \in Y$, we obtain

$$\phi(x) = F(x,y^*) \leq \min_{z \in Y} \max_{y \in Y} \ \{F(x,y) \quad \text{s.t. } f(x,y) \leq f(x,z)\} \,.$$

Next, we establish the reverse inequality. Consider the specific choice $z = y^*$, since $y^* \in \mathcal{S}(x)$, we have

$$\max_{y \in Y} \ \{F(x,y) \quad \text{s.t. } f(x,y) \leq f(x,y^*)\} = \max_{y \in Y} \ \{F(x,y) \quad \text{s.t. } y \in \mathcal{S}(x)\} = \phi(x).$$

Thus, we conclude that

$$\min_{z \in Y} \max_{y \in Y} \ \{F(x,y) \quad \text{s.t. } f(x,y) \leq f(x,z)\} \leq \phi(x).$$

This completes the proof. $\qquad\square$

## B.2 Proof for Theorem 2.1

The proof strategy is analogous to that employed in [69, Lemma A.1]. We proceed by first analyzing an auxiliary function and then leveraging its properties to establish the differentiability of $\phi_{\rho,\sigma}(x)$.

Let us define an auxiliary function $h(x,z)$ as:

$$h(x,z) := \max_{y \in Y} \psi_{\rho,\sigma}(x,y,z) = -\min_{y \in Y} -\psi_{\rho,\sigma}(x,y,z).$$

By assumption, $\psi_{\rho,\sigma}(x,y,z)$ is continuous differentiable on $X \times Y \times Y$, and $-\psi_{\rho,\sigma}(x,y,z)$ is $\mu$-strongly convex with respect to $y$ for any $(x,z) \in X \times Y$.

The $\mu$-strongly convexity of $-\psi_{\rho,\sigma}(x,y,z)$ with respect to $y$ ensures the uniqueness of the minimizer of $\min_{y \in Y} -\psi_{\rho,\sigma}(x,y,z)$ (equivalently, the maximizer of $\max_{y \in Y} \psi_{\rho,\sigma}(x,y,z)$). Let us denote this unique maximizer as $\widehat{y}^*(x,z)$. Furthermore, it can be shown that $-\psi_{\rho,\sigma}(x,\cdot,z)$ satisfies the inf-compactness condition as stated in [14, Theorem 4.13] on any point $(\bar{x},\bar{z}) \in X \times Y$. Specifically, for any $(\bar{x},\bar{z}) \in X \times Y$, there exists a constant $c \in \mathbb{R}$, a compact set $B \subset \mathbb{R}^m$, and a neighborhood $W$ of $(\bar{x},\bar{z})$ such that the level set $\{y \in Y \mid -\psi_{\rho,\sigma}(x,y,z) \leq c\}$ is nonempty and contained in $B$ for all $(x,z) \in W$.

Given that $\psi_{\rho,\sigma}(x,y,z)$ is continuously differentiable, $\widehat{y}^*(x,z)$ is unique, and the inf-compactness condition holds, we can apply [14, Theorem 4.13, Remark 4.14]. This theorem implies that $h(x,z)$ is differentiable on $X \times Y$, and its gradient is given by:

$$\nabla h(x,z) = \left( \nabla_x \psi_{\rho,\sigma}(x,\widehat{y}^*(x,z),z), \nabla_z \psi_{\rho,\sigma}(x,\widehat{y}^*(x,z),z) \right). \tag{23}$$

The strong concavity of $\psi_{\rho,\sigma}(x,y,z)$ in $y$ and the continuous differentiability of $\psi_{\rho,\sigma}$ imply that $\widehat{y}^*(x,z)$ is continuous on $X \times Y$. Since $\nabla_x \psi_{\rho,\sigma}$ and $\nabla_z \psi_{\rho,\sigma}$ are continuous by assumption, and $\widehat{y}^*(x,z)$ is continuous, it follows from (23) that $\nabla h(x,z)$ is continuous on $X \times Y$. Thus, $h(x,z)$ is continuously differentiable on $X \times Y$.

The function $\phi_{\rho,\sigma}(x)$ can be expressed using $h(x,z)$ as:

$$\phi_{\rho,\sigma}(x) = \min_{z \in Y} h(x,z). \tag{24}$$

We are given that $\psi_{\rho,\sigma}(x,y,z)$ is $\sigma$-strongly convex with respect to $z$ for any fixed $(x,y) \in X \times Y$. Since $h(x,z) := \max_{y \in Y} \psi_{\rho,\sigma}(x,y,z)$, and the maximum of a set of functions preserves strong convexity (see, e.g., [10, Theorem 2.16]), it can be shown that $h(x,z)$ is $\sigma$-strongly convex with respect to $z$ for any fixed $x \in X$. The $\sigma$-strong convexity of $h(x,z)$ with respect to $z$ ensures the uniqueness of the minimizer $z_{\rho,\sigma}^*(x) = \arg\min_{z \in Y} h(x,z)$. This strong convexity, combined with the established continuous differentiability (and thus continuity) of $h(x,z)$, ensures that $h(x,z)$ satisfies the inf-compactness condition for $z$ for any $x \in X$. Furthermore, $z_{\rho,\sigma}^*(x)$ is continuous on $X$.

We can again apply [14, Theorem 4.13, Remark 4.14] to $\phi_{\rho,\sigma}(x) = \min_{z \in Y} h(x,z)$. The conditions are met: $h(x,z)$ is continuously differentiable (as shown above), and $z_{\rho,\sigma}^*(x)$ is unique. Therefore, $\phi_{\rho,\sigma}(x)$ is differentiable on $X$, and its gradient is given by:

$$\nabla \phi_{\rho,\sigma}(x) = \nabla_x h(x,z^*) = \nabla_x \psi_{\rho,\sigma}(x,\widehat{y}^*(x,z^*),z^*),$$

where $z^*$ denotes $z_{\rho,\sigma}^*(x)$. Since $\nabla_x \psi_{\rho,\sigma}(x,y,z)$ is continuous on $X \times Y \times Y$, $\widehat{y}^*(x,z)$ is continuous on $X \times Y$ and $z_{\rho,\sigma}^*(x)$ is continuous on $X$, the composite function $\nabla \phi_{\rho,\sigma}(x)$ is continuous on $X$. Thus, $\phi_{\rho,\sigma}(x)$ is continuously differentiable.

Additionally, because $\psi_{\rho,\sigma}(x,y,z)$ is strongly concave in $y$ and strongly convex in $z$ for any $x \in X$, and because $\min_{z \in Y} \max_{y \in Y} \psi_{\rho,\sigma}(x,y,z) = \max_{y \in Y} \min_{z \in Y} \psi_{\rho,\sigma}(x,y,z)$ for any $x \in X$, it follows that $\widehat{y}^*(x,z^*) = y_{\rho,\sigma}^*(x)$ for any $x \in X$. Thus, the desired conclusion is obtained.

## B.3 Proof for Lemma 2.2

Before presenting the proof for Lemma 2.2, we first establish some auxiliary results. Throughout this subsection, given sequences $\{\rho_k\}$ and $\{\sigma_k\}$, we will use the shorthand notations $\phi_k(x)$, $\psi_k(x,y,z)$, $y_k^*(x)$ and $z_k^*(x)$ to denote $\phi_{\rho_k,\sigma_k}(x)$, $\psi_{\rho_k,\sigma_k}(x,y,z)$, $y_{\rho_k,\sigma_k}^*(x)$ and $z_{\rho_k,\sigma_k}^*(x)$, respectively, for notational brevity.

First, we establish a uniform boundedness property for the saddle point components $y_k^*(x)$ and $z_k^*(x)$.

**Lemma B.2** *Let $\{\rho_k\}$ and $\{\sigma_k\}$ be sequences such that $\rho_k \to \infty$ and $\sigma_k \to 0$ as $k \to \infty$. Let $B \subset X$ be a compact set. Then, there exists a constant $M > 0$ such that for all $k$ and all $x \in B$,*

$$\|y_k^*(x)\| \le M, \quad \text{and} \quad \|z_k^*(x)\| \le M,$$

*where $(y_k^*(x), z_k^*(x))$ is the unique saddle point of the minimax problem $\min_{z \in Y} \max_{y \in Y} \psi_{\rho_k,\sigma_k}(x,y,z)$.*

*Proof.* The proof proceeds in two parts, establishing the boundedness of $\{y_k^*(x)\}$ and $\{z_k^*(x)\}$ separately, both by contradiction.

Suppose, for the sake of contradiction, that $\{y_k^*(x)\}$ is not uniformly bounded. Then there exists a sequence $\{x_k\} \subset B$ such that $\|y_k^*(x_k)\| \to \infty$ as $k \to \infty$.

By Assumption 2, for each $x_k \in B$, there exits $\hat{y}_k, \hat{z}_k$ such that $\hat{y}_k = \hat{z}_k \in \mathcal{S}(x_k) \cap D$, where $D \subset Y$ is a compact set. Thus, the sequences $\{\hat{y}_k\}$ and $\{\hat{z}_k\}$ are uniformly bounded. Since $F(x,y)$ is continuous differentiable on $X \times Y$ and is $\mu$-strongly concave in $y$ for any $x \in X$, and $\sigma_k \to 0$, we have

$$\lim_{k \to \infty} F(x_k, y_k^*(x_k)) + \frac{\sigma_k}{2}\|\hat{z}_k\|^2 - \sigma_k \langle y_k^*(x_k), \hat{z}_k \rangle = -\infty. \tag{25}$$

Next, since $\hat{z}_k \in \mathcal{S}(x_k)$, we know that $f(x_k, y_k^*(x_k)) \geq f(x_k, \hat{z}_k)$. Given $\rho_k > 0$, it follows that:

$$
\psi_k(x_k, y_k^*(x_k), \hat{z}_k)
$$
$$
= F(x_k, y_k^*(x_k)) - \rho_k(f(x_k, y_k^*(x_k)) - f(x_k, \hat{z}_k)) + \frac{\sigma_k}{2}\|\hat{z}_k\|^2 - \sigma_k\langle y_k^*(x_k), \hat{z}_k\rangle
$$
$$
\leq F(x_k, y_k^*(x_k)) + \frac{\sigma_k}{2}\|\hat{z}_k\|^2 - \sigma_k\langle y_k^*(x_k), \hat{z}_k\rangle.
$$

From (25), we deduce:
$$
\lim_{k\to\infty} \psi_k(x_k, y_k^*(x_k), \hat{z}_k) = -\infty. \tag{26}
$$

By the saddle point property of $(y_k^*(x_k), z_k^*(x_k))$:
$$
\psi_k(x_k, y_k^*(x_k), \hat{z}_k) \geq \psi_k(x_k, y_k^*(x_k), z_k^*(x_k)) \geq \psi_k(x_k, \hat{y}_k, z_k^*(x_k)).
$$

Combining this with (26) yields:
$$
\lim_{k\to\infty} \psi_k(x_k, \hat{y}_k, z_k^*(x_k)) = -\infty. \tag{27}
$$

Since $\hat{y}_k \in \mathcal{S}(x_k)$, we have $f(x_k, z_k^*(x_k)) \geq f(x_k, \hat{y}_k)$. Thus:

$$
\psi_k(x_k, \hat{y}_k, z_k^*(x_k)) = F(x_k, \hat{y}_k) - \rho_k(f(x_k, \hat{y}_k) - f(x_k, z_k^*(x_k)) + \frac{\sigma_k}{2}\|z_k^*(x_k)\|^2 - \sigma_k\langle \hat{y}_k, z_k^*(x_k)\rangle
$$
$$
\geq F(x_k, \hat{y}_k) + \frac{\sigma_k}{2}\|z_k^*(x_k)\|^2 - \sigma_k\langle \hat{y}_k, z_k^*(x_k)\rangle
$$
$$
= F(x_k, \hat{y}_k) + \frac{\sigma_k}{2}\|z_k^*(x_k) - \hat{y}_k\|^2 - \frac{\sigma_k}{2}\|\hat{y}_k\|^2
$$
$$
\geq F(x_k, \hat{y}_k) - \frac{\sigma_k}{2}\|\hat{y}_k\|^2.
$$

Since $\{x_k\}$ and $\{\hat{y}_k\}$ are bounded, and $F(x, y)$ is continuous on $X \times Y$, $F(x_k, \hat{y}_k) - \frac{\sigma_k}{2}\|\hat{y}_k\|^2$ is bounded. As $\sigma_k \to 0$, the term $F(x_k, \hat{y}_k) - \frac{\sigma_k}{2}\|\hat{y}_k\|^2$ is bounded below. This contradicts (27). Therefore, our initial assumption was false, and there must exist $M > 0$ such that $\|y_k^*(x)\| \leq M$ for all $k$ and $x \in B$.

Next, we show that there exists $M > 0$ such that $\|z_k^*(x)\| \leq M$ for any $k$ and $x \in B$. Suppose, for the sake of contradiction, that $\{z_k^*(x)\}$ is not uniformly bounded. Then there exists sequence $\{x_k\} \subset B$ such that $\|z_k^*(x_k)\| \to \infty$ as $k \to \infty$. By Assumption 2, for each $x_k$, there exists $\hat{z}_k \in \mathcal{S}(x_k) \cap D$ for a compact set $D$, so $\{\hat{z}_k\}$ is bounded.

From the saddle point property, $z_k^*(x_k)$ minimizes $\psi_k(x_k, y_k^*(x_k), z)$ over $z \in Y$. Thus:
$$
\psi_k(x_k, y_k^*(x_k), z_k^*(x_k)) \leq \psi_k(x_k, y_k^*(x_k).\hat{z}_k)
$$

Expanding this inequality, simplifying and rearranging terms:
$$
\rho_k f(x_k, z_k^*(x_k)) + \frac{\sigma_k}{2}\|z_k^*(x_k) - y_k^*(x_k)\|^2 \leq \rho_k f(x_k, \hat{z}_k) + \frac{\sigma_k}{2}\|\hat{z}_k - y_k^*(x_k)\|^2.
$$

Combining the above inequality with the fact that $f(x_k, z_k^*(x_k)) \geq f(x_k, \hat{z}_k)$ yields that:
$$
\|z_k^*(x_k) - y_k^*(x_k)\|^2 \leq \frac{2\rho_k}{\sigma_k}\left(f(x_k, \hat{z}_k) - f(x_k, z_k^*(x_k))\right) + \|\hat{z}_k - y_k^*(x_k)\|^2 \leq \|\hat{z}_k - y_k^*(x_k)\|^2. \tag{28}
$$

The right-hand side of (28) is bounded because $\{\hat{z}_k\}$ and $\{y_k^*(x_k)\}$ are bounded. However, since $\|z_k^*(x_k)\| \to \infty$ and $\{y_k^*(x_k)\}$ is bounded, the left-hand side $\|z_k^*(x_k) - y_k^*(x_k)\|^2 \to \infty$. This presents a contradiction. Thus, our assumption was false, and there exists $M > 0$ such that for any $k$ and $x \in B$, $\|z_k^*(x)\| \leq M$. $\qquad\square$

Next, we demonstrate that accumulation points of $\{y_{\rho_k, \sigma_k}^*(x)\}$ belong to the solution set $\mathcal{S}(\bar{x})$ when $x_k \to \bar{x}$.

**Lemma B.3** *Let $\{\rho_k\}$ and $\{\sigma_k\}$ be sequences such that $\rho_k \to \infty$ and $\sigma_k \to 0$ as $k \to \infty$. Then, for any sequence $\{x_k\} \subset X$ such that $x_k \to \bar{x} \in X$ as $k \to \infty$, we have*
$$
\lim_{k\to\infty} f(x_k, y_k^*(x_k)) \leq \min_{y\in Y} f(\bar{x}, y). \tag{29}
$$

*Consequently, for any accumulation point $\bar{y}$ of sequence $\{y_k^*(x_k)\}$, we have $\bar{y} \in \mathcal{S}(\bar{x})$.*

*Proof.* Let $\hat{y}$ be an arbitrary point in $\mathcal{S}(\bar{x})$. From the saddle point property, $y_k^*(x_k)$ maximizes $\psi_k(x_k, y, z_k^*(x_k))$ over $y \in Y$. Thus:

$$\psi_k(x_k, y_k^*(x_k), z_k^*(x_k)) \geq \psi_k(x_k, \hat{y}, z_k^*(x_k)).$$

Expanding this inequality:

$$F(x_k, y_k^*(x_k)) - \rho_k f(x_k, y_k^*(x_k)) - \sigma_k \langle y_k^*(x_k), z_k^*(x_k)\rangle \geq F(x_k, \hat{y}) - \rho_k f(x_k, \hat{y}) - \sigma_k \langle \hat{y}, z_k^*(x_k)\rangle.$$

Rearranging this inequality to isolate terms involving $f$, and since $\rho_k > 0$, we can divide by $\rho_k$:

$$f(x_k, y_k^*(x_k)) - f(x_k, \hat{y}) \leq \frac{1}{\rho_k}\left(F(x_k, y_k^*(x_k)) - F(x_k, \hat{y})\right) + \frac{\sigma_k}{\rho_k}\|y_k^*(x_k) - \hat{y}\|\|z_k^*(x_k)\|.$$

By Lemma B.2, $\{y_k^*(x_k)\}$ and $\{z_k^*(x_k)\}$ are uniformly bounded. Since $F(x, y)$ is continuous on $X \times Y$ and $\{x_k\}$ converges, $F(x_k, y_k^*(x_k))$ and $F(x_k, \hat{y})$ are bounded. Given $\rho_k \to \infty$ and $\sigma_k \to 0$, the entire right-hand side of the inequality converges to 0 as $k \to \infty$. Therefore, by taking $k \to \infty$ in the above inequality, and since $f(x, y)$ is continuous on $X \times Y$, we have

$$\limsup_{k \to \infty} f(x_k, y_k^*(x_k)) \leq \lim_{k \to \infty} f(x_k, \hat{y}) = \min_{y \in Y} f(\bar{x}, y).$$

This concludes the proof. $\qquad\square$

*proof of Lemma 2.2.* We prove the first statement (i.e., $\limsup\limits_{k \to \infty} \phi_k(\bar{x}) \leq \phi(\bar{x})$) by contradiction. Suppose there exist $\bar{x} \in X$ and $\delta > 0$ such that

$$\limsup_{k \to \infty} \phi_k(\bar{x}) > \phi(\bar{x}) + \delta.$$

Then, by properties of $\limsup$, there exists a subsequence (which we re-index by $k$ for simplicity) such that

$$\lim_{k \to \infty} \phi_k(\bar{x}) > \phi(\bar{x}) + \delta.$$

Recall that $(y_k^*(\bar{x}), z_k^*(\bar{x}))$ is the saddle point for the minimax problem $\min_{z \in Y} \max_{y \in Y} \psi_k(\bar{x}, y, z)$. Thus, $\phi_k(\bar{x}) = \psi_k(\bar{x}, y_k^*(\bar{x}), z_k^*(\bar{x}))$. Expanding $\psi_k$, we have:

$$F(\bar{x}, y_k^*(\bar{x})) - \rho_k(f(\bar{x}, y_k^*(\bar{x})) - f(\bar{x}, z_k^*(\bar{x})) + \frac{\sigma_k}{2}\|z_k^*(\bar{x})\|^2 - \sigma_k\langle y_k^*(\bar{x}), z_k^*(\bar{x})\rangle \geq \phi(\bar{x}) + \delta. \quad (30)$$

By Lemma B.2, $\{y_k^*(\bar{x})\}$ is bounded. Thus, we can extract a further subsequence (again re-indexed by $k$) such that $y_k^*(\bar{x}) \to \bar{y}$ for some $\bar{y} \in Y$. By Lemma B.3, this implies $\bar{y} \in \mathcal{S}(\bar{x})$.

From the saddle point property, $z_k^*(\bar{x})$ minimizes $\psi_k(\bar{x}, y_k^*(\bar{x}), z)$ over $z \in Y$. Therefore,

$$\psi_k(\bar{x}, y_k^*(\bar{x}), z_k^*(\bar{x})) \leq \psi_k(\bar{x}, y_k^*(\bar{x}), y_k^*(\bar{x})).$$

Expanding this:

$$\rho_k f(\bar{x}, z_k^*(\bar{x})) + \frac{\sigma_k}{2}\|z_k^*(\bar{x})\|^2 - \sigma_k\langle y_k^*(\bar{x}), z_k^*(\bar{x})\rangle \leq \rho_k f(\bar{x}, y_k^*(\bar{x})) - \frac{\sigma_k}{2}\|y_k^*(\bar{x})\|^2.$$

Rearranging:

$$\rho_k\left(f(\bar{x}, z_k^*(\bar{x})) - f(\bar{x}, y_k^*(\bar{x}))\right) + \frac{\sigma_k}{2}\|z_k^*(\bar{x}) - y_k^*(\bar{x})\|^2 \leq 0.$$

Combing this with (30) yields that

$$F(\bar{x}, y_k^*(\bar{x})) - \frac{\sigma_k}{2}\|y_k^*(\bar{x})\|^2 \geq \phi(\bar{x}) + \delta.$$

Taking $k \to \infty$ in the above inequality, since $F(x, y)$ is continuous on $X \times Y$, $\{y_k^*(\bar{x})\}$ is bounded and $\sigma_k \to 0$, we have

$$F(\bar{x}, \bar{y}) \geq \phi(\bar{x}) + \delta.$$

However, since $\bar{y} \in \mathcal{S}(\bar{x})$, by the definition $\phi(\bar{x}) = \max_{y \in \mathcal{S}(\bar{x})} F(\bar{x}, y)$, we must have $F(\bar{x}, \bar{y}) \leq \phi(\bar{x})$. This leads to $\phi(\bar{x}) \geq F(\bar{x}, \bar{y}) \geq \phi(\bar{x}) + \delta$. Since $\delta > 0$, this is a contradiction. Therefore, the initial assumption was false, and we must have

$$\limsup_{k \to \infty} \phi_k(x) \leq \phi(x), \quad \forall x \in X.$$

The second conclusion then follows from this result and the Proposition 7.30 in [60]. $\qquad\square$

## B.4 Proof for Proposition 2.3

For any given $x \in X$, Assumption 2 ensures that the set $\mathcal{S}(x)$ is nonempty and closed. Combined with the $\mu$-strong concavity of $F(x, y)$ with respect to $y$, this guarantees the existence of a unique maximizer $y^*(x) \in \mathcal{S}(x)$ such that $\phi(x) = F(x, y^*)$, i.e., $y^*(x) = \arg\max_{y \in \mathcal{S}(x)} F(x, y)$.

We first establish a uniform boundedness property for $y^*(x)$ when $x$ is restricted to a compact set.

**Lemma B.4** *Let $B$ be a compact set in $X$. Then, there exists a constant $M > 0$ such that for any $x \in B$,*
$$\|y^*(x)\| \leq M,$$
*where $y^*(x) = \arg\max_{y \in \mathcal{S}(x)} F(x, y)$.*

*Proof.* Suppose, for the sake of contradiction, that such a uniform bound $M$ does not exist. Then there must exist a sequence $\{x_k\} \subset B$ such that $\|y^*(x_k)\| \to \infty$ as $k \to \infty$. According to Assumption 2, for each $x_k$, there exists an element $y_k \in \mathcal{S}(x_k) \cap D$, where $D$ is a specified compact set. Consequently, the sequence $\{y_k\}$ is uniformly bounded.

Because $F(x, y)$ is continuous differentiable on $X \times Y$ and is $\mu$-strongly concave in $y$ for any $x \in X$, $\|y^*(x_k)\| \to \infty$ leading to:
$$\lim_{k \to \infty} F(x_k, y^*(x_k)) = -\infty. \tag{31}$$
By the definition of $y^*(x_k)$ as the maximizer of $\max_{y \in \mathcal{S}(x_k)} F(x_k, y)$, and since $y_k \in \mathcal{S}(x_k) \cap D$, we have:
$$F(x_k, y^*(x_k)) \geq F(x_k, y_k).$$
Given that $(x_k, y^*(x_k)) \to -\infty$ from (31), it must also hold that:
$$\lim_{k \to \infty} F(x_k, y_k) = -\infty.$$

However, since both $\{x_k\}$ and $\{y_k\}$ are bounded, and $F(x, y)$ is continuous on $X \times Y$, the sequence $\{F(x_k, y_k)\}$ must be bounded below. This contradicts the finding that $F(x_k, y_k) \to -\infty$. Thus, our initial assumption must be false, and we get the conclusion. $\qquad\square$

Next, we establish an inequality relating $\phi_{\rho,\sigma}(x)$ and $\phi(x)$.

**Lemma B.5** *Let $\rho, \sigma > 0$ be given constants. Then, for any $x \in X$,*
$$\phi_{\rho,\sigma}(x) \geq \phi(x) - \frac{\sigma}{2}\|y^*(x)\|^2,$$
*where $y^*(x) = \arg\max_{y \in \mathcal{S}(x)} F(x, y)$.*

*Proof.* For notational brevity within this proof, let $y^*$ denote $y^*(x)$. Because $\psi_{\rho,\sigma}(x, y, z)$ is strongly concave in $y$ and strongly convex in $z$, we have:
$$\phi_{\rho,\sigma}(x) = \min_{z \in Y} \max_{y \in Y} \psi_{\rho,\sigma}(x, y, z) = \max_{y \in Y} \min_{z \in Y} \psi_{\rho,\sigma}(x, y, z).$$
From the max-min formulation, it follows that for any specific choice of $y$, such as $y = y^*$,
$$\phi_{\rho,\sigma}(x) \geq \min_{z \in Y} \psi_{\rho,\sigma}(x, y^*, z). \tag{32}$$
Since
$$\psi_{\rho,\sigma}(x, y^*, z) = F(x, y^*) - \rho f(x, y^*) + \rho f(x, z) + \frac{\sigma}{2}\|z\|^2 - \sigma\langle y^*, z\rangle,$$
to find $\min_{z \in Y} \psi_{\rho,\sigma}(x, y^*, z)$, we can minimize the terms dependent on $z$:
$$\arg\min_{z \in Y} \psi_{\rho,\sigma}(x, y^*, z) = \arg\min_{z \in Y} \left\{ \rho f(x, z) + \frac{\sigma}{2}\|z - y^*\|^2 \right\}.$$
Since $y^* \in \mathcal{S}(x)$, it follows that
$$\arg\min_{z \in Y} \psi_{\rho,\sigma}(x, y^*, z) = \{y^*\}.$$
Substituting $z = y^*$ into $\psi_{\rho,\sigma}(x, y^*, z)$:
$$\min_{z \in Y} \psi_{\rho,\sigma}(x, y^*, z) = \psi_{\rho,\sigma}(x, y^*, y^*) = F(x, y^*) - \frac{\sigma}{2}\|y^*\|^2 = \phi(x) - \frac{\sigma}{2}\|y^*\|^2.$$
Combining this with (32), the conclusion follows. $\qquad\square$

Now, we are ready to provide the proof for Proposition 2.3.

*Proof of Proposition 2.3.* For any $\epsilon > 0$ and for each $k$, by the definition of infimum, there exists an $x_k \in X$ such that

$$\phi_{\rho_k, \sigma_k}(x_k) \leq \inf_{x \in X} \phi_{\rho_k, \sigma_k}(x) + \epsilon. \tag{33}$$

Applying Lemma B.5 to $\phi_{\rho_k, \sigma_k}(x_k)$:

$$\phi_{\rho_k, \sigma_k}(x_k) \geq \phi(x_k) - \frac{\sigma_k}{2}\|y^*(x_k)\|^2 \geq \inf_{x \in X} \phi(x) - \frac{\sigma_k}{2}\|y^*(x_k)\|^2. \tag{34}$$

If $Y$ is bounded, we have that sequence $\{y^*(x_k)\}$ is bounded. Alternatively, if $X$ is bounded, Lemma B.4, establishes that $\{y^*(x_k)\}$ is bounded. Under either condition, since $\sigma_k \to 0$ as $k \to \infty$:

$$\lim_{k \to \infty} \frac{\sigma_k}{2}\|y^*(x_k)\|^2 = 0.$$

Combining this with (33) and (34):

$$\inf_{x \in X} \phi(x) \leq \liminf_{k \to \infty} \left( \inf_{x \in X} \phi_{\rho_k, \sigma_k}(x) \right) + \epsilon.$$

Since $\epsilon > 0$ was arbitrary, we can let $\epsilon \to 0$, yielding:

$$\inf_{x \in X} \phi(x) \leq \liminf_{k \to \infty} \left( \inf_{x \in X} \phi_{\rho_k, \sigma_k}(x) \right).$$

Then the conclusion follows by combining the above inequality with Lemma 2.2. $\qquad\square$

### B.5  Lower semi-continuity of $\phi(x)$

In this part, we demonstrate that the inner semi-continuity of the lower-level solution set mapping $\mathcal{S}(x)$ serves as a sufficient condition for the lower semi-continuity of the value function $\phi(x)$.

We begin by recalling the relevant definitions.

**Definition B.6** *A function $\phi(x) : \mathbb{R}^n \to \mathbb{R} \cup \{\infty\}$ is lower semi-continuous (l.s.c.) at $\bar{x}$ if for any sequence $\{x_k\}$ such that $x_k \to \bar{x}$ as $k \to \infty$, it holds that*

$$\phi(\bar{x}) \leq \liminf_{k \to \infty} \phi(x_k).$$

**Definition B.7** *A set-valued function $\mathcal{S}(x) : \mathbb{R}^n \rightrightarrows \mathbb{R}^m$ is inner semi-continuous at $\bar{x}$ if $\mathcal{S}(\bar{x}) \subseteq \liminf_{x \to \bar{x}} \mathcal{S}(x)$, where $\liminf_{x \to \bar{x}} \mathcal{S}(x) := \{y \mid \forall x_k \to \bar{x}, \exists y_k \in \mathcal{S}(x_k), s.t. y_k \to y\}$.*

**Lemma B.8** *If $\mathcal{S}(x)$ is inner semi-continuous at $\bar{x} \in X$, then $\phi(x)$ is lower semi-continuous at $\bar{x}$.*

*Proof.* Let $\{x_k\}$ be an arbitrary sequence such that $x_k \to \bar{x}$ as $k \to \infty$. If $\phi(\bar{x}) = -\infty$, the inequality hods trivially. Assume $\phi(\bar{x}) > -\infty$. For any $\epsilon > 0$, by the definition of supremum, there exists an element $y_\epsilon \in \mathcal{S}(\bar{x})$ such that

$$F(\bar{x}, y_\epsilon) \geq \phi(\bar{x}) - \epsilon.$$

Since $\mathcal{S}(x)$ is inner semi-continuous at $\bar{x}$, there exists a sequence $\{y_k\}$ such that $y_k \in \mathcal{S}(x_k)$ for each $k$, and

$$\lim_{k \to \infty} y_k = y_\epsilon.$$

Then, by the continuity of $F(x, y)$ and the fact that $F(x_k, y_k) \leq \phi(x_k)$, we have:

$$\phi(\bar{x}) - \epsilon \leq F(\bar{x}, y_\epsilon) = \lim_{k \to \infty} F(x_k, y_k) \leq \liminf_{k \to \infty} \phi(x_k). \tag{35}$$

Since this inequality holds for any arbitrary $\epsilon > 0$, we can let $\epsilon \to 0$ to conclude:

$$\phi(\bar{x}) \leq \liminf_{k \to \infty} \phi(x_k).$$

$\qquad\square$

## B.6 Proof for Lemma 2.4

*Proof of Lemma 2.4.* From Lemma B.5, for each $k$, we have the inequality:

$$\phi_{\rho_k,\sigma_k}(x_k) \geq \phi(x_k) - \frac{\sigma_k}{2}\|y^*(x_k)\|^2, \tag{36}$$

where $y^*(x) = \arg\max_{y \in \mathcal{S}(x)} F(x, y)$. Since the sequence $\{x_k\}$ converges to $\bar{x}$, it is bounded. By Lemma B.4, the sequence $\{y^*(x_k)\}$ is uniformly bounded. Given that $\sigma_k \to 0$ as $k \to \infty$ and $\{y^*(x_k)\}$ is bounded, it follows that

$$\lim_{k \to \infty} \frac{\sigma_k}{2}\|y^*(x_k)\|^2 = 0.$$

Combing with (36), we obtain

$$\liminf_{k \to \infty} \phi(x_k) \leq \liminf_{k \to \infty} \phi_{\rho_k,\sigma_k}(x_k).$$

Thus, by the lower semi-continuity of $\phi(x)$ at $\bar{x}$, we conclude that:

$$\phi(\bar{x}) \leq \liminf_{k \to \infty} \phi(x_k) \leq \liminf_{k \to \infty} \phi_{\rho_k,\sigma_k}(x_k). \tag{37}$$

This completes the proof. $\qquad\square$

## B.7 Proof for Theorem 2.6

*Proof for Theorem 2.6.* Since $\{x_k\}$ is bounded, Lemmas B.2 and B.4 imply that the sequences $\{y^*(x_k)\}$ and $\{(y_k^*(x_k), z_k^*(x_k))\}$ are also bounded.

First, we show that $\mathcal{S}(x)$ is outer semi-continuous on $X$.

Let $\{(x_j, y_j)\}$ be an arbitrary sequence such that $x_j \in X$, $y_j \in \mathcal{S}(x_j)$ and $(x_j, y_j) \to (\tilde{x}, \tilde{y})$ as $j \to \infty$. Since $y_j \in \mathcal{S}(x_j)$, we have

$$f(x_j, y_j) \leq f(x_j, y^*(\tilde{x})).$$

Taking the limit as $j \to \infty$ and using the continuity of $f$ on $X \times Y$, we obtain

$$f(\tilde{x}, \tilde{y}) \leq f(\tilde{x}, y^*(\tilde{x})) = \min_{y \in Y} f(\tilde{x}, y).$$

Hence, $\tilde{y} \in \mathcal{S}(\tilde{x})$, which shows that $\mathcal{S}(x)$ is outer semi-continuous on $X$.

Next, we establish that

$$\lim_{k \to \infty} y^*(x_k) = y^*(\bar{x}).$$

Let $\{y^*(x_j)\}$ be any subsequence of $\{y^*(x_k)\}$ such that $y_j^* \to \bar{y}$ as $j \to \infty$. By assumption, $x_j \to \bar{x}$. From the outer semi-continuity of $\mathcal{S}(x)$ established above, it follows that $\bar{y} \in \mathcal{S}(\bar{x})$. Moreover, by definition $\phi(x_j) = F(x_j, y^*(x_j))$ and $\phi(\bar{x}) = F(\bar{x}, y^*(\bar{x}))$. Using the continuity of $F$ on $X \times Y$ and the lower semi-continuity of $\phi$, we obtain

$$F(\bar{x}, \bar{y}) = \lim_{j \to \infty} F(x_j, y^*(x_j)) = \lim_{j \to \infty} \phi(x_j) \geq \phi(\bar{x}) = F(\bar{x}, y^*(\bar{x})).$$

Since $\bar{y} \in \mathcal{S}(\bar{x})$, the above inequality implies that $\bar{y} \in \arg\max_{y \in \mathcal{S}(\bar{x})} F(\bar{x}, y)$. Because $\mathcal{S}(\bar{x})$ is convex, and $F(\bar{x}, y)$ is strongly concave in $y$, this maximizer is unique, so $\bar{y} = y^*(\bar{x})$. Hence any accumulation point of sequence $\{y^*(x_k)\}$ equals $y^*(\bar{x})$. Since $\{y^*(x_k)\}$ is bounded, we conclude that $\lim_{k \to \infty} y^*(x_k) = y^*(\bar{x})$.

Third, we show that

$$\lim_{k \to \infty} y_k^*(x_k) = y^*(\bar{x}).$$

Let $\{y_j^*(x_j)\}$ be any subsequence of $\{y_k^*(x_k)\}$ such that $y_j^*(x_j) \to \bar{y}$ as $j \to \infty$. By Lemma B.3, we have $\bar{y} \in \mathcal{S}(\bar{x})$. Since $y_j^*(x_j)$ maximizes $\psi_j(x_j, y, z_j^*(x_j))$ over $y \in Y$, it follows that

$$\psi_j(x_j, y_j^*(x_j), z_j^*(x_j)) \geq \psi_j(x_j, y^*(x_j), z_j^*(x_j)).$$

Expanding both sides gives

$$F(x_j, y_j^*(x_j)) - F(x_j, y^*(x_j)) \geq \rho_j \left( f(x_j, y_j^*(x_j)) - f(x_j, y^*(x_j)) \right)$$
$$+ \sigma_j \left( \langle y_j^*(x_j), z_j^*(x_j) \rangle - \langle y^*(x_j), z_j^*(x_j) \rangle \right).$$

Since $y^*(x_j) \in \mathcal{S}(x_j)$, we have $f(x_j, y_j^*(x_j)) - f(x_j, y^*(x_j)) \geq 0$ and thus

$$F(x_j, y_j^*(x_j)) - F(x_j, y^*(x_j)) \geq \sigma_j \left( \langle y_j^*(x_j), z_j^*(x_j) \rangle - \langle y^*(x_j), z_j^*(x_j) \rangle \right)$$

Taking the limit as $j \to \infty$ and using the continuity of $F$, together with $\lim_{k \to \infty} y^*(x_k) = y^*(\bar{x})$, yields

$$F(\bar{x}, \bar{y}) - F(\bar{x}, y^*(\bar{x})) \geq 0.$$

Since $\bar{y} \in \mathcal{S}(\bar{x})$, this implies that $\bar{y} \in \arg\max_{y \in \mathcal{S}(\bar{x})} F(\bar{x}, y)$. Since $\mathcal{S}(\bar{x})$ is convex, and $F(\bar{x}, y)$ is strongly concave in $y$, we must have $\bar{y} = y^*(\bar{x})$. Therefore, all accumulation points of $\{y_k^*(x_k)\}$ equal $y^*(\bar{x})$. Then, the boundedness of $\{y_k^*(x_k)\}$ implies that $\lim_{k \to \infty} y_k^*(x_k) = y^*(\bar{x})$.

Finally, we show that

$$\lim_{k \to \infty} z_k^*(x_k) = y^*(\bar{x}).$$

Since $z_k^*(x_k)$ minimizes $\psi_k(x_k, y_k^*(x_k), z)$ over $z \in Y$, we have

$$\psi_k(x_k, y_k^*(x_k), z_k^*(x_k)) \leq \psi_k(x_k, y_k^*(x_k), y^*(x_k)).$$

Expanding this inequality gives

$$\rho_k f(x_k, z_k^*(x_k)) + \frac{\sigma_k}{2} \|z_k^*(x_k)\|^2 - \sigma_k \langle y_k^*(x_k), z_k^*(x_k) \rangle$$
$$\leq \rho_k f(x_k, y^*(x_k)) + \frac{\sigma_k}{2} \|y^*(x_k)\|^2 - \sigma_k \langle y_k^*(x_k), y^*(x_k) \rangle.$$

Rearranging terms yields

$$\rho_k \left( f(x_k, z_k^*(x_k)) - f(x_k, y^*(x_k)) \right) + \frac{\sigma_k}{2} \|z_k^*(x_k) - y_k^*(x_k)\|^2 \leq \frac{\sigma_k}{2} \|y^*(x_k) - y_k^*(x_k)\|^2.$$

Since $y^*(x_k) \in \mathcal{S}(x_k)$, it follows that $f(x_k, z_k^*(x_k)) - f(x_k, y^*(x_k) \geq 0$ and hence

$$\|z_k^*(x_k) - y_k^*(x_k)\|^2 \leq \|y^*(x_k) - y_k^*(x_k)\|^2.$$

Because $\lim_{k \to \infty} y^*(x_k) = y^*(\bar{x}) = \lim_{k \to \infty} y_k^*(x_k)$, we have

$$\lim_{k \to \infty} \|y^*(x_k) - y_k^*(x_k)\| = 0.$$

Taking $k \to \infty$ in the above inequality yields

$$\lim_{k \to \infty} \|z_k^*(x_k) - y_k^*(x_k)\| = 0,$$

and consequently,

$$\lim_{k \to \infty} z_k^*(x_k) = \lim_{k \to \infty} y_k^*(x_k) = y^*(\bar{x}).$$

This completes the proof.

$\qquad\square$

## C Proof for Section 4

Throughout this part, we assume Assumption 4, which states that $X$ is a bounded set.

Given sequences $\rho_k$ and $\sigma_k$, for notational conciseness, we employ the shorthand notations $\phi_k(x)$, $\psi_k(x, y, z)$, $y_k^*(x)$ and $z_k^*(x)$ to denote $\phi_{\rho_k, \sigma_k}(x)$, $\psi_{\rho_k, \sigma_k}(x, y, z)$, $y_{\rho_k, \sigma_k}^*(x)$ and $z_{\rho_k, \sigma_k}^*(x)$, respectively. We use $u$ to denote the pair $u := (y, z)$, and correspondingly, $u^k := (y^k, z^k)$ and $u_k^*(x) = (y_k^*(x), z_k^*(x))$. The symbols $\mathcal{N}_X(x)$, $\mathcal{N}_Y(y)$ and $\mathcal{N}_{Y \times Y}(x, y)$ denote the normal cones to the sets $X$, $Y$ and $Y \times Y$ at $x$, $y$ and $(x, y)$, respectively.

Let $L_F$ and $L_f$ denote the Lipschitz constants of $\nabla F(x, y)$ and $\nabla f(x, y)$ on $X \times Y$, respectively.

Consider the sequences $\{\rho_k\}$ and $\{\sigma_k\}$ such that $\rho_k \to \infty$ and $\sigma_k \to 0$ as $k \to \infty$. As established in Lemma B.2, the quantity $M_y := \sup_{k,x \in X} \max\{\|y_k^*(x)\|, \|z_k^*(x)\|\}$ is finite. This implies that the collections of points $\{y_k^*(x)\}$ and $\{z_k^*(x)\}$ are bounded. Given that $X$ is bounded, and $f$ and its gradient $\nabla f$ are assumed to be continuous on $X \times Y$, the continuity over this effectively bounded domain of evaluation ensures that the suprema $M_f := \sup_{k,x \in X} \max\{|f(x, y_k^*(x))|, |f(x, z_k^*(x))|\}$ and $M_{\nabla f} := \sup_{k,x \in X} \max\{\|\nabla f(x, y_k^*(x))\|, \|\nabla f(x, z_k^*(x))\|\}$ are also finite.

With given sequences $\{\rho_k\}$ and $\{\sigma_k\}$, for each $k$, we define the operator $T_k : \mathbb{R}^{n+2m} \to \mathbb{R}^{2m}$ as

$$T_k(x, y, z) := (-\nabla_y \psi_k(x, y, z), \nabla_z \psi_k(x, y, z)).$$

By assumption, for any fixed $x \in X$, the function $\psi_k(x, y, z)$ is $\sigma_k$-strongly convex in $z$ and $\mu$-strongly concave in $y$. Consequently, invoking [60, Theorem 12.17 and Exercise 12.59], it follows that for a fixed $x \in X$, the operator $T_k(x, \cdot, \cdot)$ exhibits strong monotonicity with respect to $(y, z)$:

$$\langle T_k(x, u) - T_k(x, u'), u - u' \rangle \geq \mu \|y - y'\|^2 + \sigma_k \|z - z'\|^2, \quad \forall u, u' \in Y \times Y. \tag{38}$$

Furthermore, under the assumption that the gradients of $F$ and $f$ are Lipschitz continuous, the operator $T_k(x, \cdot, \cdot)$ is also Lipschitz continuous with respect to $(y, z)$ for any fixed $x \in X$:

$$\|T_k(x, u) - T_k(x, u')\| \leq \max\{(L_F + \rho_k L_f + \sigma_k), (\rho_k L_f + 2\sigma_k)\} \|u - u'\|, \quad \forall u, u' \in Y \times Y. \tag{39}$$

## C.1 Auxiliary Lemmas

To establish the convergence properties of SiPBA(Algorithm 1), we first introduce several auxiliary lemmas pertaining to the behavior of the iterative sequence. The following lemma demonstrates a contraction property for the sequence $u_k := (y_k, z_k)$.

**Lemma C.1** *Let $\{\rho_k\}$ and $\{\sigma_k\}$ be sequences such that $\rho_k, \sigma_k > 0$. Define $\bar{\sigma}_k = \min\{\sigma_k, \mu\}$. Suppose the step-size sequence $\{\beta_k\}$ satisfies $0 < \beta_k < \frac{\bar{\sigma}_k}{(L_F + \rho_k L_f + 2\sigma_k)^2}$ for each $k$. Let $\{(x^k, y^k, z^k)\}$ be the sequence generated by SiPBA(Algorithm 1). Then, the iterate $u^k$ and $u^{k+1}$ satisfy:*

$$\|u^{k+1} - u_k^*(x^k)\|^2 \leq (1 - \bar{\sigma}_k \beta_k)\|u^k - u_k^*(x^k)\|^2. \tag{40}$$

*Proof.* The update rule for $u^{k+1}$ can be expressed in the compact form:

$$u^{k+1} = \text{Proj}_{Y \times Y}\left(u^k - \beta_k T(x^k, u^k)\right).$$

Recall that $u_k^*(x^k) = (y_k^*(x^k), z_k^*(x^k))$ is is the unique saddle point of the minimax problem $\min_{z \in Y} \max_{y \in Y} \psi_k(x^k, y, z)$. From the first-order optimality conditions, $u_k^*(x^k)$ satisfies:

$$0 \in T_k(x^k, u_k^*(x^k)) + \mathcal{N}_{Y \times Y}(u_k^*(x^k)),$$

which implies

$$u_k^*(x^k) = \text{Proj}_{Y \times Y}\left(u_k^*(x^k) - \beta_k T(x^k, u_k^*(x^k))\right).$$

Utilizing the non-expansiveness of the projection operator, the strongly monotonicity of $T_k$ in (38) and its Lipschitz continuity with respect to $u$ in (39), we can apply standard results from the analysis of projected fixed-point iterations [24, Theorem 12.1.2]. If the step size $\beta_k \in (0, 2\min\{\sigma_k, \mu\}/(L_F + \rho_k L_f + 2\sigma_k)^2)$, then:

$$\|u^{k+1} - u_k^*(x^k)\|^2 \leq (1 + (L_F + \rho_k L_f + 2\sigma_k)^2 \beta_k^2 - 2\beta_k \min\{\sigma_k, \mu\})\|u^k - u_k^*(x^k)\|^2.$$

Thus, when $0 < \beta_k < \frac{\min\{\sigma_k, \mu\}}{(L_F + \rho_k L_f + 2\sigma_k)^2}$, it holds that

$$\|u^{k+1} - u_k^*(x^k)\|^2 \leq (1 - \beta_k \min\{\sigma_k, \mu\})\|u^k - u_k^*(x^k)\|^2.$$

$\square$

The subsequent lemma is dedicated to establishing the Lipschitz continuity of $u_k^*(x)$.

**Lemma C.2** *Let $\{\rho_k\}$ and $\{\sigma_k\}$ be sequences such that $\rho_k, \sigma_k > 0$. Define $\bar{\sigma}_k = \min\{\sigma_k, \mu\}$. Then, for any $x, x' \in X$, the corresponding saddle points $u_k^*(x)$ and $u_k^*(x')$ satisfy:*

$$\|u_k^*(x') - u_k^*(x)\| \leq \frac{L_F + 2\rho_k L_f}{\bar{\sigma}_k} \|x' - x\|. \tag{41}$$

*Proof.* Because $u_k^*(x) = (y_k^*(x), z_k^*(x))$ and $u_k^*(x') = (y_k^*(x'), z_k^*(x'))$ are saddle points to the minimax problem $\min_{z \in Y} \max_{y \in Y} \psi_k(x, y, z)$, and $\min_{z \in Y} \max_{y \in Y} \psi_k(x', y, z)$, respectively. According to the first-order optimality conditions, these saddle points must satisfy:

$$0 \in T_k(x, u_k^*(x)) + \mathcal{N}_{Y \times Y}(u_k^*(x)), \tag{42}$$

and

$$0 \in T_k(x', u_k^*(x')) + \mathcal{N}_{Y \times Y}(u_k^*(x')).$$

Next, we analyze the Lipschitz continuity of $T_k(x, u)$ with respect to $x$. The first component difference is

$$- \nabla_y \psi_k(x', u_k^*(x')) + \nabla_y \psi_k(x, u_k^*(x'))$$
$$= - \nabla_y F(x', y_k^*(x')) + \nabla_y F(x, y_k^*(x')) + \rho_k \left( \nabla_y f(x', y_k^*(x')) - \nabla_y f(x, y_k^*(x')) \right).$$

And the second component difference is

$$\nabla_z \psi_k(x', u_k^*(x')) - \nabla_z \psi_k(x, u_k^*(x')) = \rho_k \left( \nabla_y f(x', z_k^*(x')) - \nabla_y f(x, z_k^*(x')) \right).$$

Thus, we obtain

$$\|T_k(x', u_k^*(x')) - T_k(x, u_k^*(x'))\| \leq L_F \|x' - x\| + 2\rho_k L_f \|x' - x\|. \tag{43}$$

Next, we use the fact that

$$T_k(x, u_k^*(x')) - T_k(x', u_k^*(x')) \in T_k(x, u_k^*(x')) + \mathcal{N}_{Y \times Y}(u_k^*(x')).$$

and apply the strongly monotonicity of $T_k$ from (38), along with the monotonicity of the normal cone $\mathcal{N}_{Y \times Y}$ and (45). This leads to the following inequality:

$$\mu \|y_k^*(x') - y_k^*(x)\|^2 + \sigma_k \|z_k^*(x') - z_k^*(x)\|^2$$
$$\leq \langle T_k(x, u_k^*(x')) - T_k(x', u_k^*(x')), u_k^*(x') - u_k^*(x) \rangle$$
$$\leq \|T_k(x, u_k^*(x')) - T_k(x', u_k^*(x'))\| \|u_k^*(x') - u_k^*(x)\|.$$

By substituting the bound from (43) into the above inequality, we obtain

$$\min\{\sigma_k, \mu\} \|u_k^*(x') - u_k^*(x)\| \leq L_F \|x' - x\| + 2\rho_k L_f \|x' - x\|.$$

This completes the proof.

$\square$

**Lemma C.3** *Let $\{\rho_k\}$ and $\{\sigma_k\}$ be sequences such that $\rho_{k+1} \geq \rho_k > 0$, $\sigma_k \geq \sigma_{k+1} > 0$. Define $\bar{\sigma}_k = \min\{\sigma_k, \mu\}$. Then, for any fixed $x \in X$, we have*

$$\|u_{k+1}^*(x) - u_k^*(x)\| \leq \frac{2(\rho_{k+1} - \rho_k)}{\bar{\sigma}_k} M_{\nabla f} + \frac{3(\sigma_k - \sigma_{k+1})}{\bar{\sigma}_k} M_y. \tag{44}$$

*Proof.* Because $u_k^*(x) = (y_k^*(x), z_k^*(x))$ and $u_{k+1}^*(x) = (y_{k+1}^*(x), z_{k+1}^*(x))$ are saddle points to the minimax problem $\min_{z \in Y} \max_{y \in Y} \psi_k(x, y, z)$, and $\min_{z \in Y} \max_{y \in Y} \psi_{k+1}(x, y, z)$, respectively. According to the first-order optimality conditions, these saddle points satisfy:

$$0 \in T_k(x, u_k^*(x)) + \mathcal{N}_{Y \times Y}(u_k^*(x)), \tag{45}$$

and

$$0 \in T_{k+1}(x, u_{k+1}^*(x)) + \mathcal{N}_{Y \times Y}(u_{k+1}^*(x)).$$

Next, we expand the differences between the gradients of $\psi_k$ and $\psi_{k+1}$ at $u_{k+1}^*(x)$:

$$- \nabla_y \psi_{k+1}(x, u_{k+1}^*(x)) + \nabla_y \psi_k(x, u_{k+1}^*(x))$$
$$= (\rho_{k+1} - \rho_k) \nabla_y f(x, y_{k+1}^*(x)) + (\sigma_{k+1} - \sigma_k) z_{k+1}^*(x),$$

and

$$\nabla_z \psi_{k+1}(x, u^*_{k+1}(x)) - \nabla_z \psi_k(x, u^*_{k+1}(x))$$
$$= (\rho_{k+1} - \rho_k)\nabla_y f(x, z^*_{k+1}(x)) + (\sigma_{k+1} - \sigma_k)\left(z^*_{k+1}(x) - y^*_{k+1}(x)\right).$$

Thus, we have the following bound for the difference between the operators $T_k$ and $T_{k+1}$:

$$\|T_{k+1}(x, u^*_{k+1}(x)) - T_k(x, u^*_{k+1}(x))\| \leq 2(\rho_{k+1} - \rho_k)M_{\nabla f} + 3(\sigma_k - \sigma_{k+1})M_y. \qquad (46)$$

Now, using the fact that

$$T_k(x, u^*_{k+1}(x)) - T_{k+1}(x, u^*_{k+1}(x)) \in T_k(x, u^*_{k+1}(x)) + \mathcal{N}_{Y \times Y}(u^*_{k+1}(x)),$$

and combining this with the strongly monotonicity of $T_k$ from (38), the monotonicity of the normal cone $\mathcal{N}_{Y \times Y}$ and (45), we get

$$\mu\|y^*_{k+1}(x) - y^*_k(x)\|^2 + \sigma_k\|z^*_{k+1}(x) - z^*_k(x)\|^2$$
$$\leq \langle T_{k+1}(x, u^*_{k+1}(x)) - T_k(x, u^*_{k+1}(x)), u^*_k(x) - u^*_{k+1}(x)\rangle$$
$$\leq \|T_{k+1}(x, u^*_{k+1}(x)) - T_k(x, u^*_{k+1}(x))\|\|u^*_{k+1}(x) - u^*_k(x)\|.$$

By substituting the bound from (46) into this inequality, we obtain

$$\min\{\sigma_k, \mu\}\|u^*_{k+1}(x) - u^*_k(x)\| \leq 2(\rho_{k+1} - \rho_k)M_{\nabla f} + 3(\sigma_k - \sigma_{k+1})M_y.$$

This completes the proof. □

**Lemma C.4** *Let $\{\rho_k\}$ and $\{\sigma_k\}$ be sequences such that $\rho_k, \sigma_k > 0$. Define $\bar{\sigma}_k = \min\{\sigma_k, \mu\}$. Then, for any $x, x' \in X$, we have*

$$\|\nabla \phi_k(x') - \nabla \phi_k(x)\| \leq L_{\phi_k}\|x' - x\|, \qquad (47)$$

*where $L_{\phi_k} := \frac{(L_F + 2\rho_k L_f)(L_F + 2\rho_k L_f + \bar{\sigma}_k)}{\bar{\sigma}_k}$.*

*Proof.* From the expression for $\nabla \phi_k(x)$ given in Theorem 2.1, we have the following:

$$\|\nabla \phi_k(x) - \nabla \phi_k(x')\| = \|\nabla_x \psi_k(x, u^*_k(x)) - \nabla_x \psi_k(x', u^*_k(x'))\|$$
$$\leq \|\nabla_x F(x, y^*_k(x)) - \nabla_x F(x', y^*_k(x'))\|$$
$$+ \rho_k\|\nabla_x f(x, y^*_k(x)) - \nabla_x f(x', y^*_k(x'))\|$$
$$+ \rho_k\|\nabla_x f(x, y^*_k(x)) - \nabla_x f(x', y^*_k(x'))\|$$
$$\leq (L_F + \rho_k L_f)(\|x - x'\| + \|y^*_k(x) - y^*_k(x')\|) \qquad (48)$$
$$+ \rho_k L_f(\|x - x'\| + \|z^*_k(x) - z^*_k(x')\|)$$
$$\leq (L_F + 2\rho_k L_f)(\|x - x'\| + \|u^*_k(x) - u^*_k(x')\|)$$
$$\leq \frac{(L_F + 2\rho_k L_f)(L_F + 2\rho_k L_f + \bar{\sigma}_k)}{\bar{\sigma}_k}\|x - x'\|$$

where the final inequality follows from Lemma C.2.

□

By synthesizing the results from Lemmas C.1-C.4, the following lemma characterizes the evolution of the squared norm of the tracking error, $\|u^k - u^*_k(x^k)\|^2$.

**Lemma C.5** *Let $\{\rho_k\}$ and $\{\sigma_k\}$ be sequences such that $\rho_{k+1} \geq \rho_k > 0$, $\sigma_k \geq \sigma_{k+1} > 0$. Define $\bar{\sigma}_k = \min\{\sigma_k, \mu\}$. Suppose the step-size sequence $\{\beta_k\}$ satisfies $0 < \beta_k < \frac{\bar{\sigma}_k}{(L_F + \rho_k L_f + 2\sigma_k)^2}$ for each $k$. Let $\{(x^k, y^k, z^k)\}$ be the sequence generated by SiPBA (Algorithm 1). Then, the following inequality holds:*

$$\|u^{k+1} - u^*_{k+1}(x^{k+1})\|^2 - \|u^k - u^*_k(x^k)\|^2$$
$$\leq -\frac{1}{2}\beta_k\bar{\sigma}_k\|u^k - u^*_k(x^k)\|^2 + 2(1 + \frac{2}{\beta_k\bar{\sigma}_k})\frac{(L_F + 2\rho_k L_f)^2}{\bar{\sigma}_k^2}\|x^{k+1} - x^k\|^2 \qquad (49)$$
$$+ 2(1 + \frac{2}{\beta_k\bar{\sigma}_k})\left(\frac{8(\rho_{k+1} - \rho_k)^2}{\bar{\sigma}_k^2}M_{\nabla f}^2 + \frac{18(\sigma_k - \sigma_{k+1})^2}{\bar{\sigma}_k^2}M_y^2\right).$$

*Proof.* Using the Cauchy-Schwarz inequality for any $\delta > 0$, we obtain the following:

$$\|u^{k+1} - u^*_{k+1}(x^{k+1})\|^2$$

$$\leq (1+\delta)\|u^{k+1} - u^*_k(x^k)\|^2 + (1+\frac{1}{\delta})\|u^*_{k+1}(x^{k+1}) - u^*_k(x^k)\|^2$$

$$\leq (1+\delta)\|u^{k+1} - u^*_k(x^k)\|^2 + 2(1+\frac{1}{\delta})\|u^*_k(x^{k+1}) - u^*_k(x^k)\|^2 \qquad (50)$$

$$+ 2(1+\frac{1}{\delta})\|u^*_{k+1}(x^{k+1}) - u^*_k(x^{k+1})\|^2.$$

Next, take $\delta = \frac{1}{2}\beta_k\bar{\sigma}_k$ in the above inequality. By applying Lemma C.1, we obtain the following bound:

$$(1+\delta)\|u^{k+1} - u^*_k(x^k)\|^2 \leq (1 - \frac{1}{2}\beta_k\bar{\sigma}_k)\|u^k - u^*_k(x^k)\|^2.$$

Using Lemma C.2, we can further bound the second term as follows:

$$2(1+\frac{1}{\delta})\|u^*_k(x^{k+1}) - u^*_k(x^k)\|^2 \leq 2(1+\frac{2}{\beta_k\bar{\sigma}_k})\frac{(L_F + 2\rho_k L_f)^2}{\bar{\sigma}_k^2}\|x^{k+1} - x^k\|^2.$$

Next, applying Lemma C.3 with $x = x^{k+1}$, we obtain

$$2(1+\frac{1}{\delta})\|u^*_{k+1}(x^{k+1}) - u^*_k(x^{k+1})\|^2$$

$$\leq 2(1+\frac{2}{\beta_k\bar{\sigma}_k})\left(\frac{8(\rho_{k+1}-\rho_k)^2}{\bar{\sigma}_k^2}M^2_{\nabla f} + \frac{18(\sigma_k - \sigma_{k+1})^2}{\bar{\sigma}_k^2}M^2_y\right).$$

Finally, combining the above three inequalities with (50), we arrive at the desired inequality.

$\square$

**Lemma C.6** *Let $\{\rho_k\}$ and $\{\sigma_k\}$ be sequences such that $\rho_{k+1} \geq \rho_k > 0$, $\sigma_k \geq \sigma_{k+1} > 0$. Then, for any $x \in X$, we have*

$$\phi_{k+1}(x) - \phi_k(x) \leq (\sigma_k - \sigma_{k+1})\frac{M^2_y}{2} + 2(\rho_{k+1} - \rho_k)M_f. \qquad (51)$$

*Proof.* We begin with the expression for $\phi_{k+1}(x)$ as follows:

$$\phi_{k+1}(x) = \min_{z\in Y}\max_{y\in Y}\psi_{k+1}(x,y,z).$$

This leads to the inequality

$$\phi_{k+1}(x) = \min_{z\in Y}\psi_{k+1}(x, y^*_{k+1}(x), z)$$

$$\leq \psi_{k+1}(x, y^*_{k+1}(x), z^*_k(x))$$

$$= F(x, y^*_{k+1}(x)) - \rho_{k+1}(f(x, y^*_{k+1}(x)) - f(x, z^*_k(x)))$$

$$+ \frac{\sigma_{k+1}}{2}\|y^*_{k+1}(x) - z^*_k(x)\|^2 - \frac{\sigma_{k+1}}{2}\|y^*_{k+1}(x)\|^2$$

$$\leq F(x, y^*_{k+1}(x)) - \rho_k(f(x, y^*_{k+1}(x)) - f(x, z^*_k(x)))$$

$$+ \frac{\sigma_k}{2}\|y^*_{k+1}(x) - z^*_k(x)\|^2 - \frac{\sigma_k}{2}\|y^*_{k+1}(x)\|^2 + \frac{\sigma_k - \sigma_{k+1}}{2}\|y^*_{k+1}(x)\|^2$$

$$+ (\rho_k - \rho_{k+1})(f(x, y^*_{k+1}(x)) - f(x, z^*_k(x)))$$

$$\leq \max_{y\in Y}\left\{F(x,y) - \rho_k(f(x,y) - f(x, z^*_k(x))) + \frac{\sigma_k}{2}\|y - z^*_k(x)\|^2 - \frac{\sigma_k}{2}\|y\|^2\right\}$$

$$+ \frac{\sigma_k - \sigma_{k+1}}{2}\|y^*_{k+1}(x)\|^2 + (\rho_k - \rho_{k+1})(f(x, y^*_{k+1}(x)) - f(x, z^*_k(x)))$$

$$\leq \max_{y\in Y}\psi_k(x, y, z^*_k(x)) + \frac{\sigma_k - \sigma_{k+1}}{2}M^2_y + 2(\rho_{k+1} - \rho_k)M_f.$$

This completes the proof, as the final inequality is derived from the fact that $\phi_k(x) = \max_{y\in Y}\psi_k(x, y, z^*_k(x))$.

$\square$

The subsequent lemma characterizes the descent property of the value function $\phi_k(x_k)$ across iterations.

**Lemma C.7** *Let $\{\rho_k\}$ and $\{\sigma_k\}$ be sequences such that $\rho_{k+1} \geq \rho_k > 0$, $\sigma_k \geq \sigma_{k+1} > 0$. Define $\bar{\sigma}_k = \min\{\sigma_k, \mu\}$. Suppose the step-size sequence $\{\beta_k\}$ satisfies $0 < \beta_k < \frac{\bar{\sigma}_k}{(L_F + \rho_k L_f + 2\sigma_k)^2}$ for each $k$. Let $\{(x^k, y^k, z^k)\}$ be the sequence generated by SiPBA (Algorithm 1). Then, we have*

$$\phi_{k+1}(x^{k+1}) - \phi_k(x^k) + \left(\frac{1}{2\alpha_k} - \frac{L_{\phi_k}}{2}\right)\|x^{k+1} - x^k\|^2$$

$$\leq \frac{\alpha_k}{2}(L_F + 2\rho_k L_f)^2(1 - \bar{\sigma}_k\beta_k)\|u^k - u_k^*(x^k)\|^2 + (\sigma_k - \sigma_{k+1})\frac{M_y^2}{2} + 2(\rho_{k+1} - \rho_k)M_f, \tag{52}$$

*where $L_{\phi_k} := \frac{(L_F + 2\rho_k L_f)(L_F + 2\rho_k L_f + \bar{\sigma}_k)}{\bar{\sigma}_k}$.*

*Proof.* We decompose the total difference as follows:

$$\phi_{k+1}(x^{k+1}) - \phi_k(x^k) = \phi_{k+1}(x^{k+1}) - \phi_k(x^{k+1}) + \phi_k(x^{k+1}) - \phi_k(x^k). \tag{53}$$

For the first term, applying Lemma C.6 with $x = x^{k+1}$:

$$\phi_{k+1}(x^{k+1}) - \phi_k(x^{k+1}) \leq (\sigma_k - \sigma_{k+1})\frac{M_y^2}{2} + 2(\rho_{k+1} - \rho_k)M_f. \tag{54}$$

For the second term, $\phi_k(x^{k+1}) - \phi_k(x^k)$, we use the $L_{\phi_k}$-Lipschitz continuity of $\nabla\phi_k(x)$ established in Lemma C.4. A standard descent inequality (cf. [10, Lemma 5.7] for smooth functions) states:

$$\phi_k(x^{k+1}) - \phi_k(x^k) \leq \langle \nabla\phi_k(x^k), x^{k+1} - x^k \rangle + \frac{L_{\phi_k}}{2}\|x^{k+1} - x^k\|^2. \tag{55}$$

Next, applying the update rule for $x^{k+1}$, we get

$$\frac{1}{\alpha_k}\|x^{k+1} - x^k\|^2 \leq \langle -\nabla_x\psi_k(x^k, y^{k+1}, z^{k+1}), x^{k+1} - x^k \rangle.$$

By combining this inequality with the previous one, and using the formula for $\nabla\phi_k(x^k)$ given in Theorem 2.1, we obtain

$$\phi_k(x^{k+1}) - \phi_k(x^k) + \left(\frac{1}{\alpha_k} - \frac{L_{\phi_k}}{2}\right)\|x^{k+1} - x^k\|^2$$

$$\leq \langle \nabla_x\psi_k(x^k, y_k^*(x^k), z_k^*(x^k)) - \nabla_x\psi_k(x^k, y^{k+1}, z^{k+1}), x^{k+1} - x^k \rangle$$

$$\leq \left((L_F + \rho_k L_f)\|y^{k+1} - y_k^*(x^k)\| + \rho_k L_f\|z^{k+1} - z_k^*(x^k)\|\right)\|x^{k+1} - x^k\| \tag{56}$$

$$\leq \frac{\alpha_k}{2}(L_F + 2\rho_k L_f)^2\|u^{k+1} - u_k^*(x^k)\|^2 + \frac{1}{2\alpha_k}\|x^{k+1} - x^k\|^2$$

$$\leq \frac{\alpha_k}{2}(L_F + 2\rho_k L_f)^2(1 - \bar{\sigma}_k\beta_k)\|u^k - u_k^*(x^k)\|^2 + \frac{1}{2\alpha_k}\|x^{k+1} - x^k\|^2,$$

where the last inequality follows from Lemma C.1. The conclusion follows by combining the above inequality with (55) and (54).

$\square$

**Lemma C.8** *Let $\{\rho_k\}$ and $\{\sigma_k\}$ be sequences such that $\rho_k > 0$, $\sigma_k > 0$ and $\sigma_k \to 0$ as $k \to \infty$. Furthermore, assume that $\phi(x)$ is bounded below on $X$, i.e., $\inf_{x \in X} \phi(x) > -\infty$. Then, there exists a constant $\underline{\phi}$ such that, for any $\{x^k\} \subset X$, we have*

$$\phi_k(x^k) \geq \underline{\phi}.$$

*Proof.* According to Lemma B.5, for any $k$, the following inequality holds:

$$\phi_k(x^k) \geq \phi(x^k) - \frac{\sigma_k}{2}\|y^*(x^k)\|^2 \geq \inf_{x \in X} \phi(x) - \frac{\sigma_k}{2}\|y^*(x^k)\|^2, \tag{57}$$

where $y^*(x) = \arg\max_{y \in \mathcal{S}(x)} F(x, y)$.

Next, by Lemma B.4, we have that there exists $M_{y^*} > 0$ such that for all $k$,

$$\|y^*(x^k)\| \leq M_{y^*}.$$

Thus, we can bound the second term in the inequality:

$$\phi_k(x^k) \geq \inf_{x \in X} \phi(x) - \frac{\sigma_k}{2} M_{y^*}^2,$$

Taking the limit as $k \to \infty$ and using the fact that $\sigma_k \to 0$, we obtain

$$\liminf_{k \to \infty} \phi_k(x^k) \geq \inf_{x \in X} \phi(x),$$

and then the conclusion follows. $\qquad\qquad\qquad\qquad\qquad\qquad\qquad\qquad\qquad\qquad\square$

## C.2 Proof for Proposition 4.1

*Proof of Proposition 4.1 .* Given $\beta_k = \beta_0 k^{-2p-q}$, and $\sigma_k = \sigma_0 k^{-q}$, the constant ratio $\beta_0/\sigma_0$ can be chosen sufficiently small to ensure that for all $k \geq 1$, the following inequality holds:

$$0 < \beta_k < \frac{\bar{\sigma}_k}{(L_F + \rho_k L_f + 2\sigma_k)^2}.$$

Recall the merit function,

$$V_k = a_k(\phi_k(x^k) - \underline{\phi}) + b_k \|u^k - u_k^*(x^k)\|^2.$$

Applying Lemmas C.7 and C.5, specifically equations (52) and (49), and using the facts that $a_{k+1} \leq a_k$ and $b_{k+1} \leq b_k$, we obtain:

$$V_{k+1} - V_k$$
$$= a_{k+1}(\phi_{k+1}(x^{k+1}) - \underline{\phi}) - a_k(\phi_k(x^k) - \underline{\phi}) + b_{k+1}\|u^{k+1} - u_{k+1}^*(x^{k+1})\|^2 - b_k\|u^k - u_k^*(x^k)\|^2$$
$$\leq a_k(\phi_{k+1}(x^{k+1}) - \phi_k(x^k)) + b_k(\|u^{k+1} - u_{k+1}^*(x^{k+1})\|^2 - \|u^k - u_k^*(x^k)\|^2)$$
$$\leq -a_k\left(\frac{1}{2\alpha_k} - \frac{L_{\phi_k}}{2}\right)\|x^{k+1} - x^k\|^2 + a_k\frac{\alpha_k}{2}(L_F + 2\rho_k L_f)^2(1 - \bar{\sigma}_k\beta_k)\|u^k - u_k^*(x^k)\|^2$$
$$+ a_k(\sigma_k - \sigma_{k+1})\frac{M_y^2}{2} + 2a_k(\rho_{k+1} - \rho_k)M_f$$
$$- \frac{1}{2}b_k\beta_k\bar{\sigma}_k\|u^k - u_k^*(x^k)\|^2 + 2b_k(1 + \frac{2}{\beta_k\bar{\sigma}_k})\frac{(L_F + 2\rho_k L_f)^2}{\bar{\sigma}_k^2}\|x^{k+1} - x^k\|^2$$
$$+ 2b_k(1 + \frac{2}{\beta_k\bar{\sigma}_k})\left(\frac{8(\rho_{k+1} - \rho_k)^2}{\bar{\sigma}_k^2}M_{\nabla f}^2 + \frac{18(\sigma_k - \sigma_{k+1})^2}{\bar{\sigma}_k^2}M_y^2\right).$$

$$(58)$$

The parameters are set according to the schedules: $\alpha_k = \alpha_0 k^{-s}$, $\beta_k = \beta_0 k^{-2p-q}$, $b_k = k^{-t}$, $\sigma_k = \sigma_0 k^{-q}$ and $\rho_k = \rho_0 k^p$. We have that

$$b_k\beta_k\sigma_k = \beta_0\sigma_0 k^{-2p-2q-t}.$$

Since $s > t + 4p + 2q$, it follows for sufficiently large $k$ that

$$\alpha_k(L_F + 2\rho_k L_f)^2 < \frac{1}{2}b_k\beta_k\bar{\sigma}_k.$$

Therefore,

$$\frac{\alpha_k}{2}(L_F + 2\rho_k L_f)^2(1 - \bar{\sigma}_k\beta_k)\|u^k - u_k^*(x^k)\|^2 - \frac{1}{2}b_k\beta_k\bar{\sigma}_k\|u^k - u_k^*(x^k)\|^2$$
$$< -\frac{1}{4}b_k\beta_k\bar{\sigma}_k\|u^k - u_k^*(x^k)\|^2$$

Furthermore, since $a_k = k^{-s}$, $b_k = k^{-t}$, $\sigma_k = \sigma_0 k^{-q}$ and $\rho_k = \rho_0 k^p$, we find that there exists $C > 0$ such that

$$\frac{b_k}{a_k}(1 + \frac{2}{\beta_k\bar{\sigma}_k})\frac{(L_F + 2\rho_k L_f)^2}{\bar{\sigma}_k^2} \leq Ck^{s-t+4p+4q},$$

and
$$L_{\phi_k} = \frac{(L_F + 2\rho_k L_f)(L_F + 2\rho_k L_f + \bar{\sigma}_k)}{\bar{\sigma}_k} \leq Ck^{2p+q}.$$

Given that $\alpha_k = \alpha_0 k^{-s}$, $t > 4p + 4q$ and $s > t + 4p + 2q > 2p + q$, we conclude that for sufficiently large $k$:
$$\frac{1}{2\alpha_k} - \frac{L_{\phi_k}}{2} - \frac{2b_k}{a_k}(1 + \frac{2}{\beta_k \bar{\sigma}_k})\frac{(L_F + 2\rho_k L_f)^2}{\bar{\sigma}_k^2} > \frac{1}{4\alpha_k}.$$

Substituting this and the earlier bound into (58), we deduce that for large $k$:

$$\begin{aligned}
V_{k+1} - V_k \leq &-\frac{a_k}{4\alpha_k}\|x^{k+1} - x^k\|^2 - \frac{1}{4}b_k \beta_k \bar{\sigma}_k \|u^k - u_k^*(x^k)\|^2 \\
&+ a_k(\sigma_k - \sigma_{k+1})\frac{M_y^2}{2} + 2a_k(\rho_{k+1} - \rho_k)M_f \\
&+ 2b_k(1 + \frac{2}{\beta_k \bar{\sigma}_k})\left(\frac{8(\rho_{k+1} - \rho_k)^2}{\bar{\sigma}_k^2}M_{\nabla f}^2 + \frac{18(\sigma_k - \sigma_{k+1})^2}{\bar{\sigma}_k^2}M_y^2\right).
\end{aligned} \tag{59}$$

Next, we show that the sum of the positive terms on the right-hand side of (59) is bounded. Since $a_k = k^{-s} \leq 1$ and $\sigma_k = \sigma_0 k^{-q}$, we have
$$\sum_{k=1}^{\infty} a_k(\sigma_k - \sigma_{k+1}) \leq \sum_{k=1}^{\infty}(\sigma_k - \sigma_{k+1}) \leq \sigma_0.$$

With $a_k = k^{-s}$, $\rho_k = \rho_0 k^p$ and $s > 2p + q$, there exits $C > 0$ such that
$$a_k(\rho_{k+1} - \rho_k) \leq Ck^{-2p-q}((k+1)^p - k^p) \leq Ck^{-p-q}\frac{p}{k} \leq pCk^{-p-q-1},$$

which implies
$$\sum_{k=1}^{\infty} 2a_k(\rho_{k+1} - \rho_k)M_f < \infty.$$

Regarding the remaining terms, since $\beta_k = \beta_0 k^{-2p-q}$, $b_k = k^{-t}$, $\sigma_k = \sigma_0 k^{-q}$, $\rho_k = \rho_0 k^p$ and $t > 4p + 4q$, there exists $C > 0$ such that

$$\begin{aligned}
b_k(1 + \frac{2}{\beta_k \bar{\sigma}_k})\frac{(\rho_{k+1} - \rho_k)^2}{\bar{\sigma}_k^2} &\leq Ck^{-t+2p+4q}((k+1)^p - k^p)^2 \\
&\leq Ck^{-t+4p+4q}\frac{p^2}{k^2} \\
&\leq p^2 Ck^{-t+4p+4q-2}.
\end{aligned}$$

Thus, the sum
$$\sum_{k=1}^{\infty} 2b_k(1 + \frac{2}{\beta_k \bar{\sigma}_k})\frac{8(\rho_{k+1} - \rho_k)^2}{\bar{\sigma}_k^2}M_{\nabla f}^2 < \infty.$$

Similarly, there exists $C > 0$ such that
$$2b_k(1 + \frac{2}{\beta_k \bar{\sigma}_k})\frac{18(\sigma_k - \sigma_{k+1})^2}{\bar{\sigma}_k^2} \leq Ck^{-t+2p+2q-2}.$$

Since $t > 2p + 2q$, the sum
$$\sum_{k=1}^{\infty} 2b_k(1 + \frac{2}{\beta_k \bar{\sigma}_k})\frac{18(\sigma_k - \sigma_{k+1})^2}{\bar{\sigma}_k^2}M_y^2 < \infty.$$

This completes the proof. □

## C.3 Proof for Theorem 4.2

*Proof of Theorem 4.2.* The conditions $s \geq 8p + 8q$ and $t = 4p + 5q$ are chosen to satisfy the requirements of Proposition 4.1. From Proposition 4.1, and noting that $V_k \geq 0$ for all $k$, we have the following summations:

$$\sum_{k=1}^{\infty} \frac{a_k}{\alpha_k} \|x^{k+1} - x^k\|^2 + \sum_{k=1}^{\infty} b_k \beta_k \bar{\sigma}_k \|u^k - u_k^*(x^k)\|^2 < \infty.$$

Rewriting the first sum, we have:

$$\sum_{k=1}^{\infty} a_k \alpha_k \frac{1}{\alpha_k^2} \|x^{k+1} - x^k\|^2 + \sum_{k=1}^{\infty} b_k \beta_k \bar{\sigma}_k \|u^k - u_k^*(x^k)\|^2 < \infty. \tag{60}$$

Since the terms in these convergent series are non-negative, it follows that for any $K > 0$:

$$\min_{0 < k < K} a_k \alpha_k \frac{1}{\alpha_k^2} \|x^{k+1} - x^k\|^2 = O(1/K), \quad \text{and} \quad \min_{0 < k < K} b_k \beta_k \bar{\sigma}_k \|u^k - u_k^*(x^k)\|^2 = O(1/K).$$

The parameter schedules are $\alpha_k = \alpha_0 k^{-s}$, $\beta_k = \beta_0 k^{-2p-q}$, $a_k = k^{-s}$, $b_k = k^{-4p-5q}$, $\sigma_k = \sigma_0 k^{-q}$ and $\rho_k = \rho_0 k^p$. We have

$$a_k \alpha_k = \alpha_0 k^{-2s}, \quad \text{and} \quad b_k \beta_k \bar{\sigma}_k = \beta_0 \sigma_0 k^{-6p-7q}.$$

Under the conditions $s < 1/2$ and $6p + 7q < 1$, we we deduce the convergence rates:

$$\min_{0 < k < K} \frac{1}{\alpha_k^2} \|x^{k+1} - x^k\|^2 = O(1/K^{1-2s}),$$

and

$$\min_{0 < k < K} \|u^k - u_k^*(x^k)\|^2 = O(1/K^{1-6p-7q}).$$

Furthermore, since $b_k \beta_k \sigma_k / (L_F + 2\rho_k L_f) = O(1/K^{-7p-7q})$ and $7p - 7q < 1$, the summability in (60) implies

$$\liminf_{k \to \infty} \max\{\|x^{k+1} - x^k\|/\alpha_k, \ (L_F + 2\rho_k L_f)\|u^k - u_k^*(x^k)\|, \|u^k - u_k^*(x^k)\|\} = 0.$$

From Theorem 2.1, we have

$$\frac{1}{\alpha_k} \|\mathrm{Proj}_{\mathcal{X}}(x_k - \alpha_k \nabla \phi_k(x^k) - \mathrm{Proj}_{\mathcal{X}}(x_k - \alpha_k d_x^k)\|$$
$$\leq \|\nabla \phi_k(x^k) - d_x^k)\|$$
$$= \|\nabla_x \psi_k(x^k, y_k^*(x^k), z_k^*(x^k)) - \nabla_x \psi_k(x^k, y^{k+1}, z^{k+1})\|$$
$$\leq (L_F + 2\rho_k L_f)\|u^{k+1} - u_k^*(x^k)\|$$
$$\leq (L_F + 2\rho_k L_f)\|u^k - u_k^*(x^k)\|,$$

where the first inequality follows from the nonexpansiveness of the projection operator $\mathrm{Proj}_{\mathcal{X}}$, and the last inequality follows from Lemma C.1. Combining this with the previous equality yields:

$$\liminf_{k \to \infty} \frac{1}{\alpha_k} \|x^k - \mathrm{Proj}_{\mathcal{X}}(x_k - \alpha_k \nabla \phi_k(x^k))\|$$
$$= \liminf_{k \to \infty} \frac{1}{\alpha_k} \|x^k - \mathrm{Proj}_{\mathcal{X}}(x_k - \alpha_k \nabla \phi_k(x^k)) + \mathrm{Proj}_{\mathcal{X}}(x_k - \alpha_k d_x^k) - x^{k+1}\|$$
$$\leq \liminf_{k \to \infty} \left( \frac{1}{\alpha_k} \|\mathrm{Proj}_{\mathcal{X}}(x_k - \alpha_k \nabla \phi_k(x^k)) - \mathrm{Proj}_{\mathcal{X}}(x_k - \alpha_k d_x^k)\| + \|x^k - x^{k+1}\| \right)$$
$$\leq \liminf_{k \to \infty} (L_F + 2\rho_k L_f)\|u^k - u_k^*(x^k)\| + \|x^k - x^{k+1}\|$$
$$\leq \liminf_{k \to \infty} 2 \max\{\|x^{k+1} - x^k\|/\alpha_k, \ (L_F + 2\rho_k L_f)\|u^k - u_k^*(x^k)\|\}$$
$$= 0.$$

This completes the proof of stationarity. $\qquad\square$

## C.4  Proof for Corollary 4.3

We first establish the following auxiliary result.

**Lemma C.9** *Assume $\phi(x)$ is lower semi-continuous on $X$. Let $\{\rho_k\}$ and $\{\sigma_k\}$ be sequences such that $\rho_k \to \infty$ and $\sigma_k \to 0$. Then, for any $\epsilon > 0$, there exists $K > 0$ such that for all $k \geq K$,*

$$\phi_k(x) \leq \phi(x) + \epsilon, \qquad \forall x \in X. \tag{61}$$

*Proof.* Assume, for the sake of contradiction, that the statement is false. Then there exists an $\epsilon_0 > 0$ and sequence $\{x_k\} \subset X$ such that

$$\lim_{k \to \infty} \phi_k(x_k) > \phi(x_k) + \epsilon_0.$$

Since $X$ is compact, by passing to a further subsequence if necessary, we can assume $x_k \to \bar{x} \in X$. Recall that $(y_k^*(x_k), z_k^*(x_k))$ is the saddle point of the minimax problem $\min_{z \in Y} \max_{y \in Y} \psi_k(x_k, y, z)$. Thus, $\phi_k(x_k) = \psi_k(x_k, y_k^*(x_k), z_k^*(x_k))$ and by the definition of $\psi_k$, we have

$$F(x_k, y_k^*(x_k)) - \rho_k(f(x_k, y_k^*(x_k)) - f(x_k, z_k^*(x_k)) + \frac{\sigma_k}{2}\|z_k^*(x_k)\|^2 - \sigma_k \langle y_k^*(x_k), z_k^*(x_k) \rangle$$

$$\geq \phi(x_k) + \epsilon.$$

$$\tag{62}$$

By Lemma B.2, the sequence $\{y_k^*(x_k)\}$ is bounded. Thus, by passing to another subsequence if necessary, we can assume $y_k^*(x_k) \to \bar{y}$ for some $\bar{y} \in Y$. It follows from Lemma B.3 that $\bar{y} \in \mathcal{S}(\bar{x})$.

Since $z_k^*(x_k)$ is a minimizer of $\psi_k(x_k, y^*(x_k), z)$ over $z \in Y$, we have

$$\psi_k(x_k, y_k^*(x_k), z_k^*(x_k)) \leq \psi_k(x_k, y_k^*(x_k), y_k^*(x_k)).$$

Substituting the definition of $\psi_k$, this yields:

$$\rho_k f(x_k, z_k^*(\bar{x})) + \frac{\sigma_k}{2}\|z_k^*(x_k)\|^2 - \sigma_k \langle y_k^*(x_k), z_k^*(x_k) \rangle \leq \rho_k f(x_k, y_k^*(x_k)) - \frac{\sigma_k}{2}\|y_k^*(x_k)\|^2.$$

This simplifies to:

$$\rho_k \left( f(\bar{x}, z_k^*(x_k)) - f(\bar{x}, y_k^*(x_k)) \right) + \frac{\sigma_k}{2}\|z_k^*(x_k) - y_k^*(x_k)\|^2 \leq 0.$$

Combining this with (62), we have

$$F(x_k, y_k^*(x_k)) \geq \phi(x_k) + \epsilon.$$

Taking the limit as $k \to \infty$, continuity of $F(x, y)$ and lower semicontinuity of $\phi(x)$ yield

$$F(\bar{x}, \bar{y}) \geq \phi(\bar{x}) + \epsilon.$$

However, since $\bar{y} \in \mathcal{S}(\bar{x})$, we must have

$$\phi(\bar{x}) \geq F(\bar{x}, \bar{y}),$$

leading to a contradiction. Thus, the claim follows. $\qquad \square$

*Proof of Corollary 4.3.* From Theorem 4.2, we have the stationarity condition:

$$\liminf_{k \to \infty} \frac{1}{\alpha_k}\|x^k - \text{Proj}_{\mathcal{X}}(x_k - \alpha_k \nabla \phi_k(x^k))\| = 0.$$

Thus, we can find a subsequence $\{x^{k_j}\}$ such that $\lim_{j \to \infty} \|x^{k_j} - \text{Proj}_{\mathcal{X}}(x_{k_j} - \alpha_{k_j} \nabla \phi_{k_j}(x^{k_j}))\| = 0$. This condition, together with the Lipschitz continuity of $\nabla \phi_{k_j}$ established in Lemma C.4, implies that $x^{k_j}$ is an approximate stationary point for $\phi_{k_j}$. Standard arguments then show that for any $\epsilon > 0$, there exists $K_0 > 0$ such that for all $k_j \geq K_0$, there is a $\delta_j > 0$ such that

$$\phi_{k_j}(x) \geq \phi_{k_j}(x^{k_j}) - \epsilon\|x - x^{k_j}\|, \qquad \forall x \in \mathbb{B}_{\delta_j}(x^{k_j}) \cap X.$$

According to Lemma B.5, for each $k_j$:

$$\phi_{k_j}(x^{k_j}) \geq \phi(x^{k_j}) - \frac{\sigma_{k_j}}{2}\|y^*(x^{k_j})\|^2 \tag{63}$$

where $y^*(x) = \arg\max_{y \in \mathcal{S}(x)} F(x, y)$. By Lemma B.4, there exists $M_{y^*} > 0$ such that for any $k_j$,

$$\|y^*(x^{k_j})\| \le M_{y^*}.$$

Since $\sigma_{k_j} \to 0$ as $k_j \to \infty$, we can obtain from (63) that for any $\tilde{\epsilon} > 0$, there exists $K_0 > 0$ such that for each $k_j \ge K$,

$$\phi_{k_j}(x^{k_j}) \ge \phi(x^{k_j}) - \frac{\tilde{\epsilon}}{2}.$$

Furthermore, by Lemma C.9, for any $\tilde{\epsilon} > 0$, there exists $K_0 > 0$ such that for any $k_j \ge K_0$,

$$\phi_{k_j}(x) \le \phi(x) + \frac{\tilde{\epsilon}}{2}, \qquad \forall x \in X.$$

Combining these inequalities yields that for any $\epsilon > 0$ and $\tilde{\epsilon} > 0$, we can find $K_0 > 0$ such that for each $k_j \ge K_0$, there exists $\delta_j > 0$ such that

$$\phi(x) \ge \phi(x^{k_j}) - \epsilon\|x - x^{k_j}\| - \tilde{\epsilon}, \qquad \forall x \in \mathbb{B}_{\delta_{k_j}}(x^j) \cap X.$$

Since for all $K > K_0$, we can find some $k_j > K$, the proof is completed. $\qquad\square$

