# OpenReview forum: "A Single-Loop Gradient Algorithm for Pessimistic Bilevel Optimization via Smooth Approximation"
_NeurIPS.cc/2025/Conference — NeurIPS 2025 poster_

### Official Review · Reviewer_roeL · 2025-06-22

**Clarity:** 3
**Significance:** 2
**Originality:** 3
**Rating:** 3
**Confidence:** 5

**Summary:**

The paper presents a novel approach to solving Pessimistic Bilevel Optimization (PBO) problems by proposing a single-loop gradient-based algorithm named SiPBA. The core technique involves constructing a smooth approximation of the PBO problem using penalization and regularization methods. This approximation transforms the original non-smooth PBO into a smooth optimization problem, enabling the application of efficient gradient-based methods. The authors provide theoretical validation for the proposed smooth approximation scheme and establish convergence guarantees.

**Questions:**

See the above weakness.

**Ethical Concerns:**

["NO or VERY MINOR ethics concerns only"]

**Final Justification:**

Partial issues are well addressed. And I decided to raise the scores to 3.

The current experiments are limited to low-dimensional cases and some classical statistical problems. Indeed, the bilevel optimization has been widely extended to many fields in machine learning scenarios, which this work ignores.

Theoretically, the **min-max and max-min type problems are not equally consistent** except for the convex-concave scenario [Sion's Theorem]. This is the key concern in my review, **which still has not been addressed in the authors' response**. **Some assumptions are indeed rarely shown in some previous relevant works**, which could be some kind of **strong without applicable examples in practice**.

Moreover, this is a good algorithm for some specific tasks, e.g., maximum AUC bilevel problem.

**Limitations:**

yes

**Quality:**

2

**Strengths And Weaknesses:**

Strengths

1. Algorithmic Innovation. The paper successfully relaxes the original PBO problem to a strongly convex-strongly concave formulation, enabling the development of a single-loop first-order gradient algorithm.

2. Theoretical and Empirical Validation. The authors provide rigorous theoretical analysis, including convergence rates and stationarity conditions. Additionally, the empirical results demonstrate the effectiveness and efficiency.

3. The paper is well-structured and clearly presented. The technical details are well-explained, and the mathematical derivations are thorough.

Weaknesses

1. Related Work. This paper lacks a detailed discussion of existing work on first-order optimization algorithms and single-loop methods for both Optimistic and Pessimistic Bilevel Optimization [1-7]. This omission makes it difficult to assess the novelty and improvements over existing methods. Additionally, the lack of experimental comparisons with other PBO algorithms limits the evaluation of the proposed method's performance.

[1] A Fully Single Loop Algorithm for Bilevel Optimization without Hessian Inverse

[2] BOME! Bilevel Optimization Made Easy: A Simple First-Order Approach

[3] Hyperparameter Tuning Through Pessimistic Bilevel Optimization

[4] A fully first-order method for stochastic bilevel optimization

[5] One way to solve pessimistic bilevel optimization problems

[6] On penalty methods for nonconvex bilevel optimization and first-order stochastic approximation

[7] Scholtes Relaxation Method for Pessimistic Bilevel Optimization

2. Theoretical Gaps. The paper constructs an approximation function that ensures strong convexity in \(z\) and strong concavity in \(y\). While this is a simple strategy for problem relaxation, I am not quite sure about the strong concavity property. Moreover, the min-max problem is transformed into a max-min problem using Sion's theorem. Such relaxation is similar to conjugate gradient method, which require statements on the prima-dual gap. The authors are kindly suggested to provide a theoretical bound on the gap between the optimal solutions of the original and approximated problems.

3. Convergence Results. The convergence results presented in Proposition 4.1 and Theorem 4.2 are somewhat weak. The derivation of the merit function \(V_k\) includes additional constant terms that are not thoroughly discussed. I’ve checked partial proof. It is unclear how these terms are introduced or whether they depend on specific assumptions or parameter choices.

Additionally, the paper only provides bounds on the minimum of the error terms \min_k \|x^{k+1}-x^k\|, which does not strongly support the algorithm's convergence guarantees. A more detailed analysis of the asymptotic or expected convergence rates of V_{k+1} - V_k, x^k-x^* or x^{k+1} - x^k would be beneficial.

4. The authors are encouraged to analyze the algorithmic (computational and storage) complexity [8], which is crucial for comparison among existing algorithms, and understanding the algorithm's practical applicability and performance in large-scale problems.

[8] On the complexity of first-order methods in stochastic bilevel optimization

5. The experimental section lacks comparisons with other existing PBO optimization algorithms . Some commonly investigated problems in PBO [9,10] are also suggested in the experiments, instead of merely on simple synthetic data.

[9] The Standard Pessimistic Bilevel Problem

[10] Revisiting and Advancing Fast Adversarial Training Through The Lens of Bi-Level Optimization

---

> ### Author Rebuttal · Authors · 2025-07-31
>
> We sincerely thank the reviewer for their time and comments. We are grateful for their acknowledgment of our work's primary contributions and appreciate the opportunity to clarify our work and address the concerns raised.
>
> Below, we address each point in detail.
>
> ---
>
> ## 1. On Related Work and Experimental Comparisons
>
> We appreciate the reviewer's suggestions for related work and would like to clarify how our paper is positioned with respect to these references. We also take this opportunity to present new experimental results that further benchmark our method against relevant algorithms.
>
> ### Discussion of Prior Literature:
>
> The reviewer mentioned several references, a number of which were already cited and discussed in our manuscript.
>
> * References [2, 4, 6, 7] in the reviewer's comments correspond to references [38, 34, 35, 11] in our paper, and their relationship to our work is discussed in the Related Work part.
>
> * Regarding the other references suggested:
>
>     * Reference [1]: This work studies first-order algorithms for optimistic bilevel optimization. Our paper, in contrast, focuses on the pessimistic formulation (PBO), which addresses a different class of problems.
>
>     * Reference [3]: This paper applies PBO models to hyperparameter tuning. The algorithm it employs is the one proposed in our reference [11], which we have already discussed.
>
>     * Reference [5]: This is a very recent work (published April 2025) that tackles a specific subclass of PBO where the upper-level objective is quadratic and the lower-level is linear. It proposes a single-level DC (Difference of Convex) program reformulation. The resulting algorithm is neither single-loop nor gradient-based in the same vein as our method, making it less directly comparable.
>
> Furthermore, the reviewer suggested experiments on problems from [9, 10]. We have carefully examined these works and concluded they are not suitable for benchmarking our method.
>
> * Reference [9]: The PBO applications in this work are from economics and involve integer variables, which falls outside the continuous optimization setting considered in our paper and the broader machine learning context.
>
> * Reference[10]: This work focuses on the optimistic bilevel optimization model, which is fundamentally different from the PBO problem we study.
>
> Given this, we respectfully maintain that our original manuscript provided a detailed discussion of the relevant existing literature.
>
> ### Experimental Comparisons:
>
> We wish to clarify that our original manuscript did contain experimental comparisons, specifically benchmarking our proposed SiPBA against the AdaProx-PD algorithm [2] on a synthetic example. To further strengthen our evaluation and address the reviewer's concern, we have now expanded these experiments to include additional important baselines: AdaProx-SG [2] and the Compact/Detailed Scholtes relaxation methods [3]. The comprehensive results for each method are summarized in our response to Reviewer ckLr, which demonstrate that SiPBA is not only significantly faster but also more robust, consistently converging to a valid solution across all runs.
>
> ---
>
> ## 2. On the Theoretical Analysis
>
> * **Strong Concavity**: The reviewer questioned the strong concavity of $\psi_k(x,y,z)$ (defined in Eq. (3)) with respect to $y$. This property follows directly from the strong concavity of the upper-level objective $F(x,y)$ with respect to $y$, which is an assumption stated in Assumption 1.
>
> * **Min-Max Interchange**: The reviewer also raised a question about the interchange of minimization and maximization operators in Line 146: $\min_{z \in Y} \max_{y \in Y} \psi_k(x, y, z) = \max_{y \in Y} \min_{z \in Y} \psi_k(x, y, z)$. As we discussed in Lines 143-146, this equality holds due to the existence of a saddle point for $\psi_k(x, y, z) $ with respect to variables $(y,z)$. This is a direct consequence of classical min-max theorems (e.g., [VII, Theorem 4.3.1, 12] or [Theorem 3.1.29, 13]). There is no primal-dual gap in this step.
>
> * **Convergence of the Approximation**: Finally, we would like to highlight Theorem 2.5, which establishes the convergence of our smoothing approximation to the original PBO problem. This theorem formally proves that the solution to the smoothed problem converges to a true solution of the original PBO problem as the smoothing parameter approaches zero, justifying the correctness of our smoothing framework.
>
> ---
>
> ## 3. On Convergence Results
>
> * **Definition of Constants**: The reviewer commented on the constants used in the merit function $V_k$. All terms, including $\underline{\phi}$, $a_k$ and $b_k$, are explicitly defined in the paper. Specifically, $\underline{\phi}$ is discussed in Lines 223-224, and the precise formulas for $a_k$ and $b_k$ are provided in the statement of Proposition 4.1 in Line 233.
>
> * **Convergence Guarantees**: The reviewer's claim that our paper "only provides bounds on the minimum of the error terms" is a misinterpretation of our results. In Theorem 4.2, we establish the asymptotic convergence of the sequence generated by our algorithm, explicitly showing that key error terms approach zero:
>
>    $$
>   \lim_{k \rightarrow \infty} \frac{1}{\alpha_k} \|\|x^k - \mathrm{Proj}_X(x^k - \alpha_k \nabla \phi_k(x^k))\|\| = 0,
>    $$
>
>    $$
>    \lim_{k \rightarrow \infty} \|\| u^k - u_{k}^{\star}(x^k) \|\| = 0.
>    $$
>
>     Furthermore, establishing a convergence rate based on the metric $\min_{0 \le k < K} \frac{1}{\alpha_k^2} \|\|x^{k+1} - x^k\|\|^2$
>     is a standard and widely accepted practice for gradient-based methods applied to general non-convex optimization problems (see, e.g., [Theorem 10.15, 14]).
>
> ---
>
> ## 4. On Algorithmic Complexity
>
> We thank the reviewer for suggesting that we include a discussion of complexity. We will add the following analysis to the revised manuscript.
>
> * **Computational Complexity**: At each iteration, SiPBA performs gradient evaluations ($\nabla_xF, \nabla_x f, \nabla_y F, \nabla_y f$) and several vector operations. Since the cost of gradient evaluation is typically linear in the variable dimension, the per-iteration computational complexity of SiPBA is $\mathcal{O}(n + m)$.
>
> * **Storage Complexity**: The algorithm exhibits low memory requirements. To compute the next iterate, it only needs to store the variables from the preceding step $(x^k, y^k, z^k)$ and a few intermediate vectors. Consequently, the storage complexity is $\mathcal{O}(n + 2m)$.
>
> This lean computational and memory footprint confirms that SiPBA is scalable and well-suited for practical, high-dimensional applications, representing a key advantage of our approach.
>
> ### References
>
> [1] J. Li, B. Gu, and H. Huang. A fully single loop algorithm for bilevel optimization without hessian inverse. In *the AAAI Conference on Artificial Intelligence*, 2022.
>
> [2] B. Liu, M. Ye, S. J. Wright, P. Stone, and Q. Liu. BOME! Bilevel Optimization Made Easy: A Simple First-Order Approach. In *Advances in Neural Information Processing Systems*, 2022.
>
> [3] M. A. Ustun, L. Xu, B. Zeng, and X. Qian. Hyperparameter Tuning Through Pessimistic Bilevel Optimization. In *arXiv:2412.03666*, 2024.
>
> [4] J. Kwon and D. Kwon and S. J. Wright and R. D. Nowak A fully first-order method for stochastic bilevel optimization. In *International Conference on Machine Learning*, 2023.
>
> [5] A. S. Strekalovsky. One way to solve pessimistic bilevel optimization problems. In *Optimization*, 2025.
>
> [6] J. Kwon and D. Kwon and S. J. Wright and R. D. Nowak On Penalty Methods for Nonconvex Bilevel Optimization and First-Order Stochastic Approximation. In *International Conference on Learning Representations*, 2024.
>
> [7] I. Benchouk, L. O. Jolaoso, K. Nachi, and A. B. Zemkoho. Scholtes relaxation method for pessimistic bilevel optimization. *Set-Valued and Variational Analysis*, 2025.
>
> [8] J. Kwon, D. Kwon, and H. Lyu. On the complexity of first-order methods in stochastic bilevel optimization. *International Conference on Machine Learning*, 2024.
>
> [9] L. Lampariello, S. Sagratella and O. Stein. The standard pessimistic bilevel problem. *SIAM Journal on Optimization*, 2019.
>
> [10] Y. Zhang, G. Zhang, P. Khanduri, M. Hong, S. Chang and S. Liu. Revisiting and advancing fast adversarial training through the lens of bi-level optimization. *International Conference on Machine Learning*, 2022.
>
> [11] B. Zeng. A practical scheme to compute the pessimistic bilevel optimization problem. *INFORMS Journal on Computing*, 2020.
>
> [12] JB. Hiriart-Urruty and C. Lemaréchal. Convex analysis and minimization algorithms I: Fundamentals. In *Springer science and business media*, 2013.
>
> [13] Y. Nesterov. Lectures on convex optimization. In *Springer International Publishing*, 2018.
>
> [14] A. Beck. First-order methods in optimization. In *SIAM*, 2017.

---

> ### Comment · Reviewer_roeL · 2025-08-01
> **response**
>
> Thanks for your detailed response, which solves most of the raised issues! And I've raised the rate accordingly.
>
> However, some questions still remain as follows.
>
> (1) EXP: Current empirical results, including the responses, are kind of limited, which may  not be sufficient to verify the motivation for "SiPBA is scalable and well-suited for practical, high-dimensional applications, representing a key advantage of our approach". Moreover, could the authors provide the empirical results on higher dimensional cases?
>
> (2) Theory: Since the convex-concave in Assumptions 1,2 may not be satisfied in practice [1], could the author provide with some ML examples satisfying the assumptions on convex/concave PBO settings as in this paper if possible? Or some relevant PBO works with similar assumptions?
>
> I would be grateful if these concerns could also be well addressed.
>
> [1] A global optimization algorithm for generalized semi-infinite, continuous minimax with coupled constraints and bi-level problems
>
> [2] BOME! Bilevel Optimization Made Easy: A Simple First-Order Approach

---

> > ### Author Response · Authors · 2025-08-02
> >
> > We thank the reviewer for their valuable feedback and have provided responses to the new questions and concerns below.
> >
> > ### **On Empirical Results in High-Dimensional Cases**
> >
> > We appreciate the reviewer’s suggestion to provide further evidence of SiPBA's scalability. We would like to clarify that our original submission included two experiments specifically designed to demonstrate scalability, and we have now added a third experiment on a large-scale deep learning task to further strengthen this point.
> >
> > * **Existing Synthetic and Spam Classification Experiments:**
> >
> > In our synthetic experiment, we evaluated SiPBA on problems with dimensions ranging from 200 to 1,000. As shown in Figure 1(b) of our paper, the number of iterations and computation time increase gracefully with the dimensionality, providing initial evidence of scalability.
> >
> >
> > Moreover, our spam classification experiment also represents a high-dimensional application. The model is formulated as:
> > $$
> > 		\min_{w\in\mathbb{R}^n} \max_{\hat{x}} l(w,\hat{x},y) + \lambda_1 \mathrm{Reg}(w)
> > 		\quad \text{s.t.}
> > 		\hat{x} \in \arg\min_{x'\in\mathcal{X}} l'(w,x') + \lambda_2 ||\varphi(x') - \varphi(x)||^2,
> > $$
> > 	In this experiment setting, the dimension of the upper-level variable $w$ (the classifier parameters) is approximately 3,000. The lower-level variable $\hat{x}$, representing adversarial examples for a set of 500 emails, has a dimension well over 100,000. This experiment showcases SiPBA's ability to handle high-dimensional applications.
> >
> > * **New Experiment: Deep Hyper-Representation Learning:**
> >
> > To further demonstrate the practical utility and scalability of SiPBA, we have added a new experiment on deep hyper-representation learning using the following PBO formulation:
> > $$
> > \min_{\theta\in\Theta}\max_{w}\frac{1}{m_1}||f(X_{val},H)w-\mathbf{y}_{val}||^2,
> > $$
> >
> > $$
> > \text{s.t.} w\in\underset{w^\prime\in\mathcal{W}}{\arg\min} \frac{1}{m_2}||f(X_{train},H)w^\prime-\mathbf{y}_{train}||^2,
> > $$
> > where $f(\cdot, \theta)$ is a feature extractor based on the LeNet-5 architecture, and  $w$ s a subsequent linear layer. In this setup, the dimension of the upper-level variable $\theta$ is 60,856 and the dimension of the lower-level variable $w$ is 850.
> >
> > We evaluate performance on the MNIST and FashionMNIST datasets, comparing it against established baselines.  Each dataset is randomly split into 50,000 training samples, 10,000 validation samples, and 10,000 test samples. The results, measured by test accuracy, are presented below.
> >
> > | Dataset       | SiPBA    | PZOBO | AID-FP | AID-CG |
> > | :------------ | :------- | :---- | :----- | :----- |
> > | MNIST         | **99.1** | 95.8  | 97.6   | 92.5   |
> > | Fashion-MNIST | **89.6** | 78.8  | 82.1   | 78.8   |
> >
> >
> > The  performance of SiPBA in these experiments provides strong evidence of its effectiveness and applicability to high-dimensional machine learning problems.
> >
> >
> >
> > ### **On the Convex-Concave Assumptions**
> >
> > Regarding the assumptions on the objective functions (Assumption 1: concavity of $F(x,y)$ in $y$; Assumption 2: convexity of $f(x,y)$ in $y$), we would like to offer the following clarifications.
> >
> > The convexity of the lower-level objective $f(x,y)$ with respect to the lower-level variable $y$ is a condition met by many machine learning models. For instance, in our paper, the lower-level problems in both the spam classification task given above and the new deep hyper-representation experiment given above satisfy this convexity requirement.
> >
> > Furthermore, the combination of upper-level concavity (Assumption 1) and lower-level convexity (Assumption 2) is a common condition for analyzing PBO problems. These, or very similar, assumptions are utilized in other works in the PBO literature, such as the "bracket assumptions" in [1] and Assumption 1 in [2].
> >
> > [1] L. Lampariello, S. Sagratella and O. Stein. The standard pessimistic bilevel problem. SIAM Journal on Optimization, 2019.
> >
> > [2] Z. Guan, D. Sow, S. Lin, and Y. Liang. AdaProx: A novel method for bilevel optimization under pessimistic framework. Conference on Parsimony and Learning, 2025.
> >
> > We hope these clarifications adequately address the reviewer's concerns.

---

### Official Review · Reviewer_5FpQ · 2025-06-27

**Clarity:** 3
**Significance:** 3
**Originality:** 3
**Rating:** 5
**Confidence:** 5

**Summary:**

This paper presents a numerical method for solving pessimistic bilevel optimization (PBO) problems, a class of bilevel optimization that has received less attention in the literature compared to its optimistic counterpart. The authors introduce a novel smoothing technique for the PBO value function, leveraging penalization and regularization. This approach transforms the value function into a smooth function corresponding to a strongly convex-strongly concave minimax problem.

Building upon this smoothed formulation, the authors propose a fully first-order, single-loop algorithm for solving PBO.  The paper provides a rigorous theoretical convergence analysis of the proposed algorithm. Furthermore, the effectiveness and efficiency of the method are validated through a series of numerical experiments on both synthetic and real-world problems.

**Questions:**

1. On page 3, in lines 119 and 122, the word "and" appears to be duplicated. Please correct this for clarity.

2. Regarding the regularized function defined in Equation (3), could the authors elaborate on the motivation for including the coupling term $\langle y, z\rangle $? What is the specific role of this term? The reviewer understands the need to ensure the function is strongly convex with respect to $z$. Would the quadratic regularization term alone be insufficient to achieve this, or does the coupling term serve an additional, distinct purpose in the analysis or performance of the method?

3. Could the authors precisely define the "SVM" and "Logistic" models listed in Table 2 and explain how they relate to the PBO model formulated in Equation (16)? The basis for comparison in this experiment is unclear. Please specify which algorithms are being compared. Was the proposed method benchmarked against other PBO algorithms, or does the comparison involve other methodologies?

**Ethical Concerns:**

["NO or VERY MINOR ethics concerns only"]

**Final Justification:**

I have read all rebuttals, and keep my score.

**Limitations:**

Yes

**Quality:**

3

**Strengths And Weaknesses:**

Strengths :
1. This paper addresses PBO, a challenging and less-explored area of bilevel optimization. The inherent difficulty of PBO, which includes a maximization problem at the upper level, has limited the development of efficient algorithms. This work makes a valuable contribution by focusing on single-loop, first-order methods for this problem class.

2. The proposed methodology is both clear and sound. The authors introduce a novel technique to construct a smooth approximation of the PBO problem using penalization and regularization. The paper provides a detailed analysis to justify the convergence of this approximation to the true PBO problem, a contribution that may inspire further research in this domain.

3. Based on the smoothed approximation, the paper presents a single-loop algorithm that relies solely on first-order information. This design choice makes the proposed algorithm simple to implement and computationally efficient, rendering it suitable for large-scale applications.

4. The authors provide a non-asymptotic convergence analysis for the proposed algorithm. This rigorous theoretical support substantiates the algorithm's validity and strengthens the paper's overall contribution.

5. The numerical experiments demonstrate the practical effectiveness and efficiency of the proposed algorithm.

Weaknesses:
The primary area for improvement lies in the numerical experiments. Several descriptions of the experimental setup and results lack clarity. Furthermore, the empirical evaluation is somewhat limited due to the small number of baseline algorithms used for comparison. Although the literature on PBO algorithms is not extensive, several existing methods could have been included, at least in the experiments on synthetic data. A more comprehensive comparison would provide a stronger case for the advantages of the proposed approach.

---

> ### Author Rebuttal · Authors · 2025-07-30
>
> We sincerely thank the reviewer for their insightful comments and constructive suggestions. We are grateful for their acknowledgment of our work's primary contributions: proposing a gradient-based method for Pessimistic Bilevel Optimization (PBO) that avoids second-order derivatives and the need for an inner-loop optimization solver.
> Below, we address the specific concerns raised.
>
> ---
>
> ###  1. On the Weaknesses of Numerical Experiment
>
> We thank the reviewer for this feedback. We have revised the experimental section to improve clarity, particularly the spam classification setup, which we detail in our response to Question 3.
>
> We agree that including more baseline comparisons is valuable. Accordingly, we have expanded our synthetic experiments to include additional methods. As detailed in our response to Reviewer ckLr, these new results further highlight the effectiveness of our proposed method, SiPBA. We will incorporate these results into the revised manuscript.
>
> ---
>
> ###  2. On the Typo on Page 3
>
> We appreciate the reviewer for the attention to detail. This typo has been corrected in the revised manuscript.
>
> ---
>
> ###  3. On the Additional Bilinear Term in the Smoothing Approach
>
> We appreciate this insightful question. The bilinear term is indeed crucial for the theoretical guarantees of our proposed method.
>
> While adding a quadratic term to ensure strong convexity is a common technique in optimization, its specific form in our work is not arbitrary. Its inclusion is theoretically motivated and essential for the convergence of our smoothing approximation. As noted in the manuscript, this coupling term is critical for establishing the key inequality in the proof of Lemma 2.2:
>
> $$
> \rho_{k}(f(\bar{x}, z_{k}^{\star}(\bar{x})) - f(\bar{x}, y_{k}^{\star}(\bar{x}))) + \frac{\sigma_k}{2} \|\| z_{k}^{\star}(\bar{x}) - y_{k}^{\star}(\bar{x})\|\|^2 \le 0.
> $$
>
> This result forms the foundation of our subsequent convergence analysis, underpinning both Proposition 2.3 and Theorem 2.5.
>
> ---
>
> ### 4. On the "SVM" and "Logistic" Models in Table 2
>
> We apologize for the lack of clarity and thank the reviewer for the opportunity to elaborate on our experimental design for the spam classification task. The original experiments were designed to compare the performance of the PBO model trained using SiPBA with standard single-level models implemented in `scikit-learn`. For clarity, we restate the PBO formulation considered in the spam classification task:
>
> $$
> \min_{w\in\mathbb{R}^n} \max_{\hat{x}}l(w,\hat{x},y) + \lambda_1 \mathrm{Reg}(w)
> \quad \text{s.t.}
> \hat{x} \in \arg\min_{x'\in\mathcal{X}} l'(w,x') + \lambda_2 \|\|\varphi(x') - \varphi(x)\|\|^2,
> $$
>
> where $ w $ denotes the classifier parameters, $ (x,y) $ represents the vectorized training data, $ l $ (resp. $ l' $) corresponds to the classifier (resp. adversarial generator) loss, $ \mathrm{Reg}(\cdot) $ denotes the regularization term, and $ \varphi(\cdot) $ characterizes the feature mapping.
>
> We perform a two-part empirical comparison:
>
> - PBO solvers: We compare the PBO model above trained using SiPBA (with $\varphi(\mathbf{x}) $ as the top $ k $ principal components) to the same model trained using the SQP-based method with $ \varphi(\mathbf{x}) = \mathbf{x} $, as the setting in [2].
> - Single-level baselines: We also compare the SiPBA-trained PBO model against a standard single-level model:
>
> $$
> \min_{w} l(w,x,y) + \lambda_1 \mathrm{Reg}(w),
> $$
> implemented using the `scikit-learn` library.
>
> For both $ l $ and $ l' $, we consider hinge loss and cross-entropy loss, leading to the following method names:
>
> - SiPBA-Hinge / SiPBA-CE: Our proposed PBO solver using SiPBA.
> - SQP-Hinge / SQP-CE: SQP-based method from [3].
> - Single-Hinge / Single-CE: Standard single-level classifiers corresponding to SVM and Logistic Regression.
>
> The results over ten runs are summarized in the table below, which indicate that the PBO model (either trained with SQP or SiPBA) exhibits superior cross-domain performance compared to the single-level models. Moreover, the SiPBA-trained models achieve the best overall accuracy and F1 score.
>
> > **Table**: Accuracy (Acc) and F1 score (F1) on four spam corpora, training on TREC06, TREC07, EnronSpam or LingSpam.
>
> | Train Set | Model         | TREC06       | TREC07       | EnronSpam   | LingSpam    | Ave          |
> |-----------|---------------|--------------|--------------|-------------|-------------|--------------|
> | TREC06    | SiPBA-Hinge   | 96.4/94.7    | 87.3/81.0    | 70.6/70.2   | 87.6/92.7   | **85.5/84.7** |
> |           | SiPBA-CE      | 94.5/92.5    | 79.5/73.0    | 70.9/71.8   | 87.6/92.8   | **83.1/82.5** |
> |           | SQP-Hinge     | 93.1/90.0    | 89.2/83.2    | 69.0/66.7   | 89.0/93.4   | 85.1/83.3     |
> |           | SQP-CE        | 93.6/91.3    | 78.9/72.4    | 70.7/71.4   | 87.2/92.6   | 82.6/81.9     |
> |           | Single-Hinge  | 95.4/93.1    | 89.3/82.8    | 63.9/46.5   | 75.5/82.5   | 81.0/76.2     |
> |           | Single-CE     | 93.8/90.4    | 88.5/79.6    | 56.9/24.1   | 55.1/62.6   | 73.6/64.2     |
> | TREC07    | SiPBA-Hinge   | 68.9/16.8    | 93.7/89.7    | 57.0/33.7   | 50.5/57.6   | **67.5/49.5** |
> |           | SiPBA-CE      | 71.7/56.9    | 98.1/97.2    | 68.3/68.8   | 64.6/75.5   | **75.7/74.6** |
> |           | SQP-Hinge     | 68.9/17.2    | 95.3/92.5    | 55.0/21.0   | 29.9/28.1   | 62.3/39.7     |
> |           | SQP-CE        | 71.3/56.9    | 97.7/96.6    | 68.4/69.7   | 70.1/80.5   | 76.9/75.9     |
> |           | Single-Hinge  | 65.4/1.9     | 97.7/96.4    | 50.9/0.2    | 16.6/0.3    | 57.7/24.7     |
> |           | Single-CE     | 66.4/3.4     | 95.7/93.0    | 51.0/0.8    | 17.3/1.8    | 57.6/24.8     |
> | EnronSpam | SiPBA-Hinge   | 75.8/61.8    | 72.1/28.0    | 95.9/95.8   | 59.6/67.4   | **75.9/63.3** |
> |           | SiPBA-CE      | 76.3/62.8    | 74.0/34.4    | 95.2/95.0   | 64.0/72.0   | **77.4/66.1** |
> |           | SQP-Hinge     | 77.5/61.7    | 70.5/22.8    | 96.1/96.0   | 52.3/59.3   | 74.1/60.0     |
> |           | SQP-CE        | 76.0/62.6    | 73.4/32.9    | 94.9/94.8   | 63.0/71.0   | 76.8/65.3     |
> |           | Single-Hinge  | 76.8/56.0    | 69.3/15.0    | 95.8/95.6   | 47.2/52.3   | 72.3/54.7     |
> |           | Single-CE     | 76.4/55.4    | 70.0/19.2    | 95.6/95.3   | 43.1/46.9   | 71.3/54.2     |
> | LingSpam  | SiPBA-Hinge   | 63.4/59.1    | 66.2/51.2    | 71.1/65.4   | 99.4/99.6   | **75.0/68.8** |
> |           | SiPBA-CE      | 71.8/48.5    | 69.0/27.6    | 59.1/34.3   | 91.8/94.8   | **72.9/51.3** |
> |           | SQP-Hinge     | 42.5/53.8    | 45.3/52.0    | 72.5/65.8   | 98.2/99.0   | 64.6/67.7     |
> |           | SQP-CE        | 72.0/49.5    | 68.9/26.2    | 58.9/33.9   | 91.9/94.8   | 72.9/51.1     |
> |           | Single-Hinge  | 37.2/51.9    | 38.6/50.6    | 56.7/69.0   | 95.7/97.5   | 57.1/67.3     |
> |           | Single-CE     | 34.5/51.0    | 34.0/50.1    | 51.3/66.8   | 91.4/95.1   | 52.8/65.8     |
>
> ---
>
> This clarification and the full results will be incorporated into Section 5.2 of the revised manuscript.
>
> ---
>
> ### References
>
> [1] Z. Guan, D. Sow, S. Lin, and Y. Liang. *AdaProx: A novel method for bilevel optimization under pessimistic framework*. Conference on Parsimony and Learning, 2025.
> [2] I. Benchouk, L. O. Jolaoso, K. Nachi, and A. B. Zemkoho. *Scholtes relaxation method for pessimistic bilevel optimization*. Set-Valued and Variational Analysis, 2025.
> [3] M. Brückner and T. Scheffer. *Stackelberg games for adversarial prediction problems*. In International Conference on Knowledge Discovery and Data Mining, 2011.

---

### Official Review · Reviewer_iXKX · 2025-07-03

**Clarity:** 3
**Significance:** 3
**Originality:** 3
**Rating:** 4
**Confidence:** 3

**Summary:**

The paper addresses pessimistic bilevel optimization and introduces a single-loop algorithm based on a smooth approximation of a reformulated value function. The authors establish asymptotic convergence with respect to the smoothing parameter in the approximation and also provide a non-asymptotic convergence rate.

**Questions:**

1. What is the intuition behind this smoothing approach, which incorporates both a quadratic term and a bilinear term? Does this technique have precedent in the literature? What would happen if we only add $\sigma ||z||^2$ in the smoothing? Can the non-asymptotic results in Section 4 still hold? It seems in [28], they only used something similar to $\sigma ||z||^2$ for smoothing.

2. How do the results compare to those in reference [28]? Can authors add a remark under Theorem 4.2 on the choice of $s, p, q$ so that $||x^{k+1} - x^k||^2 / \alpha_k^2$ and $||u^k - u_k^* (x^k)||^2$ converge in the same order of rate?

3. The convergence notion in Theorem 4.2 is tied to the reformulation in (2) and smoothing approximation. Does this convergence imply $F(x^k , y^k)$ is close to $\min_{y \in S(x^k)} F(x^k , y)$? Are there other commonly used notions of convergence, and what is the relationship between these notions?

**Ethical Concerns:**

["NO or VERY MINOR ethics concerns only"]

**Final Justification:**

I would like to maintain the current rating. Authors should include additional remarks on Theorem 4.2 in the revised version.

**Limitations:**

N.A

**Paper Formatting Concerns:**

N.A

**Quality:**

3

**Strengths And Weaknesses:**

Strengths:

1. The algorithm is single-loop and gradient-based, which is easy to implement.

2. The idea of smoothing seems effective here.

Weaknesses:

1. The upper-level function $F(x, y)$ is also assume to be strongly convex in $y$. This assumption is usually not needed in (optimistic) bilevel optimization. I think the main reason for this assumption is that the unique $y_{\rho, \sigma}^* (x), z_{\rho, \sigma}^* (x)$ is needed in gradient of value function $ \phi_{\rho, \sigma}^* (x)$. Is there any way to remove this assumption?

2. The penalty method and the associated first-order updates are not particularly novel. However, the smoothing technique introduced here appears to be new.

---

> ### Author Rebuttal · Authors · 2025-07-30
>
> We thank the reviewer for their insightful feedback. We are grateful for their acknowledgment of our work's primary contribution: proposing a single-loop gradient-based method for Pessimistic Bilevel Optimization (PBO). We believe this is a valuable contribution to a relatively underexplored area.
>
> ---
>
> ### 1. On the Strong Concavity Assumption
>
> The reviewer is correct that the strong concavity assumption on the upper-level objective $F(x,y)$ is primarily used to ensure the uniqueness of the saddle point and the differentiability of the smoothed value function $\phi_{\rho,\sigma}(x)$. Without this, $\phi_{\rho,\sigma}(x)$ may become non-differentiable.
>
> One way to weaken this assumption to mere concavity in $y$ would be to add a regularization term, such as $-\delta_k \|\|y\|\|^2$ with vanishing $\delta_k$. We opted for the strong concavity assumption to maintain clarity and focus. Our goal was to present the core ideas of our new smoothing approximation and single-loop algorithm as clearly as possible. Introducing an additional hyperparameter would complicate the theoretical analysis and could obscure the central contributions of this work. We therefore made a deliberate choice to prioritize clarity in this paper, leaving the exploration of this relaxation for future work.
>
> ---
>
> ### 2. On the Novelty of the Proposed Method
>
> We appreciate  the reviewer for the recognition of the novelty of our smoothing strategy. While we agree that penalty methods and first-order updates are established techniques in optimization, we wish to clarify that their synthesis and application to create a single-loop, gradient-based method for PBO are novel.
>
> Specifically, these techniques are not merely combined, but are strategically integrated. The penalty strategy is a key component within our smoothing approximation framework, and the first-order updates are subsequently enabled by this new smoothing. It is this complete, integrated methodology that constitutes the main contribution and allows for an efficient single-loop PBO algorithm.
>
> ---
>
> ### 3. On the Intuition Behind the Smoothing Approach
>
> We thank the reviewer for this valuable question. The reviewer correctly notes that adding a quadratic term to ensure strong convexity/concavity is a common technique. Our key innovation, however, lies in the introduction of an additional bilinear term. To our knowledge, no prior work has incorporated both a quadratic and a bilinear term for smoothing a bilevel optimization problem.
>
> This bilinear term is not arbitrary; it is theoretically essential for the convergence of our smoothing approximation. As noted in the manuscript, without this coupling term, we cannot establish the key inequality in the proof of Lemma 2.2:
>
> $$
> \rho_{k}(f(\bar{x}, z_{k}^{\star}(\bar{x})) - f(\bar{x}, y_{k}^{\star}(\bar{x}))) + \frac{\sigma_k}{2} \|\| z_{k}^{\star}(\bar{x}) - y_{k}^{\star}(\bar{x})\|\|^2 \le 0.
> $$
>
> This result forms the foundation of our subsequent convergence analysis, underpinning both Proposition 2.3 and Theorem 2.5.
>
> The work in [28] considers a different setting and relies on an additional assumption on the lower-level Hessian (there exists a constant $\kappa > 0$ such that $\lambda_{\min}(\nabla^2_{yy} f(x,y)) > \kappa$ for all $\nabla_y g(x,y) \neq 0$), which our method does not require. This additional condition is what permits their use of a simpler smoothing term. Our approach, by design, is more general; the inclusion of the bilinear term is the critical element.
>
> ---
>
> ### 4. On the Comparison with [28] and Discussion on Convergence Rates
>
> We appreciate  the reviewer for this question. Our work differs from [28] in several aspects:
>
> * Assumptions: [28] requires stricter assumptions, including Lipschitz continuous Hessians and a strictly positive definite lower-level Hessian, none of which are needed for our method.
>
> * Methodology and Metric: AdaProx [28] follows a different path by reformulating the PBO problem via KKT conditions of the maximum subproblem, leading to a constrained single-level problem that is structurally distinct from our smoothing-based formulation. Consequently, their stationarity metric is different from ours, and their method relies on second-order information.
>
> Given these significant differences in assumptions and formulation, a direct theoretical comparison of the convergence rates would not be meaningful.
>
> For the question regarding the convergence rates of terms $\|\|x^{k+1} - x^k\|\|^2/\alpha_k^k$ and $\|\|u^k - u_{k}^{\star}(x^k)\|\|^2$ stated in Theorem 4.2, as it requires that $8p + 8q \le s $.
> Therefore, the convergence rate of the term $\|\|x^{k+1} - x^k\|\|^2/\alpha_k^2$ is at most $\mathcal{O}(1/K^{1-16p-16q})$. This is strictly slower than the $\mathcal{O}(1/K^{1-6p-7q})$ rate of the term $\|\| u^k -u^{\star}_k(x^k)\|\|^2$. We will add this clarification to the revised manuscript.
>
> ---
>
> ### 5. On the Convergence Notion in Theorem 4.2
>
> The reviewer's understanding is correct. We can indeed show from Theorem 4.2 that $y_k$ will eventually be close to the solution of $\max_{y \in \mathcal{S}(x_k)} F(x_k,y)$, and therefore $F(x^k,y^k) - \max_{y \in \mathcal{S}(x_k)} F(x_k,y)) \rightarrow 0$. More rigorously, under the assumptions of Theorem 4.2, we can establish that
>
> $$
> \lim_{k \to \infty} y_k^{\star}(x_k) = y^*(\bar{x}),
> $$
>
> where $y^*(\bar{x})=\underset{y\in \mathcal{S}(\bar{x})}{\arg\max} F(\bar{x},y)$. We will add this result to the revised manuscript.
>
> Regarding other convergence notions, the choice of a stationarity metric is tied to the specific reformulation an algorithm is based on. As the reviewer correctly notes, our metric is based on the reformulation in (2) and its smooth approximation in (6), whereas the metric for AdaProx [28] is based on a constrained single-level reformulation with KKT conditions of the max subproblem. Establishing a formal relationship between these distinct metrics is a non-trivial direction for future research, as it would likely require additional assumptions to bridge stationarity conditions of different problem formulations.

---

> > ### Comment · Reviewer_iXKX · 2025-08-05
> >
> > Thank the authors for detailed replies.
> >
> > 3. I understand that the bilinear term is critical for the asymptotic results discussed in Section 2. Is it also necessary for establishing the non-asymptotic rate presented in Section 4?
> >
> > 4. I suggest including a more detailed discussion in the manuscript comparing this work with related studies. What are the effects of s, p, q? Is smaller always better for these parameters?

---

> > > ### Author Response · Authors · 2025-08-06
> > >
> > > We sincerely thank the reviewer for their constructive feedback and insightful questions. We provide the following clarifications and will incorporate these discussions into the revised manuscript.
> > >
> > > **On the Necessity of the Bilinear Term**
> > >
> > > We thank the reviewer for this question. The bilinear term is indeed critical for ensuring the non-asymptotic convergence results presented in Section 4 are meaningful.
> > >
> > > To elaborate, if we were to formulate an approximate value function, denoted as $\varphi_{\rho_k,\sigma_k} (x)$,
> > >  without the bilinear term, it would be technically possible to adapt the proof techniques from Section 4 to show  non-asymptotic convergence results for the minimization problem $\min_x \varphi_{\rho_k,\sigma_k}(x)$.
> > > However, this convergence result would be misleading. Without the bilinear term,  $\varphi_{\rho_k,\sigma_k} (x)$ is no longer a valid approximation for the value function of the original PBO problem. Consequently, convergence results related to the minimum of this misspecified objective does not imply meaningful convergence results to the original PBO problem. The bilinear term is precisely what ensures that the sequence of solutions to our approximation problems converges to the solution of the PBO problem. Therefore, its inclusion is fundamental to the validity and significance of our non-asymptotic convergence analysis. Furthermore, as a direct consequence of its absence, the key result presented in Corollary 4.3 would no longer hold. We will add a discussion to the revised manuscript .
> > >
> > >  **On the Effects Parameters $s,p,q$**
> > >
> > > We thank the reviewer for this valuable suggestion. We will incorporate a more detailed discussion on the role of these parameters and their interplay into the revised manuscript.
> > >
> > > The parameters s, p, and q collectively manage the fundamental trade-off between the quality of the value function approximation and the step size of the iterates. Their roles are as follows:
> > >
> > > * Parameters $p$ and $q$ govern the evolution of the approximate value function $\phi_{\rho_k,\sigma_k}(x)$. Specifically, $p$ controls the growth rate of the penalty coefficient $\rho_k=\rho_0k^p$, and $q$ controls the decay rate of the regularization coefficient $\sigma_k=\sigma_0k^{-1}$. Larger values of $p$ and $q$ cause the approximate value function to approach the true value objective more rapidly.
> > >
> > > * Parameter $s$ controls the decay rate of the step size $\alpha_k = \alpha_0 k^{-s}$ for updating the iterate $x^k$.
> > >
> > > These parameters are linked by the theoretical condition $s \geq 8p + 8q$,  which reveals a crucial trade-off:
> > >
> > > * Large $p,q$: Choosing large values for $p$ and $q$ accelerates the convergence of $\phi_{\rho_k,\sigma_k}(x)$ to the true value function. However, to satisfy the condition, this necessitates a larger value for $s$. A larger $s$ leads to a smaller step size $\alpha_k $, which can slow down the convergence of the iterates $x^k$.
> > >
> > > * Small $p,q$: Choosing small values for $p$ and $q$ allows for a smaller $s$, which in turn permits a larger step size $\alpha_k $ and potentially faster convergence of  $x^k$. The drawback is that the approximate objective $\phi_{\rho_k,\sigma_k}(x)$ converges to the true value function more slowly, which can negatively impact the overall performance of the algorithm.
> > >
> > > Therefore, to answer the reviewer's question directly, smaller values for these parameters are not universally better. A delicate balance must be struck. In practice, this trade-off requires careful tuning. For our experiments, we adopted the setting $p=q=0.01, s=0.16$, which satisfies the theoretical condition and demonstrated good empirical performance. We will include a detailed discussion of this trade-off in the revised manuscript as suggested.

---

### Official Review · Reviewer_ckLr · 2025-07-04

**Clarity:** 3
**Significance:** 2
**Originality:** 2
**Rating:** 4
**Confidence:** 4

**Summary:**

This work focuses on addressing the relatively underexplored area of Pessimistic Bilevel Optimization (PBO) by proposing a single-loop, fully first-order algorithm. Specifically, it introduces a smooth approximation to the PBO problem through penalization and regularization, and builds on this to develop SiPBA, a gradient-based method that avoids second-order derivatives and inner-loop optimization. Theoretical analysis is provided to justify the approximation and prove the convergence of SiPBA.

**Questions:**

My questions are based on the weakness:

In the recent literature on Optimistic Bilevel Optimization, it has been shown that the lower-level optimization problem can be viewed as a soft constraint to the upper-level. In this work, since some approximation and smoothing techniques are adopted, does the lower-level optimization problem in Pessimistic Bilevel Optimization also act as a soft constraint to the upper-level problem?

Although the proposed framework is claimed to be single-loop, each iteration involves multiple gradient projection steps. Given that the projection in this work is applied to the main variables (which are often non-convex and complex), does this still qualify the framework as fully single-loop in practice? Could the authors provide more discussion or clarification on this point?

Assumption 3 appears to be uncommon in the context of bilevel optimization. Could the authors elaborate on the motivation and practicality of this assumption? In particular, are there empirical results or real-world scenarios that support its validity?

How does the proposed method compare with existing bilevel optimization methods in terms of non-asymptotic convergence rates? Can the authors provide a theoretical or empirical comparison?

The experimental validation appears limited, and the practical applications of Pessimistic Bilevel Optimization in machine learning are not well demonstrated. Could the authors include more experiments on additional tasks or provide further discussion on the potential applications of PBO in the introduction?

**Ethical Concerns:**

["NO or VERY MINOR ethics concerns only"]

**Final Justification:**

My concerns have been addressed. I’ve increased the confidence from 3 to 4.

**Limitations:**

yes

**Quality:**

3

**Strengths And Weaknesses:**

Strengths:

1. The studied problem is interesting.

2. The presentation in this work is good.

Weaknesses:

1. The recent work on Optimistic Bilevel Optimization has shown that the lower-level optimization problem is a soft constraint to the upper-level ones. Whether the lower-level optimization problem is also a soft constraint in Pessimistic Bilevel Optimization? Since some approximation and smoothing tricks are used in this manuscript, and the soft constraint means that this constraint can be violated to some extent without resulting in a meaningless solution.

2. Although the framework proposed in this work is single-loop. Each step in this framework is computationally complex since multiple gradient projection steps are used. I understand that the gradient projection step is efficient if the feasible region set is simple, for instance, the feasible region for some dual variables. However, the projection set in this algorithm is for the main variables, which are usually complex and non-convex. Operating the projection in each iteration needs to solve a non-convex optimization problem. Thus, more discussions on whether the proposed method is fully single-loop need to be added.

3. The Assumption 3 is not a widely-used assumption in bilevel optimization, can you provide more discussions on this assumption? Additionally, whether the assumption is practical? Can you provide more results, e.g., experimental results, to support this assumption?

4. Can you compare the non-asymptotic convergence rates of the proposed method with the existing bilevel optimization methods?

5. The experimental results are limited. It seems that PBO has limited applications in machine learning. Can you provide more experimental results on additional application? More discussions on the applications of PBO in the introduction can also improve this manuscript. The reviewer will appreciate this work not only for its theoretical analysis of the optimization problem, but also for the practical relevance of the studied problem, which has broad applications in machine learning (since NeurIPS is a ML conference).

---

> ### Author Rebuttal · Authors · 2025-07-30
>
> We thank the reviewer for their insightful comments and constructive suggestions. We are grateful for their acknowledgment of our work's primary contributions: proposing a gradient-based method for Pessimistic Bilevel Optimization (PBO) that avoids second-order derivatives and the need for an inner-loop optimization solver. We believe this is a valuable contribution to a relatively underexplored area.
>
> Below, we address the specific concerns raised.
>
> ---
>
> ### 1. On the Lower-Level Problem as a Soft Constraint
>
> Regarding the question of whether the lower-level optimization acts as a soft constraint, the reviewer's intuition is correct. This result is established in Theorem 3.6 of [1]. The theorem shows that if the lower-level problem constraint is relaxed to $f(x,y) \le \inf_y f(x,y) + \epsilon$, the solution to this relaxed problem converges to the solution of the original PBO as the violation $\epsilon \rightarrow 0$. We will add this interpretation in the revised manuscript.
>
> ---
>
> ### 2. On the "Single-Loop" Nature of the Algorithm
>
> We appreciate the opportunity to clarify our "single-loop" claim. Our use of this term emphasizes that our algorithm does not require an inner loop to iteratively solve the minimax problem in Eq. (4) to approximate the gradient of $\phi_{\rho,\sigma}$ at each iteration. We agree that this advantage relies on the assumption that the projections onto the feasible sets $X$ and $Y$ are computationally simple (i.e., have a closed-form solution or can be computed efficiently). We will add a statement to the manuscript to explicitly clarify this condition for the single-loop claim.
>
> ---
>
> ### 3. On the Justification for Assumption 3
>
> We agree that Assumption 3 is not a commonly used assumption in bilevel optimization. In fact, Assumption 3 is equivalent to the closedness of the function’s epigraph, and it is a common assumption for ensuring the existence of a function's minimum (see, e.g., Theorem 1.9 of [2]).
>
> We recognize that verifying Assumption 3 directly can be challenging. For this reason, we provide a more practical sufficient condition immediately following the assumption: the inner semi-continuity of the lower-level solution map $\mathcal{S}(x)$. This condition holds for all problem instances considered in our numerical experiments under mild assumptions, thereby satisfying Assumption 3. We will expand this discussion in the revised manuscript to make the motivation and verification of this assumption clearer.
>
> ---
>
> ### 4. On the Comparison of Non-Asymptotic Convergence Rates
>
> We thank the reviewer for raising this important point. To the best of our knowledge, AdaProx [3] is the only other algorithm for PBO with a non-asymptotic convergence rate analysis. However, a direct theoretical comparison of our rate with that of AdaProx is not straightforward for several fundamental reasons:
>
> * Different Assumptions: AdaProx requires additional assumptions, including Lipschitz continuity of the Hessians for both objective functions and the strict positive definiteness of the lower-level Hessian, none of which are required by our method.
> * Different Methodology and Metric: AdaProx reformulates the PBO problem using a relaxed KKT system, which leads to a different stationary metric for measuring convergence.
>
> Given these differences, a direct comparison of the theoretical convergence rates would not be meaningful.
>
> To provide a comprehensive  numerical comparison, we have expanded our numerical evaluation on the synthetic example from Section 5.1 to include AdaProx-SG and the Compact/Detailed Scholtes (Scholtes-C/D) relaxation methods [4]. We ran SiPBA, AdaProx-PD, We report the minimum and maximum relative errors of the outputs, the number of valid runs (achieving $\epsilon_{\text{rel}} < 10^{-4}$), and the average runtime (to reach $\epsilon_{\text{rel}} < 10^{-4}$)  of those valid runs. The results over10 times from random initial points, summarized in the table below, demonstrate that SiPBA is not only significantly faster but also more robust, consistently converging to a valid solution.
>
> **Table**: Performance comparison of SiPBA, AdaProx variants, and Scholtes relaxations on synthetic experiment ($n=100$).
> |                               | SiPBA               | AdaProx-PD          | AdaProx-SG          | Scholtes-C          | Scholtes-D          |
> | :---------------------------- | :------------------ | :------------------ | :------------------ | :------------------ | :------------------ |
> | Min. $(\epsilon_{\text{rel}})$ | $1.22 \times 10^{-6}$ | $1.79 \times 10^{-7}$ | $3.80 \times 10^{-6}$ | $1.12 \times 10^{-5}$ | $9.60 \times 10^{-5}$ |
> | Max. $(\epsilon_{\text{rel}})$ | $1.45 \times 10^{-6}$ | $1.53 \times 10^{-5}$ | $1.02 \times 10^{-4}$ | $0.10$              | $1.06$              |
> | Valid Runs                    | 10/10               | 10/10               | 9/10                | 1/10                | 1/10                |
> | Ave. Time (s)                 | 1.03                | 87.44               | 4.07                | 23.12               | 23.81               |
>
>
>
> ---
>
> ### 5. On Experimental Scope and Practical Applications
>
> We appreciate the feedback regarding the experimental results and the practical applications of PBO. A key motivation for our work is to provide an efficient and accessible algorithm for PBO, thereby encouraging its adoption in a wider range of machine learning applications.
>
> To further demonstrate the practical utility of SiPBA, we have added a new experiment on deep hyper-representation using real-world datasets. The problem is formulated as:
>
> $$
> \min_{\theta\in\Theta}\max_{w}\frac{1}{m_1}\|\|f(X_{val},\theta)w-\mathbf{y}_{val}\|\|^2,
> $$
>
> $$
> \text{s.t.} w\in\underset{w^\prime\in\mathcal{W}}{\arg\min} \frac{1}{m_2}\|\|f(X_{train},\theta)w^\prime-\mathbf{y}_{train}\|\|^2,
> $$
>
> where $f(\cdot, \theta)$ represents a neural network parameterized by $\theta$, and $w$ corresponds to a linear layer. We adopt the LeNet-5 architecture [5] as the feature extractor $f(\cdot, \theta)$ and evaluate performance on the MNIST and FashionMNIST datasets. Each dataset is randomly split into 50,000 training samples, 10,000 validation samples, and 10,000 test samples, with performance evaluated based on test accuracy. The accuracy on test data are reported below:
>
> **Table**:Test accuracy (%) on MNIST and Fashion MNIST.
> | Dataset       | SiPBA          | PZOBO | AID-FP | AID-CG |
> | :------------ | :------------- | :---- | :----- | :----- |
> | MNIST         | **99.1** | 95.8  | 97.6   | 92.5   |
> | Fashion-MNIST | **89.6** | 78.8  | 82.1   | 78.8   |
>
>
> These results indicate that the PBO model optimized by SiPBA exhibits superior generalization performance, further demonstrating the promise of PBO in machine learning applications.
>
> Actually, a growing body of work has explored the potential of PBO in various machine learning applications recently. We will add a discussion to the revised manuscript.
> Specifically, adversarial learning has been formulated as a PBO problem [6,7], and decision-focused learning in contextual optimization has been addressed within the PBO framework [8,9]. The use of PBO for hyperparameter optimization has also been investigated [10]. Outside the machine learning domain, PBO has found applications in many practical scenarios, including but not limited to demand response management [11], rank pricing and second-best toll pricing [12,13], production-distribution planning [14], and gene knockout model in biological systems [15].
>
> ---
>
> ### References
>
> [1] L. Lampariello, S. Sagratella and O. Stein. The standard pessimistic bilevel problem. *SIAM Journal on Optimization*, 2019.
>
> [2] R. T. Rockafellar, and R. J. Wets. Variational analysis. *Springer Berlin Heidelberg*,1998.
>
> [3] Z. Guan, D. Sow, S. Lin, and Y. Liang. AdaProx: A novel method for bilevel optimization under pessimistic framework. *Conference on Parsimony and Learning*, 2025.
>
> [4] I. Benchouk, L. O. Jolaoso, K. Nachi, and A. B. Zemkoho. Scholtes relaxation method for pessimistic bilevel optimization. *Set-Valued and Variational Analysis*, 2025.
>
> [5] D. Sow, K. Ji, and Y. Liang. On the convergence theory for hessian-free bilevel algorithms. *Advances in Neural Information Processing Systems*, 2022.
>
> [6] M. Brückner and T. Scheffer. Stackelberg games for adversarial prediction problems. *International Conference on Knowledge Discovery and Data Mining*, 2011.
>
> [7] D. Benfield, S. Coniglio, M. Kunc, P. T. Vuong and A. B. Zemkoho. Classification under strategic adversary manipulation using pessimistic bilevel optimisation. In *arXiv:2410.20284*, 2024.
>
> [8] V. Bucarey, S. Calderón, G. Muñoz, and F. Semet. Decision-focused predictions via pessimistic bilevel optimization: A computational study. *Integration of Constraint Programming, Artificial Intelligence, and Operations Research*, 2024.
>
> [9] D. Jiménez, B. K. Pagnoncelli and H. Yaman. Pessimistic bilevel optimization approach for decision-focused learning. In *arXiv:2501.16826*, 2025.
>
> [10] M. A. Ustun, L. Xu, B. Zeng, and X. Qian. Hyperparameter Tuning Through Pessimistic Bilevel Optimization. In *arXiv:2412.03666*, 2024.
>
> [11] T. Kis, A. Kovács, and C. Mészáros. On optimistic and pessimistic bilevel optimization models for demand response management. *Energies*, 2021.
>
> [12] H.I. Calvete, C Galé, A Hernández and J. A. Iranzo. A novel approach to pessimistic bilevel problems. An application to the rank pricing problem with ties. In *Optimization*, 2024.
>
> [13] X. Ban, S. Lu, M.Ferris and H. X. Liu. Risk averse second best toll pricing. In *Transportation and Traffic Theory 2009: Golden Jubilee*, 2009.
>
> [14] Y. Zheng, G. Zhang, J. Han and J. Lu. Pessimistic bilevel optimization model for risk-averse production-distribution planning. In *Information Sciences*, 2016.
>
> [15] B. Zeng. A practical scheme to compute the pessimistic bilevel optimization problem. *INFORMS Journal on Computing*, 2020.

---

> > ### Comment · Reviewer_ckLr · 2025-08-03
> >
> > Thanks for the rebuttal. Some of my concerns have been addressed. However, I still have some concerns.
> >
> > - More discussions about the projection steps and single-loop nature used in this work should be added.
> >
> > - More explanations and discussions about Assumption 3 are required to be added.
> >
> > - Regarding your new experiment on deep hyper-representation, could you elaborate on the motivation behind this task? Additionally, what is the rationale for formulating it as a PBO problem, and why is PBO particularly suitable for solving it?

---

> > > ### Author Response · Authors · 2025-08-03
> > >
> > > We thank the reviewer for the constructive follow-up comments. As suggested, we will incorporate the detailed discussions from our rebuttal regarding the projection steps, the single-loop nature of our algorithm, and Assumption 3 into the revised manuscript to enhance its clarity.
> > >
> > > We appreciate the opportunity to clarify our experiment on  hyper-representation. Our setup is adapted from the "Learning Robust Hyper-representation" task proposed in [2], which formulates the problem using a PBO model.
> > >
> > > The broader task of learning hyper-representations was first formulated as an optimistic bilevel optimization problem in [1], with the goal of finding an optimal representation mapping (parameterized by $\theta$) for ground-level classifiers (parameterized by $w$). The authors of [2] proposed using a PBO model specifically to learn robust hyper-representations. The rationale for this PBO model, as discussed in [2], is its ability to handle the potential multiplicity of optimal solutions in the lower-level problem. By finding a representation $\theta$ that performs well even under a worst-case choice of ground classifier weights $w^*$ (the lower-level problem), the PBO model ensures the resulting hyper-representation is robustly effective.
> > >
> > > While the experiments in [2] were limited to a special case where the function $f(\cdot, \theta)$ is linear with respect to $\theta$, our new experiments demonstrate the effectiveness of our proposed SiPBA in a setting where $f(\cdot, \theta)$ is a non-linear neural network.
> > > This demonstrate the practical effectiveness of SiPBA on  complex problems.
> > >
> > > [1] L. Franceschi, P. Frasconi, S. Salzo, R. Grazzi, and M. Pontil. Bilevel programming for hyperparameter optimization and meta-learning, International Conference on Machine Learning, 2018
> > >
> > > [2] Z. Guan, D. Sow, S. Lin, and Y. Liang. AdaProx: A novel method for bilevel optimization under pessimistic framework. Conference on Parsimony and Learning, 2025.

---

> > > > ### Comment · Reviewer_ckLr · 2025-08-04
> > > >
> > > > Thanks for the replies. I suggest the authors provide and add these discussions in the revised manuscript.
> > > >
> > > > After discussing with the authors, my concerns have been addressed.

---

> > > > > ### Author Response · Authors · 2025-08-04
> > > > >
> > > > > We thank the reviewer for the valuable feedback and are pleased that all concerns have been addressed. As suggested, we will incorporate the discussions on deep hyper-representation into the revised manuscript to enhance its clarity.

---

### Decision · Program_Chairs · 2025-09-17

**Decision:**

Accept (poster)

**Comment:**

The work studies the underexplored setting of pessimistic bilevel optimization. The authors propose a single-loop first-order gradient algorithm based on a smooth approximation of the PBO value function. The theoretical contributions include the design of a penalization–regularization-based smooth approximation and a convergence analysis of SiPBA with non-asymptotic guarantees. The algorithm avoids computing exact second-order derivatives, which makes it appealing from a computational standpoint. Experiments on synthetic problems, spam classification, and hyper-representation learning illustrate its feasibility and efficiency, with comparisons to existing methods such as AdaProx.

The main strength of the work lies in tackling a relatively less studied PBO problem through a first-order and single-loop approach. The theoretical analysis is carefully developed, the algorithm is easy to implement, and the smooth approximation technique represents the core technical contribution. The empirical results, though limited, provide some evidence of the method’s effectiveness and scalability.

Nevertheless, there are still some weaknesses. The theoretical framework relies on restrictive assumptions (e.g., strong convexity/concavity requirements). The smooth approximation depends heavily on heuristic penalization and regularization terms, which, while functional, limit the sense of conceptual novelty. The theoretical results are also confined to stationarity. The experimental validation, though expanded in the rebuttal, remains modest in scope and does not convincingly demonstrate broad applicability or real-world advantages.

In the rebuttal, the authors clarified the role of assumptions, expanded their discussion of the single-loop claim, and added experiments on hyper-representation learning. These efforts improved clarity and provided additional empirical evidence, but the fundamental limitations remain: restrictive conditions, the heuristic nature of the approximation, and limited experimental scope.

In summary, the paper makes a technically solid and timely contribution to the niche area of pessimistic bilevel optimization, offering a new algorithmic perspective. Its novelty lies primarily in the problem framing and smooth approximation. The work is sound and has value, though its scientific depth is limited.